# SUN-DSBO: A Structured Unified Framework for Nonconvex Decentralized Stochastic Bilevel Optimization

## Abstract

Decentralized stochastic bilevel optimization (DSBO) is a powerful tool for various machine learning tasks, including decentralized meta-learning and hyperparameter tuning. Existing DSBO methods primarily address problems with strongly convex lower-level objective functions. However, nonconvex objective functions are increasingly prevalent in modern deep learning. In this work, we introduce SUN-DSBO, a Structured Unified framework for Nonconvex DSBO, in which both the upper- and lower-level objective functions may be nonconvex. Notably, SUN-DSBO offers the flexibility to incorporate decentralized stochastic gradient descent or various techniques for mitigating data heterogeneity, such as gradient tracking (GT). We demonstrate that SUN-DSBO-GT, an adaptation of the GT technique within our framework, achieves a linear speedup with respect to the number of agents. This is accomplished without relying on restrictive assumptions, such as gradient boundedness or any specific assumptions regarding gradient heterogeneity. Numerical experiments validate the effectiveness of our method.

## 1 Introduction

Decentralized stochastic bilevel optimization (DSBO) offers an effective framework for multiple agents collaborating to solve nested optimization problems over decentralized communication networks. It has garnered increasing interest, with numerous applications in decentralized meta-learning (Kayaalp et al., 2022; Yang & Kwok, 2024), multi-agent reinforcement learning (Li et al., 2023; Zheng & Gu, 2024), hyperparameter tuning (Yang et al., 2022; Zhu et al., 2024), and decentralized adversarial training (Sinha et al., 2018; Liu et al., 2020). In distributed optimization, the original centralized optimization faces a bottleneck communication problem, particularly when the network is large (Lian et al., 2017).

In this work, we consider the following DSBO problem over a decentralized communication network of $n$ agents:

$$\min_{x \in \mathbb{R}^{d_x}, y \in \mathbb{R}^{d_y}} F(x, y) := \frac{1}{n} \sum_{i=1}^{n} f_i(x, y) \qquad \text{(upper-level)}$$

$$\text{s.t.} \quad y \in \underset{\tilde{y} \in \mathbb{R}^{d_y}}{\operatorname{argmin}} \, G(x, \tilde{y}) := \frac{1}{n} \sum_{i=1}^{n} g_i(x, \tilde{y}), \qquad \text{(lower-level)} \tag{1}$$

where the local (private) objective functions $f_i$ and $g_i$ for the $i$-th agent are defined as:

$$f_i(x, y) := \mathbb{E}_{\xi_{f,i}}[f_i(x, y; \xi_{f,i})], \quad g_i(x, y) := bbE_{\xi_{g,i}}\big[g_i(x, y; \xi_{g,i})\big].$$

Above, we do not assume that $g_i(x, y)$ is strongly convex in $y$, which is a common assumption in most existing DSBO methods. Consequently, the model in problem (1) can accommodate a wide range of practical scenarios in which the lower-level problem is nonconvex. Since the lower-level optimal solution may not be unique, the model in (1) adopts the framework of optimistic bilevel optimization (Liu et al., 2021; Zhang et al., 2024).

When the lower-level problem is strongly convex, it admits a unique optimal solution, denoted by $y^*(x)$, which is smooth under mild smoothness assumptions. In such cases, by employing

various decentralized strategies to efficiently estimate the gradient of $F(x, y^*(x))$ (referred to as the hypergradient in bilevel optimization), the DSBO problem can be tackled using several efficient methods. These methods can generally be categorized into two approaches: the approximate implicit differentiation (AID)-based approach (Yang et al., 2022; Lu et al., 2022; Chen et al., 2023; 2025; Gao et al., 2023a; Dong et al., 2023; Chen & Wang, 2024; Kong et al., 2025; Zhu et al., 2024) and the value function-based approach (Wang et al., 2024). For instance, Zhu et al. (2024) propose SPARKLE, a unified single-loop primal-dual framework for DSBO problems, and explore the advantages of various strategies (including mixed strategies) for mitigating data heterogeneity within SPARKLE, such as gradient tracking (GT) (Xu et al., 2015; Lorenzo & Scutari, 2016; Nedic et al., 2017), EXTRA (Shi et al., 2015), and Exact-Diffusion (ED) (Yuan et al., 2018; Li et al., 2019; Yuan et al., 2020). SPARKLE is an AID-based method that utilizes second-order derivative information to approximate the hypergradient. Another line of research has focused on addressing DSBO problems using only first-order derivative information. For example, Wang et al. (2024) propose DSGDA-GT, a novel algorithm that requires only first-order oracles with GT to solve the DSBO problem.

Although the above DSBO methods are developed from the perspective of the hypergradient, they can also be applied to DSBO problems with potentially nonconvex lower-level objectives, albeit without theoretical guarantees. Therefore, a natural and practical question arises: *Can we develop an efficient algorithm with provable convergence for nonconvex decentralized stochastic bilevel optimization?*

## 1.1 First attempt: regularization

Observe that most existing DSBO methods incorporate a regularization term into the lower-level objective in several experiments to enhance convexity. This observation motivates a straightforward approach to the problem at hand: when dealing with a learning task formulated as a DSBO problem with a nonconvex lower-level objective, one may first add a regularization term to its lower-level objective and then apply an existing DSBO algorithm. This approach raises several important questions: Does it work effectively? What are the implications of introducing such regularization?

We present our main findings through illustrative experiments. For the setting, we adopt the widely used data hyper-cleaning task on a corrupted FashionMNIST dataset, employing a two-layer MLP model for training (Kong et al., 2025; Zhu et al., 2024). A quadratic regularization term, $\alpha\|y\|^2$, is consistently incorporated into the lower-level objective. In the experiments, we gradually increase the regularization parameter $\alpha$ from 0.001 to 0.1. The test accuracy of two representative algorithms, SPARKLE-GT (Zhu et al., 2024) and DSGDA-GT (Wang et al., 2024), is presented in Figure 1. These results highlight a key dilemma in regularization parameter tuning: *a large $\alpha$ leads to worse performance, while a small $\alpha$ does not guarantee strong convexity.* Consequently, a significant gap remains between the study of DSBO in strongly convex and nonconvex settings.

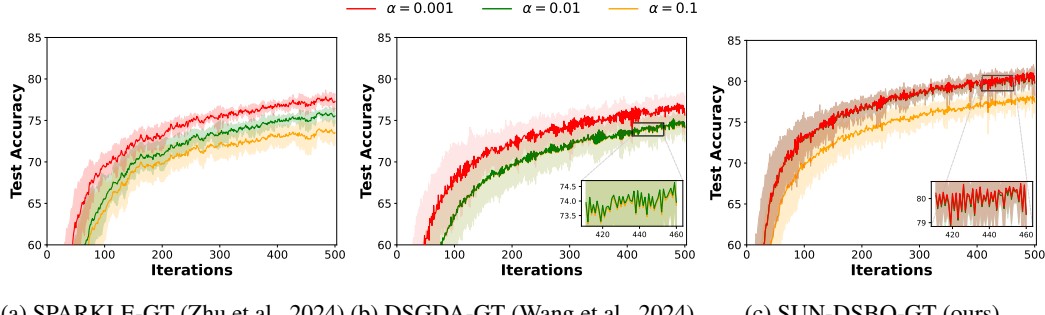

(a) SPARKLE-GT (Zhu et al., 2024) (b) DSGDA-GT (Wang et al., 2024) (c) SUN-DSBO-GT (ours)

Figure 1: Comparison of the algorithms on hyper-cleaning with different regularization parameters.

## 1.2 Main contributions

The goal of this work is to advance the current understanding of solving DSBO problems with strongly convex lower-level structures to encompass a broader class that includes potentially nonconvex lower-level problems. The main contributions are summarized as follows:

- We introduce SUN-DSBO, a Structured Unified framework for Nonconvex DSBO, in which the update directions for the variables depend linearly on both the upper- and lower-level objective functions. This design supports various decentralized strategies, stochastic estimators, and techniques for mitigating data heterogeneity. Specifically, within this framework, we propose SUN-DSBO-SE, an adaptation of decentralized stochastic gradient descent, and SUN-DSBO-GT, an adaptation of the GT technique.

- Theoretically, we provide a finite-time convergence analysis for SUN-DSBO-SE/GT under more relaxed assumptions than those required by existing methods. In particular, we show that SUN-DSBO-GT achieves linear speedup with respect to the number of agents, without relying on restrictive conditions such as gradient boundedness or specific assumptions about gradient heterogeneity. A detailed comparison is provided in Table 1 in Appendix A.2.

- We conduct numerical experiments to compare the proposed algorithms with state-of-the-art baselines and perform ablation studies for SUN-DSBO-SE and SUN-DSBO-GT. The empirical results demonstrate the effectiveness of our approach (see, e.g., Figure 1).

**Related works.** In general, both centralized and decentralized bilevel optimization problems with nonconvex or non-strongly convex lower-level (LL) objectives are computationally intractable without additional assumptions or relaxations (Kwon et al., 2024). In centralized bilevel optimization, various techniques have been proposed to address the inherent difficulty, including AID-based methods (Huang, 2023; Xiao et al., 2023) and value function-based penalty methods (Liu et al., 2022; Shen & Chen, 2023) under the LL Polyak-Łojasiewicz (PL) condition, as well as Moreau envelope-based penalty methods (Liu et al., 2024) for problems with general nonconvex LL objectives. Despite advances in centralized bilevel optimization, developing efficient and provably convergent algorithms for DSBO is far from a straightforward extension of their centralized counterparts, as it requires addressing data heterogeneity and ensuring consensus among multiple agents (see, e.g., (Yang et al., 2022; Zhu et al., 2024; Wang et al., 2024)). Most existing DSBO methods assume that the lower-level objectives are strongly convex. Recently, Qin et al. (2025) investigated decentralized bilevel optimization in a personalized DSBO setting, relaxing the strong convexity assumption to mere convexity. The key idea is to introduce a diminishing quadratic regularization to the LL objective in order to obtain a good approximation and retain the uniqueness of the solution to the augmented LL problem. Consequently, convexity remains essential in their approach, which cannot be directly applied to our problem (1) with potentially nonconvex lower-level objectives. Building upon the Moreau envelope–based penalty method by Liu et al. (2024), we propose an alternative procedure. Further details are provided in Appendix E, which discusses the challenges and nontrivial aspects of our analysis to underscore our contributions. A comprehensive review of additional relevant works on centralized and decentralized bilevel optimization is presented in Appendix A.

## 2 PROPOSED METHOD

### 2.1 PRELIMINARIES

This work builds upon the Moreau envelope-based penalty method proposed in Liu et al. (2024) for centralized deterministic bilevel optimization. First, problem (1) can be reformulated as follows:

$$\min_{(x,y)\in\mathbb{R}^{d_x}\times\mathbb{R}^{d_y}} F(x,y) \quad \text{s.t.} \quad G(x,y) - V_\gamma(x,y) \leq 0, \tag{2}$$

where $V_\gamma$ is the Moreau envelope of $G$, defined as

$$V_\gamma(x,y) := \min_{\theta\in\mathbb{R}^{d_y}} \left\{ G(x,\theta) + \frac{1}{2\gamma}\|\theta - y\|^2 \right\}, \quad \text{for } \gamma > 0. \tag{3}$$

Since $G(x,y) - V_\gamma(x,y) \geq 0$ by the definition of $V_\gamma$, we follow Kwon et al. (2024); Liu et al. (2024) and consider the following penalty formulation to solve the constrained optimization problem (2):

$$\min_{(x,y)\in\mathbb{R}^{d_x}\times\mathbb{R}^{d_y}} \Psi_\mu(x,y) := \mu F(x,y) + G(x,y) - V_\gamma(x,y), \tag{4}$$

where $\mu > 0$ is either sufficiently small or gradually diminishing.

We observe that this reformulation possesses two favorable properties that facilitate the development and analysis of efficient algorithms for nonconvex DSBO.

**(i)** When $G(x, \cdot)$ is $L_2$-smooth for any $x$, then for each $\gamma \in (0, \frac{1}{2L_2})$, the reformulated problem (2) is equivalent to the problem: $\min_{x,y} F(x, y)$ s.t. $\nabla_y G(x, y) = 0$. Notably, this formulation is independent of $\gamma$ and remains equivalent to the original problem under either convexity or the PL condition of the lower-level objective (see Theorem A.1 of Liu et al. (2024)). More detailed discussions of the relationship between problem (1) and problem (4) is provided in Appendix D.1.

**(ii)** Unlike Liu et al. (2024), we further reformulate problem (4) equivalently as the min-max problem:

$$\min_{(x,y) \in \mathbb{R}^{d_x} \times \mathbb{R}^{d_y}} \max_{\theta \in \mathbb{R}^{d_y}} \left\{ \mu F(x, y) + G(x, y) - G(x, \theta) - \frac{1}{2\gamma} \|\theta - y\|^2 \right\}. \tag{5}$$

Notably, under the condition in (i), this is a nonconvex–strongly concave minimax optimization problem. Moreover, the objective function depends linearly on both the upper- and lower-level objectives of problem (1), which facilitates the derivation of stochastic estimates. Recall that $F(x, y) := \frac{1}{n} \sum_{i=1}^{n} f_i(x, y)$, $G(x, y) := \frac{1}{n} \sum_{i=1}^{n} g_i(x, y)$ in problem (5), where $f_i(x, y) := \mathbb{E}_{\xi_{f,i}}[f_i(x, y; \xi_{f,i})]$, and $g_i(x, y) := \mathbb{E}_{\xi_{g,i}}[g_i(x, y; \xi_{g,i})]$.

## 2.2 A UNIFIED FRAMEWORK FOR DECENTRALIZED STOCHASTIC BILEVEL OPTIMIZATION

Since problem (4) serves as an approximation of problem (2) as $\mu \to 0$ (see, e.g., Theorem A.3 in Liu et al. (2024)), decentralized minimax optimization methods (e.g., Liu et al. (2020); Xian et al. (2021)) can be directly applied to the min-max reformulation (5) with a small value of $\mu$. However, this approach may result in slower convergence (see, e.g., Kwon et al. (2023; 2024) for centralized bilevel optimization). To address this, we instead gradually decrease the penalty parameters $\{\mu_k\}$, with $\mu_k \to 0$ as the iteration index $k$ increases.

For the $k$-th iteration, let $x_i^k, y_i^k$, and $\theta_i^k$ represent the local primal and dual variables maintained by the $i$-th agent. Suppose the communication network is described by a weight matrix $\mathbf{W} = (w_{ij}) \in \mathbb{R}^{n \times n}$, where $w_{ij} = 0$ if agents $i$ and $j$ are not connected. We now present **SUN-DSBO**, a Structured Unified framework for Nonconvex Decentralized Stochastic Bilevel Optimization. In this framework, each local agent repeatedly performs the following steps:

(I) **Stochastic Gradient Computation:** Each agent $i$ computes unbiased or biased stochastic estimators $\hat{D}_{\theta,i}^k, \hat{D}_{x,i}^k$, and $\hat{D}_{y,i}^k$ for the descent directions $\tilde{D}_{\theta,i}^k, \tilde{D}_{x,i}^k$, and $\tilde{D}_{y,i}^k$, as derived from the min-max reformulation (5). These estimators are calculated using either vanilla mini-batch gradients or advanced techniques that incorporate acceleration and variance reduction:

$$\tilde{D}_{\theta,i}^k = \nabla_y g_i(x_i^k, \theta_i^k) + \frac{1}{\gamma}(\theta_i^k - y_i^k), \tag{6a}$$

$$\tilde{D}_{x,i}^k = \mu_k \nabla_x f_i(x_i^k, y_i^k) + \nabla_x g_i(x_i^k, y_i^k) - \nabla_x g_i(x_i^k, \theta_i^k), \tag{6b}$$

$$\tilde{D}_{y,i}^k = \mu_k \nabla_y f_i(x_i^k, y_i^k) + \nabla_y g_i(x_i^k, y_i^k) - \frac{1}{\gamma}(y_i^k - \theta_i^k). \tag{6c}$$

(II) **Gradient Estimator Update:** Communicate with neighbors and update the gradient estimators $\hat{D}_{\theta,i}^k, \hat{D}_{x,i}^k$, and $\hat{D}_{y,i}^k$ to $D_{\theta,i}^k, D_{x,i}^k$ and $D_{y,i}^k$ using decentralized techniques such as GT, EXTRA, and Exact-Diffusion (ED), as well as mixing strategies (Zhu et al., 2024).

(III) **Local Variable Update:** Communicate with neighbors and update the dual and primal variables using decentralized techniques, such as GT:

$$(\theta_i^{k+1}, x_i^{k+1}, y_i^{k+1}) = \sum_{j=1}^{n} w_{ij} \left( \theta_j^k - \lambda_\theta^k D_{\theta,j}^k, x_j^k - \lambda_x^k D_{x,j}^k, y_j^k - \lambda_y^k D_{y,j}^k \right). \tag{7}$$

As a summary, $\hat{D}_{\diamond,i}^k$ is a stochastic approximation of descent directions $\tilde{D}_{\diamond,i}^k$, and we use $\hat{D}$ to construct the update direction $D_{\diamond,i}^k$ through decentralized techniques, $\diamond \in \{x, y, \theta\}$.

Notably, SUN-DSBO offers the flexibility to incorporate stochastic gradient estimators and decentralized techniques. In the following sections, we study two specific examples without sub-loops to maintain good communication efficiency.

---

**Algorithm 1** SUN-DSBO-SE

---

1: **Input:** penalty parameters $\{\mu_k\}$, $\{\theta_i^0, x_i^0, y_i^0\}$, learning rates $\{\lambda_\theta^k, \lambda_x^k, \lambda_y^k\}$, weight matrix $\mathbf{W}$.
2: **for** iteration $k = 0, 1, \ldots, K-1$ **do**
3:    **for** each node $i = 1, \ldots, n$ **do**
4:       Sample $\xi_{g,i}^k$ and $\xi_{f,i}^k$, and compute

$$
\begin{aligned}
\hat{D}_{\theta,i}^k &= \nabla_y g_i(x_i^k, \theta_i^k; \xi_{g,i}^k) + \frac{1}{\gamma}(\theta_i^k - y_i^k), \\
\hat{D}_{x,i}^k &= \mu_k \nabla_x f_i(x_i^k, y_i^k; \xi_{f,i}^k) + \nabla_x g_i(x_i^k, y_i^k; \xi_{g,i}^k) - \nabla_x g_i(x_i^k, \theta_i^k; \xi_{g,i}^k), \quad (8) \\
\hat{D}_{y,i}^k &= \mu_k \nabla_y f_i(x_i^k, y_i^k; \xi_{f,i}^k) + \nabla_y g_i(x_i^k, y_i^k; \xi_{g,i}^k) - \frac{1}{\gamma}(y_i^k - \theta_i^k).
\end{aligned}
$$

5:       Communicate with neighbors and update the dual and primal variables:

$$
(\theta_i^{k+1}, x_i^{k+1}, y_i^{k+1}) = \sum_{j=1}^n w_{ij} \left( \theta_j^k - \lambda_\theta^k \hat{D}_{\theta,j}^k, x_j^k - \lambda_x^k \hat{D}_{x,j}^k, y_j^k - \lambda_y^k \hat{D}_{y,j}^k \right).
$$

6:    **end for**
7: **end for**

---

**Algorithm 2** SUN-DSBO-GT

---

1: **Input:** penalty parameters $\{\mu_k\}$, $\{\theta_i^0, x_i^0, y_i^0\}$, learning rates $\{\lambda_\theta^k, \lambda_x^k, \lambda_y^k\}$, weight matrix $\mathbf{W}$,
   $\hat{D}_{\theta,i}^{-1} = 0$, $\hat{D}_{x,i}^{-1} = 0$, $\hat{D}_{y,i}^{-1} = 0$, $D_{\theta,i}^{-1} = 0$, $D_{x,i}^{-1} = 0$, $D_{y,i}^{-1} = 0$.
2: **for** iteration $k = 0, 1, \ldots, K-1$ **do**
3:    **for** each node $i = 1, \ldots, n$ **do**
4:       Sample $\xi_{g,i}^k$ and $\xi_{f,i}^k$, and compute $\hat{D}_{\theta,i}^k$, $\hat{D}_{x,i}^k$, and $\hat{D}_{y,i}^k$ using (8).
5:       Communicate with neighbors and update $D_{\theta,i}^k$, $D_{x,i}^k$, and $D_{y,i}^k$ using GT:

$$
D_{\Diamond,i}^k = \sum_{j=1}^n w_{ij} \left( D_{\Diamond,j}^{k-1} + \hat{D}_{\Diamond,j}^k - \hat{D}_{\Diamond,j}^{k-1} \right),
$$

      where $\Diamond \in \{\theta, x, y\}$ represents the variable being tracked.
6:       Communicate with neighbors and update the dual and primal variables:

$$
(\theta_i^{k+1}, x_i^{k+1}, y_i^{k+1}) = \sum_{j=1}^n w_{ij} \left( \theta_j^k - \lambda_\theta^k D_{\theta,j}^k, x_j^k - \lambda_x^k D_{x,j}^k, y_j^k - \lambda_y^k D_{y,j}^k \right).
$$

7:    **end for**
8: **end for**

---

**SUN-DSBO-SE:** A Simple and Easily implementable algorithm within the SUN-DSBO framework. It is an adaptation of decentralized stochastic gradient descent within SUN-DSBO (summarized in Algorithm 1), without the use of any advanced decentralized techniques.

**SUN-DSBO-GT:** An algorithm with Gradient Tracking to mitigate data heterogeneity. This is an adaptation of the GT technique within SUN-DSBO-SE (summarized in Algorithm 2).

**Other possible algorithmic designs.** Note that SUN-DSBO adopts the adapt-then-combine (ATC) structure in this work. In fact, it can also be implemented using alternative decentralized structures, as discussed in Sayed (2014b); Zhu et al. (2024). More details can be found in Appendix H.

## 3 THEORETICAL ANALYSIS

### 3.1 GENERAL ASSUMPTIONS

Throughout this work, we assume that the upper-level objective $F$ is bounded below by $\underline{F}$.

**Assumption 3.1** (Smoothness). For each $i \in [n]$, the objective functions $f_i$ and $g_i$ are continuously differentiable and $L_1$- and $L_2$-smooth in $x$ and $y$, respectively.

**Assumption 3.2** (Stochastic oracle). For each $i \in [n]$ and i.i.d. samples $\xi_{f,i}$ and $\xi_{g,i}$, $\nabla f_i(x, y; \xi_{f,i})$ and $\nabla g_i(x, y; \xi_{g,i})$ are unbiased estimators of $\nabla f_i(x, y)$ and $\nabla g_i(x, y)$ with bounded variances, respectively, i.e., $\mathbb{E}\|\nabla f_i(x, y) - \nabla f_i(x, y; \xi_{f,i})\|^2 \leq \delta_f^2$ and $\mathbb{E}\|\nabla g_i(x, y) - \nabla g_i(x, y; \xi_{g,i})\|^2 \leq \delta_g^2$.

**Assumption 3.3** (Network topology). The weight matrix $\mathbf{W} \in \mathbb{R}^{n \times n}$ for the communication network is non-negative, symmetric, and doubly stochastic. Its eigenvalues satisfy $1 = \lambda_1 > \lambda_2 \geq \cdots \geq \lambda_n$, with connectivity parameter $\rho := \max\{|\lambda_2|, |\lambda_n|\} < 1$.

Assumption 3.3 on the weight matrix is crucial for ensuring the convergence of decentralized algorithms and is commonly made in the literature (e.g., Chen et al. (2023; 2025); Zhu et al. (2024) for DSBO). The parameter $\rho$ measures the connectivity of the communication network: a smaller $\rho$ (closer to zero) indicates better connectivity. In the decentralized optimization literature (e.g., Lian et al. (2017); Lu & De Sa (2021); Tang et al. (2018); Lu et al. (2019)), $1 - \rho$ corresponds to the spectral gap of the weight matrix $\mathbf{W}$ and is used to ensure the decay of the consensus error.

### 3.2 THEORETICAL RESULTS

We now present the key convergence results for two specific instances of SUN-DSBO, in terms of the primal function $\Psi_\mu(x, y)$ defined in (4), using a feasible stationarity measure. A comprehensive discussion about this stationarity measure can be found in Remark D.1 in the Appendix D. Additionally, we provide the convergence analysis in terms of the consensus error, formally defined as: $\Delta^k := \frac{1}{n} \sum_{i=1}^n \mathbb{E}\left(\|x_i^k - \bar{x}^k\|^2 + \|y_i^k - \bar{y}^k\|^2 + \|\theta_i^k - \bar{\theta}^k\|^2\right)$, where $\bar{\Diamond}^k := \frac{1}{n} \sum_{i=1}^n \Diamond_i^k$ denotes the average over all agents with $\Diamond \in \{\theta, x, y\}$.

**Results for SUN-DSBO-SE.** Since SUN-DSBO-SE does not employ advanced techniques for mitigating data heterogeneity, we adopt the bounded gradient dissimilarity condition on the objective functions, which is weaker than the bounded gradient assumptions used in Chen et al. (2025); Yang et al. (2022); Lu et al. (2022); Chen et al. (2023).

**Assumption 3.4** (Gradient dissimilarity). There exist constants $\sigma_f, \sigma_g \geq 0$ such that for all $(x, y)$, $\frac{1}{n} \sum_{i=1}^n \|\nabla f_i(x, y) - \nabla F(x, y)\|^2 \leq \sigma_f^2$ and $\frac{1}{n} \sum_{i=1}^n \|\nabla g_i(x, y) - \nabla G(x, y)\|^2 \leq \sigma_g^2$.

Under the above assumptions, we establish the convergence rate of SUN-DSBO-SE.

---

**Theorem 3.5** (Convergence rate of **SUN-DSBO-SE**). *Under Assumptions 3.1–3.4, let $\mu_k = \mu_0(k + 1)^{-p}$, where $\mu_0 > 0$, $p \in (0, 1/4)$, and $\gamma \in (0, \frac{1}{2L_2})$. Choose learning rates as follows: $\lambda_\theta^k = c_\theta n^{1/2} K^{-1/2}$, $\lambda_x^k = \lambda_y^k = c_\lambda \lambda_\theta^k$, where $c_\theta, c_\lambda$ are positive constants that satisfy the conditions in Lemma F.9. Then, for SUN-DSBO-SE, we have*

$$\frac{1}{K} \sum_{k=0}^{K-1} \mathbb{E}\|\nabla \Psi_{\mu_k}(\bar{x}^k, \bar{y}^k)\|^2 = \mathcal{O}\left(\frac{\mathcal{C}_1}{n^{1/2} K^{1/2}}\right) + \mathcal{O}\left(\frac{n(\delta_f^2 + \delta_g^2)}{(1 - \rho)^2 K}\right) + \mathcal{O}\left(\frac{n(\sigma_f^2 + \sigma_g^2)}{(1 - \rho)^2 K}\right),$$

*where $\mathcal{C}_1$ is a positive constant depending on $\delta_f^2$, $\delta_g^2$, $\sigma_f^2$, and $\sigma_g^2$, but independent of $n$ and $(1 - \rho)$.*

---

*Linear speedup and consensus error.* The convergence rate $\mathcal{O}(1/(n^{1/2} K^{1/2}))$ in Theorem 3.5 demonstrates that SUN-DSBO-SE achieves asymptotic linear speedup with respect to the number $n$ of agents. The proof of Theorem 3.5 is provided in Appendix F, where we also establish an upper bound for the consensus error:

$$\frac{1}{K} \sum_{k=0}^{K-1} \Delta^k = \mathcal{O}\left(\frac{\mathcal{C}_1}{K^{1/2}}\right) + \mathcal{O}\left(\frac{n^{3/2}(\delta_f^2 + \delta_g^2)}{(1 - \rho)^2 K}\right) + \mathcal{O}\left(\frac{n^{3/2}(\sigma_f^2 + \sigma_g^2)}{(1 - \rho)^2 K}\right).$$

The proof of Theorem 3.5 is provided in Appendix F. For further theoretical results, see Appendix D.2. Using the non-asymptotic rate established in Theorem 3.5, we derive the sample and transient iteration complexities of SUN-DSBO-SE. The sample complexity refers to the total number of stochastic gradient evaluations required to achieve an $\epsilon$-stationary solution, $\mathbb{E}\|\nabla\Psi_{\mu_k}(\bar{x}^k, \bar{y}^k)\|^2 \le \epsilon$, while the transient iteration complexity refers to the number of transient iterations required for the algorithm to achieve the asymptotic linear speedup (Kong et al., 2025; Zhu et al., 2024).

**Corollary 3.6.** *In the context of Theorem 3.5, the sample complexity of SUN-DSBO-SE is $\mathcal{O}\left(1/\epsilon^2\right)$, while the transient iteration complexity is $\mathcal{O}\left(\max\left\{n^3/(1-\rho)^4, n^3(\sigma_f^2 + \sigma_g^2)^2/(1-\rho)^4\right\}\right)$.*

**Results for SUN-DSBO-GT.** Thanks to the gradient tracking technique used in SUN-DSBO-GT, we present its convergence result, demonstrating that it achieves linear speedup without relying on any assumptions of gradient heterogeneity.

**Theorem 3.7.** *Under Assumptions 3.1–3.3, let $\mu_0 > 0$, $p \in (0, 1/4)$, $\mu_k = \mu_0(k+1)^{-p}$, and $\gamma \in (0, \frac{1}{2L_2})$. Choose $\lambda_\theta^k = c_\theta n^{1/2} K^{-1/2}$, $\lambda_x^k = \lambda_y^k = c_\lambda \lambda_\theta^k$, where $c_\theta, c_\lambda$ are positive constants that satisfy the conditions in Lemma G.6. Then, for SUN-DSBO-GT, we have*

$$\frac{1}{K}\sum_{k=0}^{K-1}\mathbb{E}\|\nabla\Psi_{\mu_k}(\bar{x}^k, \bar{y}^k)\|^2 = \mathcal{O}\left(\frac{\mathcal{C}_2}{n^{1/2}K^{1/2}}\right) + \mathcal{O}\left(\frac{n(\delta_f^2 + \delta_g^2)}{(1-\rho)^4 K}\right);$$

$$\frac{1}{K}\sum_{k=0}^{K-1}\Delta^k = \mathcal{O}\left(\frac{\mathcal{C}_2}{K^{1/2}}\right) + \mathcal{O}\left(\frac{n^{3/2}(\delta_f^2 + \delta_g^2)}{(1-\rho)^4 K}\right),$$

*where $\mathcal{C}_2$ is a positive constant depending on $\delta_f^2$, and $\delta_g^2$, but independent of $n$ and $(1-\rho)$.*

**Corollary 3.8.** *Under the conditions of Theorem 3.7, the sample complexity of SUN-DSBO-GT is $\mathcal{O}\left(1/\epsilon^2\right)$, while the transient iteration complexity is $\mathcal{O}\left(n^3/(1-\rho)^8\right)$.*

Although SUN-DSBO-GT does not make any assumptions about gradient heterogeneity, it requires slightly more communication, memory, and computational costs compared to SUN-DSBO-SE. These increased costs may impact practical performance, which will be evaluated in the next section.

## 4 EXPERIMENTS

In this section, we conduct experiments to evaluate the effectiveness of the proposed algorithms on real-world machine learning tasks. We compare the proposed SUN-DSBO-SE and SUN-DSBO-GT with the AID-based algorithms SLDBO (Dong et al., 2023), D-SOBA (Kong et al., 2025), SPARKLE-EXTRA/GT (Zhu et al., 2024), and the value function-based algorithm DSGDA-GT (Wang et al., 2024). These experiments are conducted across multiple agents connected by diverse communication networks. We use the Dirichlet distribution to create heterogeneous training and validation data for the agents, following the methodology of Lin et al. (2021); Zhu et al. (2024). The degree of heterogeneity is controlled by the parameter $\mathcal{H} > 0$, where smaller values of $\mathcal{H}$ correspond to more severe non-i.i.d. partitions. Each experiment is repeated 10 times to improve statistical reliability, and we report the mean results with standard deviations depicted as shaded areas. Further experimental details are provided in Appendix C.

**Data hyper-cleaning.** We consider a data hyper-cleaning problem in a decentralized setting (Zhu et al., 2024), with the problem formulation and experimental setup detailed in Appendix C.2. We conduct experiments on Fashion-MNIST dataset (Xiao et al., 2017) across different corruption rates (`cr`) and various values of the heterogeneity parameter $\mathcal{H}$.

In Figure 2, we evaluate all algorithms across different values of the heterogeneity parameter $\mathcal{H} \in \{0.1, 0.5, 1\}$, using a corruption rate of `cr = 0.3`. It can be observed that SUN-DSBO-GT consistently outperforms the other algorithms, with SUN-DSBO-SE achieving the second-best performance. It should be noted that Figure 2(a) demonstrates the effectiveness of the GT and

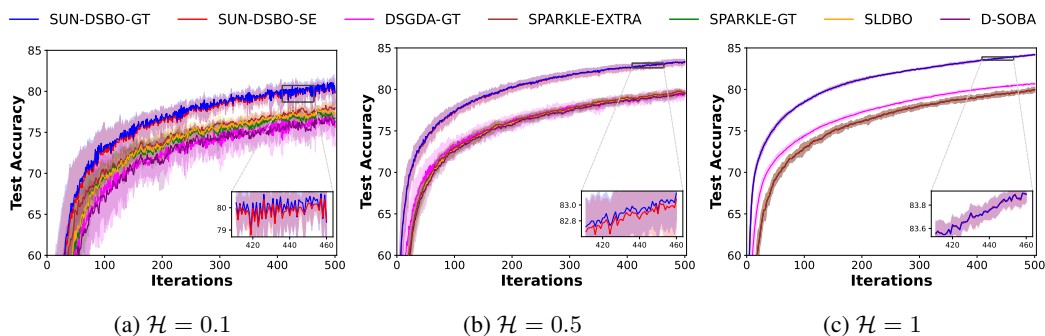

(a) $\mathcal{H} = 0.1$      (b) $\mathcal{H} = 0.5$      (c) $\mathcal{H} = 1$

Figure 2: Comparison of algorithms for data hyper-cleaning under different values of the heterogeneity parameter $\mathcal{H}$ with a corruption rate of $\mathtt{cr} = 0.3$.

EXTRA techniques in mitigating data heterogeneity, as evidenced by the comparisons between SUN-DSBO-SE and SUN-DSBO-GT, as well as D-SOBA and SPARKLE-EXTRA/GT.

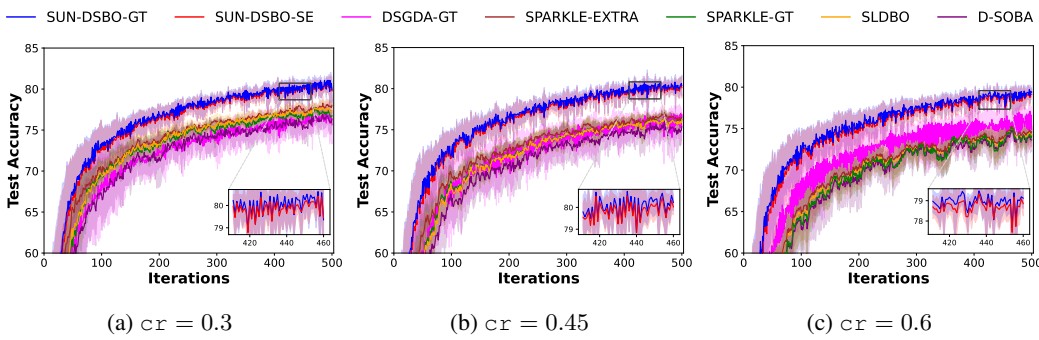

(a) $\mathtt{cr} = 0.3$      (b) $\mathtt{cr} = 0.45$      (c) $\mathtt{cr} = 0.6$

Figure 3: Comparison of algorithms for data hyper-cleaning across different corruption rates ($\mathtt{cr}$) with a heterogeneity parameter of $\mathcal{H} = 0.1$.

In Figure 3, we further compare the algorithms in a data-heterogeneous scenario while varying the corruption rates $\mathtt{cr} \in \{0.3, 0.45, 0.6\}$. It is observed that SUN-DSBO-GT and SUN-DSBO-SE consistently achieve higher accuracy and faster convergence. Moreover, Figure 3(a)-(c) show that the performance gap between SUN-DSBO-SE and SUN-DSBO-GT remains small across different $\mathtt{cr}$ values, with SUN-DSBO-GT demonstrating greater stability. Appendix B.1 provides a comparison of these algorithms in terms of runtime and presents results comparing them against single-level algorithms to demonstrate the benefits of bilevel optimization modeling and algorithms.

**Hyper-representation.** We further demonstrate the effectiveness of SUN-DSBO on a hyper-representation task (Franceschi et al., 2018; Tarzanagh et al., 2022) within a decentralized setting. This task involves a meta-learning problem where a representation and a header are jointly learned on training and validation datasets distributed across multiple agents. The problem formulation and experimental setup are provided in Appendix C.3. We conduct experiments using a 2-layer MLP on MNIST (LeCun et al., 1998) and a 7-layer CNN on CIFAR-10 (Krizhevsky, 2009), respectively.

We first evaluate all algorithms on MNIST benchmark using a two-layer MLP, as shown in Figure 4(a)-(c). When data heterogeneity is severe (i.e., smaller $\mathcal{H}$), both SUN-DSBO-SE and SUN-DSBO-GT exhibit stronger oscillations compared to other methods. Nonetheless, SUN-DSBO-GT consistently achieves the fastest convergence and highest accuracy across all values of $\mathcal{H}$. We further compare the algorithms on the more challenging CIFAR-10 dataset using a 7-layer CNN. The results in Figure 4(d)–(f) illustrate that SUN-DSBO-SE and SUN-DSBO-GT consistently outperform the other algorithms across different values of the heterogeneity parameter. Surprisingly, SUN-DSBO-SE achieves better accuracy and faster convergence than SUN-DSBO-GT.

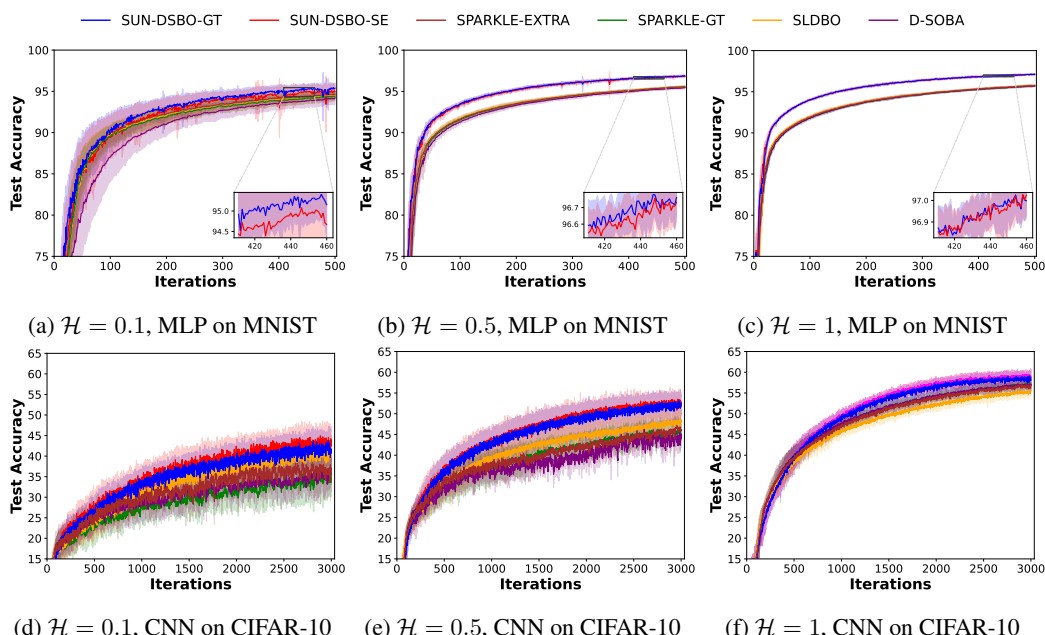

(a) $\mathcal{H} = 0.1$, MLP on MNIST  (b) $\mathcal{H} = 0.5$, MLP on MNIST  (c) $\mathcal{H} = 1$, MLP on MNIST

(d) $\mathcal{H} = 0.1$, CNN on CIFAR-10  (e) $\mathcal{H} = 0.5$, CNN on CIFAR-10  (f) $\mathcal{H} = 1$, CNN on CIFAR-10

Figure 4: Comparison of algorithms for hyper-representation under varying values of the heterogeneity parameter $\mathcal{H}$.

**Ablation study.** We compare the proposed algorithms on the hyper-cleaning task using Fashion-MNIST with different communication networks, characterized by the connectivity parameter $\rho$, where larger values indicate weaker connectivity. As $\rho$ increases in Figure 5, performance of both SUN-DSBO-SE and SUN-DSBO-GT decreases, consistent with Theorems 3.5 and 3.7. Notably, SUN-DSBO-GT demonstrates better robustness across varying weight matrices, highlighting its ability to mitigate the adverse effects of limited connectivity. Additional ablation studies are in Appendix B.3.

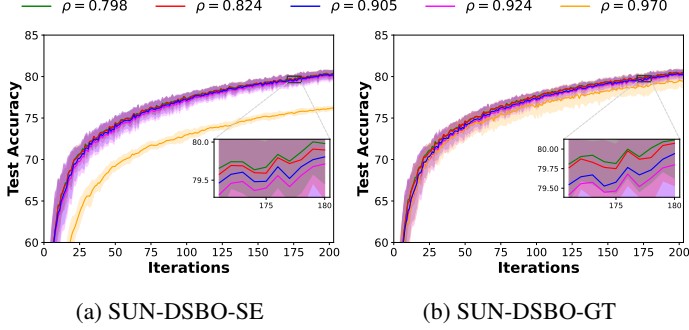

(a) SUN-DSBO-SE  (b) SUN-DSBO-GT

Figure 5: Data hyper-cleaning is performed with varying values of $\rho$, which quantifies the connectivity of the communication network. A smaller $\rho$ indicates stronger connectivity.

## 5 CONCLUSION

This paper introduces SUN-DSBO, a flexible framework for nonconvex DSBO that supports various decentralized strategies and requires only first-order stochastic gradient oracles. Convergence analysis and experimental results for two specific instances of SUN-DSBO demonstrate its effectiveness. While these results highlight the potential of the proposed framework, several issues remain unexplored, which may represent promising directions for future research.

*Unified and Improved Convergence Analysis.* The current analysis is limited to the gradient tracking (GT) technique and does not explore the potential advantages of mixed strategies. Thus, developing a more unified convergence analysis for additional variants of SUN-DSBO, similar to the approach used in Zhu et al. (2024) for the SPARKLE algorithm, represents an interesting direction for future work. Moreover, since the lower-level objective in DSBO may be nonconvex, this prevents directly applying the improved convergence analysis from single-level decentralized stochastic algorithms (e.g., (Koloskova et al., 2021)) to SUN-DSBO. It would also be interesting in future work to extend the rich theoretical results from single-level settings to SUN-DSBO, potentially improving its theoretical guarantees.

## REPRODUCIBILITY STATEMENT

To ensure reproducibility of our work, we provide all necessary resources. The source code for the SUN-DSBO algorithms is available at the following link: `https://anonymous.4open.science/r/SUN-DSBO-1616`. The datasets used are publicly available, and the data preprocessing steps are described in Section 4 in the main text and Appendix C. Detailed experimental configurations, hyperparameters, and training procedures are also included. The proof sketch and detailed proofs for SUN-DSBO-SE and SUN-DSBO-GT are provided in Appendices E, F, and G.

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

# APPENDIX

- **Expanded related work in Section A**.

- **Additional experiments in Section B**.

  - **Data hyper-cleaning in Subsection B.1**.

  - **Hyper-representation in Subsection B.2**.

  - **Ablation study in Subsection B.3**.

  - **Alternative tabular presentation in Subsection B.4.**

- **Details of experiments in Section C**.

  - **Hyperparameter tuning strategy in Subsection C.1**.

  - **Data hyper-cleaning in Subsection C.2**.

  - **Hyper-representation in Subsection C.3**.

  - **Ablation study in Subsection C.4**.

- **Additional theoretical results in Section D**.

  - **The relationship between problem (1) and problem (4) in Subsection D.1**.

  - **Additional convergence rate results in Subsection D.2**.

- **Proof sketch in Section E**.

  - **Main challenges in Subsection E.1.**

  - **Key steps in Subsection E.2.**

- **Proofs of SUN-DSBO-SE in Section F**.

  - **Notations in Subsection F.1**.

  - **Preliminary lemmas in Subsection F.2**.

  - **Convergence analysis in Subsection F.3**.

- **Proofs of SUN-DSBO-GT in Section G**.

  - **Notations in Subsection G.1**.

  - **Preliminary lemmas in Subsection G.2**.

  - **Convergence analysis in Subsection G.3**.

- **More details of SUN-DSBO in Section H**.

## A    EXPANDED RELATED WORK

### A.1    DECENTRALIZED (STOCHASTIC) OPTIMIZATION

The study of decentralized optimization has its roots in early research on parallel and distributed computation (Tsitsiklis, 1984; Bertsekas & Tsitsiklis, 2015), which primarily focused on simple averaging-based algorithms without accounting for data or system heterogeneity. With the advancement of large-scale and distributed learning, a variety of methods have been developed to address decentralized consensus optimization problems (Nedic & Ozdaglar, 2009; Nedić & Olshevsky, 2014; Sayed, 2014a; Lian et al., 2017). However, these approaches generally lack explicit mechanisms to handle heterogeneous data.

To address the challenges posed by heterogeneous data distributions, a range of algorithms have been developed that incorporate explicit correction mechanisms. Methods such as EXTRA (Shi et al., 2015), Exact Diffusion (ED) (Yuan et al., 2018), and $D^2$ (Tang et al., 2018) utilize correction techniques based on two consecutive iterates to estimate the update direction. These algorithms, along with their variants and alternative interpretations (Mokhtari & Ribeiro, 2016; Mokhtari et al., 2016; Yuan et al., 2018; Li et al., 2019; Yuan et al., 2020), are designed to reduce steady-state error and enhance convergence behavior under non-i.i.d. data conditions. Concurrently, gradient tracking (GT)-based methods (Nedic et al., 2017; Nedić et al., 2017; Pu & Nedić, 2021; Song et al., 2024) have been proposed to mitigate the discrepancy between local gradients and the global optimization direction. These approaches maintain auxiliary variables to track the global gradient and can be integrated into decentralized updates to improve stability and accuracy in heterogeneous settings.

Recent research has also incorporated practical system considerations, including communication efficiency (Liu et al., 2025), robustness to inexact or noisy updates (Shah & Bollapragada, 2025), and privacy preservation in decentralized learning (Cheng et al., 2024). These developments have substantially expanded the applicability of decentralized stochastic optimization in real-world machine learning systems.

### A.2    DECENTRALIZED (STOCHASTIC) BILEVEL OPTIMIZATION

A comparison between representative decentralized stochastic bilevel optimization (DSBO) methods and our approach is provided in Table 1. Most existing DSBO algorithms assume that the lower-level problem is strongly convex, which ensures the uniqueness of the solution and facilitates the computation of the hypergradient (Kearns, 1990). A key challenge in this setting is computing or approximating the inverse Hessian matrix associated with the global lower-level objective.

Several methods address this challenge by introducing surrogate techniques to approximate the inverse Hessian. For example, Yang et al. (2022) employ a Neumann series expansion, while Chen et al. (2023) and Chen et al. (2025) introduce the JHIP and HIGP oracles, respectively. Methods such as SLAM and DAGM enforce solution consensus among agents by penalizing deviations from a shared variable. Most of these approaches adopt a double-loop structure, which can increase both computational burden and communication overhead. To overcome these limitations, recent works have proposed single-loop alternatives aimed at improving computational and communication efficiency. For instance, Dong et al. (2023), Kong et al. (2025), and Zhu et al. (2024) adopt the SOBA framework (Dagréou et al., 2022), where the inverse-Hessian-related linear system is reformulated as a quadratic optimization problem. Although these methods eliminate the inner loop, they still depend on second-order information. As noted in Dagréou et al. (2024), computing Hessian-vector products can be significantly more expensive—in both time and memory—than gradient computations, particularly in deep learning applications. To eliminate second-order computations entirely, Wang et al. (2024) propose a fully first-order DSBO method. However, their approach still assumes strong convexity of the lower-level objective to ensure the well-posedness of the hypergradient.

In a different direction, Qin et al. (2025) propose an algorithm that avoids both strong convexity assumptions and the use of second-order information. Their framework considers personalized bilevel optimization, in which each agent maintains an independent lower-level objective. Related work by Qiu et al. (2023) also investigates this setting, but under the assumption of strong convexity. A

representative formulation of personalized DSBO is given by

$$\min_{x_i \in \mathbb{R}^{d_x}, \, y_i \in \mathcal{S}^*(x_i)} \frac{1}{n} \sum_{i=1}^{n} f_i(x_i, y_i) \ \text{ s.t. } y_i \in \mathcal{S}^*(x_i) := \arg\min_{y_i \in \mathbb{R}^{d_y}} g_i(x_i, y_i), \ x_i = x_{i'}, \text{ if } w_{ii'} \neq 0,$$

where $f_i$ and $g_i$ denote the upper- and lower-level objectives of agent $i$, and $W = [w_{ii'}]$ is the communication weight matrix.

Table 1: Comparison of the proposed algorithms with closely related DSBO methods. "Convexity" refers to the convexity of the lower-level objective: S.C. (strongly convex), N.C. (non-convex). "Heterogeneity" refers to the gradient heterogeneity assumption for the upper/lower-level objectives: BG (bounded gradient), BG* (bounded gradient for all $(x, y^*(x))$), BGD (bounded gradient dissimilarity), and Free (no assumption required). Free$^-$ indicates the reliance on certain heterogeneity conditions, but distinct from the BG, BG*, and BGD assumptions. "1-order" and "2-order" represent the Lipschitz continuity of the first- and second-order derivatives of the objectives. We use "Grad", "HVP", and "JVP" to denote oracles of gradients, Hessian-vector products, Jacobian-vector products, respectively. Note that here we only compare the corresponding *conditions*, not the convergence rates and complexity, as we use different stationarity measures, making direct comparison not feasible.

| Method | Convexity | Heterogeneity | Smoothness | Oracles |
|---|---|---|---|---|
| DSBO (Chen et al., 2025) | S.C. | BG/Free$^-$ | 1,2-order | Grad, HVP, JVP |
| Gossip DSBO (Yang et al., 2022) | S.C. | BG/Free$^-$ | 1,2-order | Grad, HVP, JVP |
| SLAM (Lu et al., 2022) | S.C. | BG/Free$^-$ | 1,2-order | Grad, HVP, JVP |
| MA-DSBO (Chen et al., 2023) | S.C. | BG/BGD | 1,2-order | Grad, HVP, JVP |
| VRDBO (Gao et al., 2023a) | S.C. | BG/Free | 1,2-order | Grad, HVP, JVP |
| LDP-DSBO (Chen & Wang, 2024) | S.C. | BG/Free | 1,2-order | Grad, HVP, JVP |
| D-SOBA (Kong et al., 2025) | S.C. | BG*/BGD | 1,2-order | Grad, HVP, JVP |
| SPARKLE (Zhu et al., 2024) | S.C. | BG*/Free | 1,2-order | Grad, HVP, JVP |
| DSGDA-GT (Wang et al., 2024) | S.C. | BG/Free | 1,2-order | Grad |
| **SUN-DSBO-SE (ours)** | N.C. | BGD/BGD | 1-order | Grad |
| **SUN-DSBO-GT (ours)** | N.C. | Free/Free | 1-order | Grad |

**Note:** For methods using Free$^-$ assumption:

- Chen et al. (2025): Assumes data on each local device are i.i.d.
- Yang et al. (2022): Assumes $\nabla_y g_i(x, y; \xi_{g,i})$ has bounded second-order moments, i.e., $\mathbb{E}[\|\nabla_y g_i(x, y; \xi_{g,i})\|^2] \leq C_g$ for some constant $C_g$.
- Lu et al. (2022): Assumes there exists $L_g$ such that $\left\| \nabla_{22}^2 f_i(x_i, y_i) - \frac{1}{m} \sum_{i=1}^{m} \nabla_{22}^2 f_i(x_i, y_i') \right\| \leq L\|y_i - y_i'\|, \forall x_i, y_i, y_i'$.

### A.3 Moreau envelope-based methods

In the single-agent setting, Moreau envelope-based reformulations have been extensively explored to address nonconvex lower-level problems. For example, Gao et al. (2023b) proposed a double-loop algorithm based on a weakly convex reformulation using the Moreau envelope, while Liu et al. (2024) developed a single-loop gradient-based method tailored to unconstrained problems. Extensions to constrained lower-level problems have been investigated in Yao et al. (2024; 2025), where proximal Lagrangian value function formulations are introduced. However, these approaches are limited to deterministic and centralized optimization settings.

## B Additional experiments

### B.1 Data hyper-cleaning

In Figure 6, we compare the convergence accuracy of each algorithm in terms of runtime under different heterogeneity levels $\mathcal{H}$. We observe that SUN-DSBO-GT consistently achieves the fastest convergence, followed by SUN-DSBO-SE across all heterogeneity settings. Comparing Figures 2

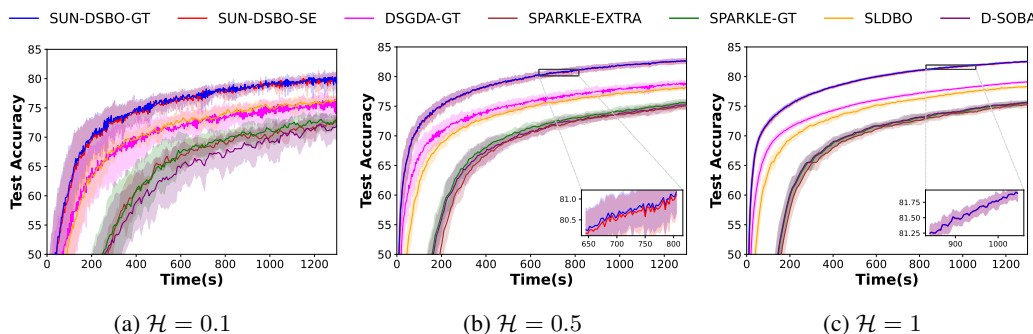

(a) $\mathcal{H} = 0.1$        (b) $\mathcal{H} = 0.5$        (c) $\mathcal{H} = 1$

Figure 6: Comparison of algorithms over runtime for data hyper-cleaning under different values of the heterogeneity parameter $\mathcal{H}$.

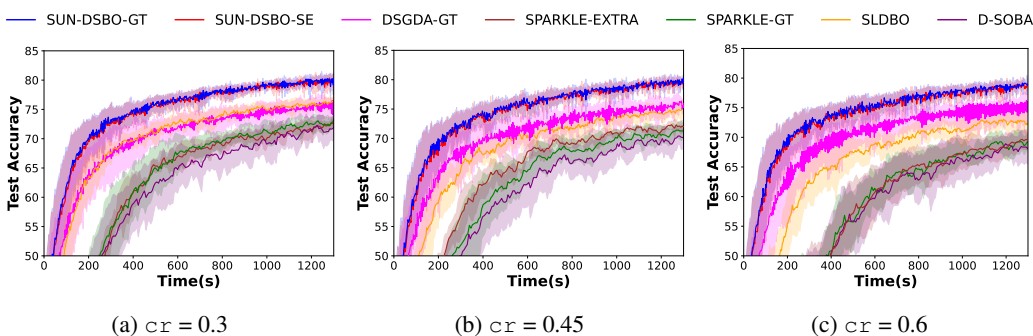

(a) cr = 0.3        (b) cr = 0.45        (c) cr = 0.6

Figure 7: Comparison of algorithms over runtime for data hyper-cleaning under different corruption rate of cr.

and 6, we note that on the Fashion-MNIST dataset, fully first-order algorithms exhibit lower runtime costs than AID-based approaches.

Figure 7 further evaluates the performance over runtime under varying corruption rates cr. SUN-DSBO-GT maintains the best efficiency, followed by SUN-DSBO-SE, both outperforming the baselines. Moreover, as shown in Figures 7(a), (b), and (c), higher levels of data heterogeneity consistently lead to weaker performance over runtime, highlighting the challenge of handling both corruption and heterogeneity simultaneously.

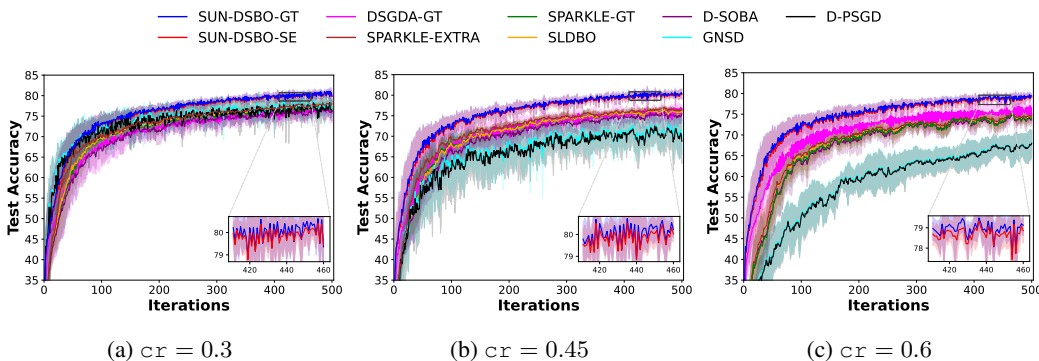

(a) cr = 0.3        (b) cr = 0.45        (c) cr = 0.6

Figure 8: Performance of data hyper-cleaning with single-level algorithms under corruption rates cr = 0.3, 0.45, and 0.6.

In Figure 8, we additionally compare with two single-level algorithms, D-PSGD (Lian et al., 2017) and GNSD (Lu et al., 2019), under different corruption rates (cr) on the Fashion-MNIST dataset.

The training set here is formed by merging the corrupted training data with the validation set. From Figure 8(a), we observe that when the label corruption rate is relatively low, both D-PSGD and GNSD maintain reasonable performance, although still inferior to SUN-DSBO-SE and SUN-DSBO-GT. However, as the corruption rate increases in Figures 8(b)–(c), the performance of D-PSGD and GNSD weakens, highlighting the necessity of employing bilevel optimization to enhance robustness under severe label noise.

## B.2 HYPER-REPRESENTATION

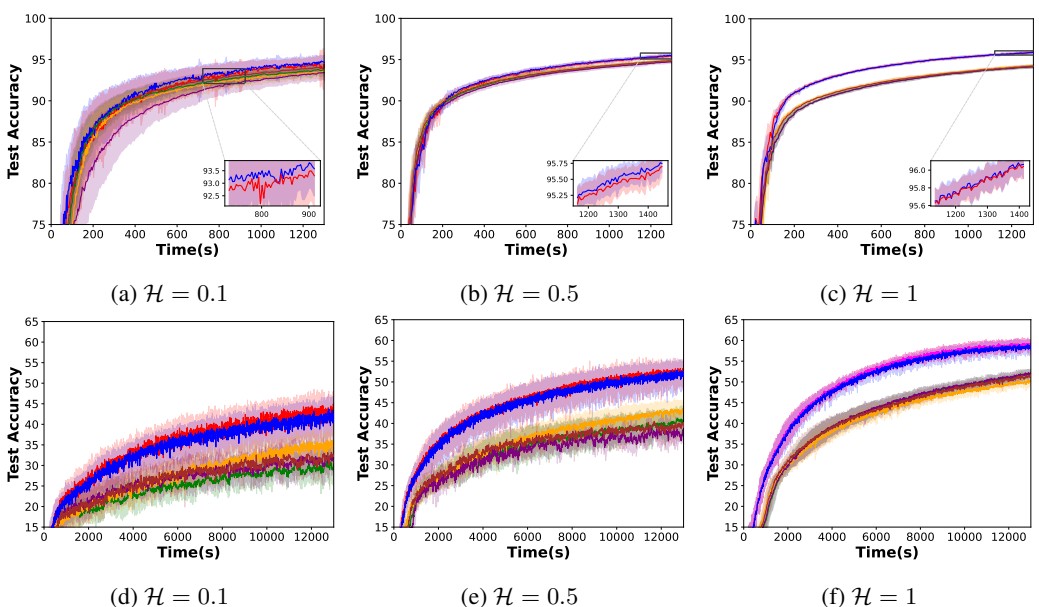

(a) $\mathcal{H} = 0.1$        (b) $\mathcal{H} = 0.5$        (c) $\mathcal{H} = 1$

(d) $\mathcal{H} = 0.1$        (e) $\mathcal{H} = 0.5$        (f) $\mathcal{H} = 1$

Figure 9: Comparison of algorithms over runtime for hyper-representation with different heterogeneity parameter $\mathcal{H}$.

In Figure 9, we compare the performance over runtime of all algorithms in terms of the accuracy achieved on the hyper-representation learning task, using a 2-layer MLP on MNIST and a 7-layer CNN on CIFAR-10, as described in Section 4.

First, Figures 9(a)–(c) present the results on MNIST using a 2-layer MLP. Across all levels of heterogeneity $\mathcal{H}$, SUN-DSBO-GT demonstrates the most efficient convergence in terms of runtime. Interestingly, a comparison between Figures 4(a)–(c) and 9(a)–(c) reveals that, on the MNIST dataset, fully first-order algorithms do not exhibit lower runtime costs than AID-based approaches.

Second, Figures 9(d)–(f) present the results on CIFAR-10 using a 7-layer CNN. Once again, SUN-DSBO-GT demonstrates the best performance over runtime across all heterogeneity levels. By comparing Figures 4(d)–(f) and 9(d)–(f), we observe that in this larger and more complex task, fully first-order algorithms consistently incur lower runtime costs than AID-based counterparts, which aligns with the expected computational overheads associated with deeper neural architectures.

## B.3 ABLATION STUDIES

**Comparison of SUN-DSBO-SE and SUN-DSBO-GT.** We compare our proposed SUN-DSBO-SE and SUN-DSBO-GT algorithms on the task of hyperparameter optimization for logistic regression, following the experimental setting in Chen et al. (2023); Wang et al. (2024). This task aims to tune the coefficient of a quadratic regularization term in logistic regression models across multiple datasets. Additional implementation and data generation details are provided in Appendix C.4.

First, we consider a bilevel hyperparameter optimization problem for binary classification on synthetic data. The synthetic dataset is generated following the procedure described in Chen et al. (2023); Wang et al. (2024), originally proposed in the single-agent setting (Pedregosa, 2016; Grazzi et al., 2020).

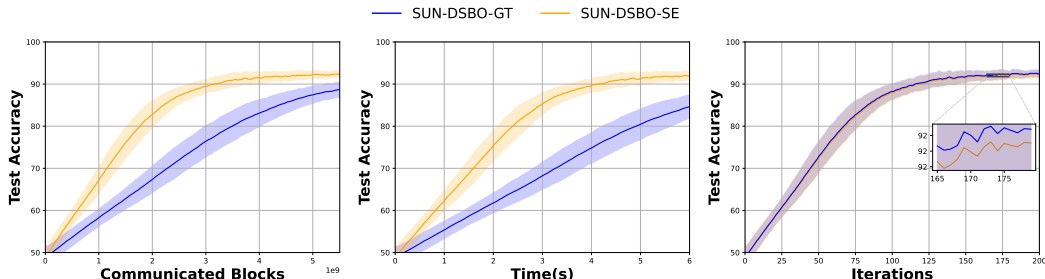

Figure 10: Test accuracy of $\ell_2$-regularized logistic regression on synthetic data with ring graph.

Additional details regarding the experimental setup are provided in Appendix C.4. From Figure 10, we make the following observations. While SUN-DSBO-GT exhibits slightly faster convergence than SUN-DSBO-SE in terms of iteration count in Figure 10(c), SUN-DSBO-SE consistently outperforms in terms of both runtime and communication cost (measured by communication blocks[1]), as shown in Figures 10(a)–(b).

Second, compared to the synthetic task, we consider a more realistic and challenging scenario: hyperparameter optimization for a linear classifier on the MNIST dataset, following the setup in Chen et al. (2025); Wang et al. (2024). The problem formulation and implementation details are provided in Appendix C.4. As shown in Figure 11, SUN-DSBO-SE consistently achieves superior performance in terms of runtime and communication efficiency (Figures 11(a)–(b)). Although SUN-DSBO-GT converges slightly faster in terms of iteration count (Figure 11(c)), it incurs higher communication and computational overheads.

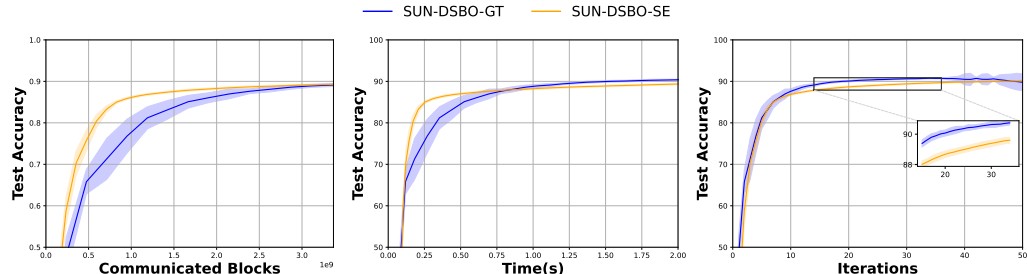

Figure 11: Test accuracy of $l_2$-regularized logistic regression on MNIST with ring graph.

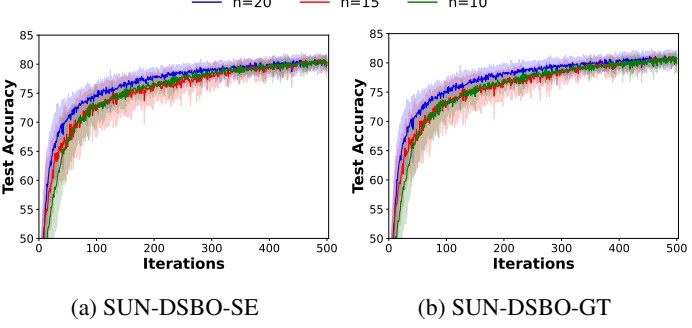

(a) SUN-DSBO-SE        (b) SUN-DSBO-GT

Figure 12: Data hyper-cleaning with different number of agents.

---

[1]A "communication block" refers to a unit used in the `tracemalloc` module in Python (see `https://docs.python.org/3/library/tracemalloc.html`) to track memory usage. Following Chen et al. (2023), we use the number of communicated blocks between agents as a proxy for communication overhead.

**Results with different number of agents.** We evaluate the proposed algorithms on the data hyper-cleaning task using Fashion-MNIST under different numbers of agents, with $n \in \{10, 15, 20\}$. As shown in Figure 12, both Sun-DSBO-SE and Sun-DSBO-GT exhibit improved performance as the number of agents $n$ increases, consistent with the trends suggested by Theorems 3.5 and 3.7.

Table 2: Performance of SUN-DSBO-SE and SUN-DSBO-GT on the hyper-representation task using MNIST under different values of $p$.

| $p$ / **Iter** | 100 | 200 | 300 | 400 | 500 |
|---|---|---|---|---|---|
| | **SUN-DSBO-SE** | | | | |
| 0 | $93.42 \pm 0.23$ | $\mathbf{95.19 \pm 0.18}$ | $\mathbf{96.00 \pm 0.19}$ | $96.51 \pm 0.17$ | $96.81 \pm 0.09$ |
| 0.0001 | $\mathbf{93.43 \pm 0.25}$ | $95.19 \pm 0.19$ | $95.95 \pm 0.20$ | $96.52 \pm 0.17$ | $\mathbf{96.82 \pm 0.11}$ |
| 0.001 | $93.42 \pm 0.26$ | $95.19 \pm 0.21$ | $\mathbf{96.00 \pm 0.23}$ | $96.54 \pm 0.14$ | $96.81 \pm 0.12$ |
| 0.01 | $93.38 \pm 0.26$ | $95.11 \pm 0.19$ | $95.92 \pm 0.22$ | $\mathbf{96.55 \pm 0.14}$ | $96.75 \pm 0.18$ |
| 0.1 | $92.95 \pm 0.26$ | $94.67 \pm 0.22$ | $95.52 \pm 0.21$ | $96.13 \pm 0.16$ | $96.43 \pm 0.12$ |
| | **SUN-DSBO-GT** | | | | |
| 0 | $\mathbf{93.53 \pm 0.22}$ | $95.23 \pm 0.19$ | $\mathbf{96.06 \pm 0.17}$ | $96.58 \pm 0.13$ | $96.86 \pm 0.12$ |
| 0.0001 | $93.51 \pm 0.23$ | $\mathbf{95.25 \pm 0.18}$ | $96.06 \pm 0.16$ | $96.58 \pm 0.14$ | $\mathbf{96.87 \pm 0.13}$ |
| 0.001 | $\mathbf{93.53 \pm 0.23}$ | $95.24 \pm 0.18$ | $96.05 \pm 0.19$ | $\mathbf{96.59 \pm 0.12}$ | $96.87 \pm 0.13$ |
| 0.01 | $93.48 \pm 0.24$ | $95.20 \pm 0.17$ | $96.00 \pm 0.21$ | $96.55 \pm 0.13$ | $96.85 \pm 0.14$ |
| 0.1 | $93.06 \pm 0.22$ | $94.77 \pm 0.17$ | $95.58 \pm 0.19$ | $96.21 \pm 0.13$ | $96.51 \pm 0.15$ |

**Results with different $p$ in Theorems 3.5 and 3.7.** We evaluate the proposed algorithms on the hyper-representation task using MNIST under different values of $p$, with $p \in 0, 0.0001, 0.001, 0.01, 0.1$. As shown in Table 2, both SUN-DSBO-SE and SUN-DSBO-GT exhibit relatively stable performance when $p = 0, 0.0001$, and $0.001$. However, when comparing $p = 0.001, 0.01$, and $0.1$, we observe that the performance of both algorithms tends to deteriorate as $p$ increases.

Table 3: Performance of SUN-DSBO-SE and SUN-DSBO-GT on MNIST hyper-representation task under different $\gamma$.

| $\gamma$ / **Iter** | 100 | 200 | 300 | 400 | 500 |
|---|---|---|---|---|---|
| | **SUN-DSBO-SE** | | | | |
| 50 | $93.42 \pm 0.26$ | $95.19 \pm 0.21$ | $96.00 \pm 0.23$ | $\mathbf{96.54 \pm 0.14}$ | $\mathbf{96.81 \pm 0.12}$ |
| 5 | $\mathbf{94.22 \pm 0.21}$ | $\mathbf{95.87 \pm 0.21}$ | $\mathbf{96.24 \pm 0.30}$ | $95.82 \pm 1.70$ | $96.57 \pm 0.26$ |
| 0.5 | $93.64 \pm 0.23$ | $95.39 \pm 0.23$ | $95.13 \pm 1.59$ | $94.43 \pm 1.89$ | $95.70 \pm 0.78$ |
| 0.05 | $92.85 \pm 0.29$ | $95.01 \pm 0.24$ | $94.50 \pm 2.04$ | $95.83 \pm 0.63$ | $95.59 \pm 0.77$ |
| 0.005 | $92.43 \pm 0.76$ | $93.79 \pm 0.42$ | $93.44 \pm 1.48$ | $93.34 \pm 1.65$ | $94.45 \pm 0.30$ |
| | **SUN-DSBO-GT** | | | | |
| 50 | $93.53 \pm 0.23$ | $95.24 \pm 0.18$ | $96.05 \pm 0.19$ | $96.59 \pm 0.12$ | $\mathbf{96.87 \pm 0.13}$ |
| 5 | $\mathbf{94.34 \pm 0.35}$ | $\mathbf{95.96 \pm 0.17}$ | $\mathbf{96.53 \pm 0.18}$ | $\mathbf{96.73 \pm 0.27}$ | $96.61 \pm 0.34$ |
| 0.5 | $93.73 \pm 0.21$ | $95.47 \pm 0.29$ | $95.82 \pm 0.64$ | $95.95 \pm 1.59$ | $95.75 \pm 0.52$ |
| 0.05 | $92.92 \pm 0.27$ | $95.07 \pm 0.21$ | $95.80 \pm 0.22$ | $95.87 \pm 0.65$ | $95.53 \pm 1.39$ |
| 0.005 | $92.78 \pm 0.53$ | $93.74 \pm 0.66$ | $94.37 \pm 0.40$ | $94.52 \pm 0.35$ | $94.43 \pm 1.47$ |

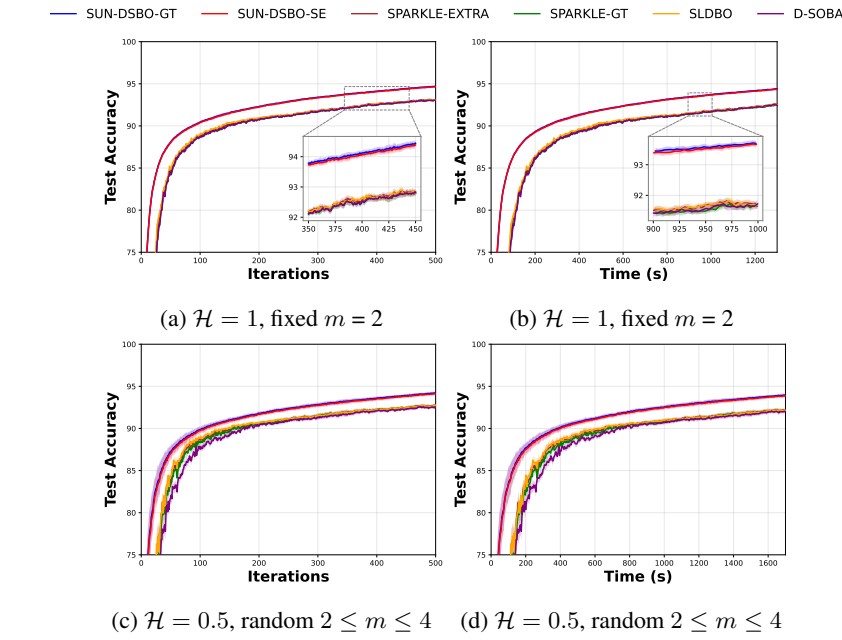

(a) $\mathcal{H} = 1$, fixed $m = 2$    (b) $\mathcal{H} = 1$, fixed $m = 2$

(c) $\mathcal{H} = 0.5$, random $2 \leq m \leq 4$    (d) $\mathcal{H} = 0.5$, random $2 \leq m \leq 4$

Figure 13: Comparison of algorithms over runtime for data hyper-representation under different dynamic topologies with varying heterogeneity. DSGDA-GT is not included due to a sudden decline in performance later, which might be due to the selection of inappropriate hyperparameters.

**Results with different $\gamma$ in (5).** We evaluate the proposed algorithms on the hyper-representation task using MNIST under different values of $\gamma$, with $\gamma \in \{50, 5, 0.5, 0.05, 0.005\}$. As shown in Table 3, both SUN-DSBO-SE and SUN-DSBO-GT achieve comparable accuracies across these choices, indicating that the performance is largely insensitive to $\gamma$. This observation is consistent with our theoretical claim in Section 2.1, item (i), that problem (5) is independent of $\gamma$. However, we also note that when $\gamma$ becomes smaller, the accuracy tends to degrade and the results become more oscillatory. We suspect this behavior may be due to the limited range of step sizes and other parameters used in the experiments.

**Results with dynamic topology.** We evaluate the proposed algorithms on the hyper-representation task using MNIST under a dynamic topology. The topology construction process involves first creating a connected symmetric adjacency matrix by randomly selecting $m$ neighbors, followed by generating a doubly stochastic matrix using Metropolis-Hastings weights. The value of $m$ can be set in two ways: fixed or random. In our experiments, as shown in Figure 13(a), we use a static value of $m = 2$, while in Figures 13(c) and (d), $m$ is chosen randomly between 2 and 4.

We observe the following:

From Figure 13(a), we can see that (i) SUN-DSBO-GT performs the best, followed by SUN-DSBO-SE in the dynamic topology environment with a fixed $m$. This suggests that the inclusion of gradient tracking in SUN-DSBO-GT helps improve performance by better adapting to the dynamic topology. (ii) From Figure 13(b), we observe that SUN-DSBO-SE/GT exhibits better robustness in more complex environments, especially when the heterogeneity parameter $\mathcal{H}$ is lower (i.e., $\mathcal{H} = 0.5$), and when the topology is dynamic with random values of $m$ (ranging from 2 to 4).

**Results with SUN-DSBO extended Algorithms.** In Figure 14, we further compare the algorithms generated by the SUN-DSBO framework with a heterogeneity parameter of $\mathcal{H} = 0.1$, while varying the corruption rates `cr` $\in \{0.3, 0.45, 0.6\}$. Specifically, we introduce SUN-DSBO-ED, which incorporates ED, SUN-DSBO-EXTRA, which incorporates EXTRA, and the hybrid strategies SUN-DSBO-ED-GT and SUN-DSBO-EXTRA-GT (where the upper-level variables use ED or EXTRA and the remaining variables use GT). Here, SUN-DSBO-ED and SUN-DSBO-EXTRA are consistent

with Figure 3, allowing us to compare the new SUN-DSBO algorithms with other algorithms by referencing Figure 3.

From Figure 14, we can observe that SUN-DSBO-ED and SUN-DSBO-EXTRA exhibit slightly faster convergence in the early stages, but their final accuracy is slightly lower compared to SUN-DSBO-GT. The hybrid strategies, SUN-DSBO-ED-GT and SUN-DSBO-EXTRA-GT, do not show significant improvements over the individual methods. Compared to algorithms incorporating heterogeneity correction techniques, SUN-DSBO-SE, which does not incorporate these techniques, shows more fluctuation in its performance.

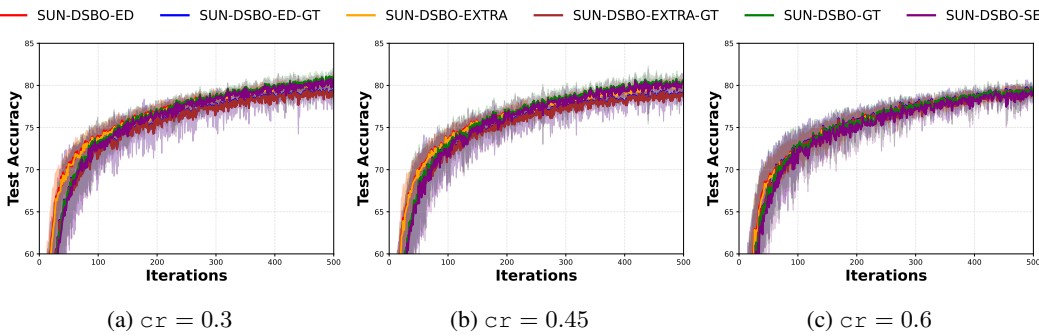

(a) $\mathtt{cr} = 0.3$        (b) $\mathtt{cr} = 0.45$        (c) $\mathtt{cr} = 0.6$

Figure 14: Comparison of SUN-DSBO variants for data hyper-cleaning across different corruption rates ($\mathtt{cr}$) with a heterogeneity parameter of $\mathcal{H} = 0.1$.

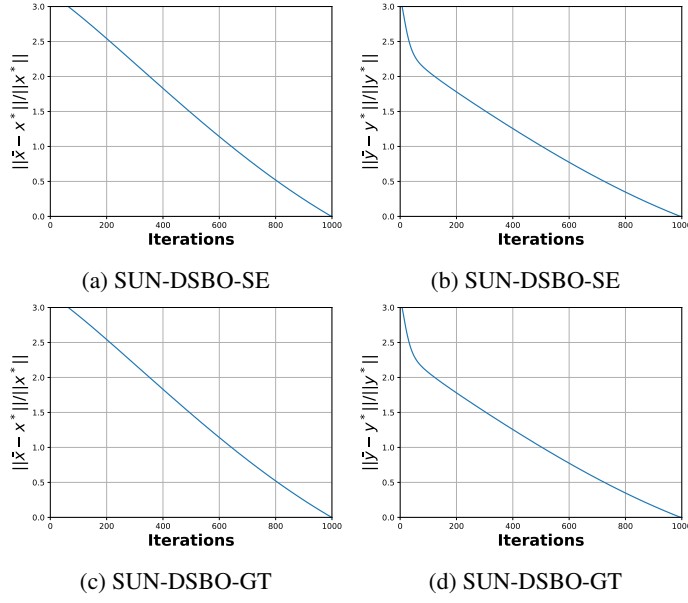

(a) SUN-DSBO-SE        (b) SUN-DSBO-SE

(c) SUN-DSBO-GT        (d) SUN-DSBO-GT

Figure 15: Illustrating the convergence curves of SUN-DSBO-SE/GT using the criteria $\frac{\|\bar{x} - x^*\|}{\|x^*\|}$ and $\frac{\|\bar{y} - y^*\|}{\|y^*\|}$, under the LL merely convex case, where $\bar{x}$ and $\bar{y}$ represent the average outputs of all agents with respect to $x$ and $y$.

**Synthetic numerical verification on lower-level merely convex case.** We demonstrate the effectiveness of the proposed method on a toy example in the LL merely convex case, expressed as

follows:

$$\min_{x \in \mathbb{R}^N, y=(y_1,y_2) \in \mathbb{R}^{2N}} \sum_{i=0}^{n-1} \left[ \frac{1}{2} \|A_i x - y_2\|^2 + \frac{1}{2} \|B_i y_1 - \mathbf{e}\|^2 \right]$$

$$\text{s.t.} \quad y = (y_1, y_2) \in \arg \min_{(y_1,y_2) \in \mathbb{R}^N} \sum_{i=0}^{n-1} \frac{1}{2} \|B_i y_1\|^2 - (A_i x)^\top y_1,$$

(9)

where $\mathbf{e}$ is the all-ones vector with a dimension depending on the problem, and $A_i$ and $B_i$ are node-specific diagonal matrices whose diagonal entries are all equal to $a_i$ and $b_i$, respectively. Here, we take $a_i = 1 + 0.1i$ and $b_i = 1 + 0.05i$ for $i = 0, \dots, n-1$. We use a 5-agent ring network in which each edge has weight $\frac{1}{3}$, and we set $N = 10$. For this experiment, we set the decaying coefficient to $\mu_k := 0.1(1 + k)^{-0.01}$, the penalty parameter to $\gamma = 10$, and the step sizes to $[0.01, 0.01, 0.1]$. The optimal solution for the lower-level problem is approximately $(1.43\mathbf{e}, 0.84\mathbf{e}, 1.58\mathbf{e})$. Since the lower-level admits an explicit solution, we substitute it into the upper-level and solve using a least-squares approximation. As shown in Figure 15, our proposed SUN-DSBO-SE/GT gradually approaches the correct solution.

### B.4 ALTERNATIVE TABULAR PRESENTATION OF MAIN EXPERIMENTS

For completeness, we provide the experimental results from the main text in tabular form, corresponding to the figures presented earlier.

Table 4 reports the results corresponding to Figure 1, presented in tabular form for clarity.

Table 5 reports the results corresponding to Figure 2, presented in tabular form for clarity.

Table 6 reports the results corresponding to Figure 3, presented in tabular form for clarity.

Tables 7 and 8 report the results corresponding to Figure 4, presented in tabular form for clarity.

Table 9 reports the results corresponding to Figure 5, presented in tabular form for clarity.

Table 4: Comparison of algorithms on hyper-cleaning with different regularization parameters, corresponding to Figure 1.

| $\alpha$ / **Iter** | 100 | 200 | 300 | 400 | 500 |
|---|---|---|---|---|---|
| | **SPARKLE-GT** (Zhu et al., 2024) | | | | |
| 0.001 | **69.48 ± 1.39** | **73.24 ± 1.39** | **75.34 ± 1.35** | **76.80 ± 0.28** | **77.30 ± 0.90** |
| 0.01 | 65.55 ± 2.81 | 70.90 ± 1.78 | 73.32 ± 1.62 | 74.83 ± 0.54 | 75.54 ± 0.90 |
| 0.1 | 64.10 ± 3.57 | 69.81 ± 1.95 | 72.11 ± 1.78 | 73.05 ± 1.12 | 73.50 ± 1.46 |
| | **DSGDA-GT** (Wang et al., 2024) | | | | |
| 0.001 | **68.59 ± 2.27** | **72.26 ± 1.29** | **74.75 ± 1.23** | **76.09 ± 0.55** | **75.63 ± 2.32** |
| 0.01 | 63.56 ± 4.24 | 69.67 ± 2.27 | 72.24 ± 1.51 | 73.26 ± 1.49 | 74.09 ± 1.74 |
| 0.1 | 63.61 ± 4.22 | 69.70 ± 2.26 | 72.31 ± 1.50 | 73.34 ± 1.46 | 74.15 ± 1.71 |
| | **SUN-DSBO-GT** (ours) | | | | |
| 0.001 | **73.32 ± 1.65** | **76.86 ± 1.06** | **78.92 ± 0.46** | **79.56 ± 0.94** | **79.89 ± 1.62** |
| 0.01 | 73.28 ± 1.68 | 76.79 ± 1.08 | 78.84 ± 0.43 | 79.45 ± 1.00 | 79.77 ± 1.71 |
| 0.1 | 69.46 ± 3.36 | 74.02 ± 1.62 | 75.98 ± 0.88 | 76.95 ± 1.10 | 77.46 ± 1.32 |

Table 5: Comparison of algorithms for data hyper-cleaning under different values of the heterogeneity parameter $\mathcal{H}$ with a corruption rate of cr $= 0.3$, corresponding to Figure 2.

| Algorithm / Iter | 100 | 200 | 300 | 400 | 500 |
|---|---|---|---|---|---|
| | | $\mathcal{H} = 0.1$ | | | |
| SUN-DSBO-GT | **73.32 ± 1.65** | **76.86 ± 1.06** | **78.92 ± 0.46** | **79.56 ± 0.94** | **79.89 ± 1.62** |
| SUN-DSBO-SE | 72.87 ± 1.79 | 76.48 ± 1.31 | 78.74 ± 0.37 | 79.38 ± 1.02 | 79.77 ± 1.15 |
| DSGDA-GT | 68.59 ± 2.27 | 72.26 ± 1.29 | 74.75 ± 1.23 | 76.09 ± 0.55 | 75.63 ± 2.32 |
| SPARKLE-GT | 69.67 ± 1.56 | 73.39 ± 1.36 | 74.98 ± 1.53 | 76.71 ± 0.30 | 76.73 ± 0.48 |
| SLDBO | 69.48 ± 1.35 | 73.31 ± 1.66 | 75.52 ± 1.53 | 76.80 ± 0.47 | 77.32 ± 1.09 |
| D-SOBA | 67.11 ± 3.29 | 71.68 ± 3.11 | 74.01 ± 2.12 | 75.43 ± 0.96 | 76.35 ± 1.03 |
| SPARKLE-EXTRA | 69.85 ± 1.24 | 73.42 ± 1.31 | 75.98 ± 0.28 | 76.84 ± 0.30 | 78.03 ± 0.59 |
| | | $\mathcal{H} = 0.5$ | | | |
| SUN-DSBO-GT | **77.24 ± 0.91** | **80.21 ± 0.55** | **81.79 ± 0.46** | **82.65 ± 0.32** | **83.36 ± 0.26** |
| SUN-DSBO-SE | 77.11 ± 0.92 | 80.12 ± 0.60 | 81.75 ± 0.47 | 82.60 ± 0.33 | 83.32 ± 0.28 |
| DSGDA-GT | 73.09 ± 0.95 | 76.02 ± 0.95 | 77.94 ± 0.51 | 78.91 ± 0.36 | 79.61 ± 0.47 |
| SPARKLE-GT | 72.83 ± 0.77 | 75.91 ± 0.57 | 77.54 ± 0.48 | 78.94 ± 0.43 | 79.52 ± 0.30 |
| SLDBO | 72.77 ± 0.67 | 75.97 ± 0.46 | 77.59 ± 0.33 | 79.01 ± 0.35 | 79.56 ± 0.29 |
| D-SOBA | 72.49 ± 0.86 | 75.60 ± 0.69 | 77.40 ± 0.64 | 78.73 ± 0.44 | 79.49 ± 0.35 |
| SPARKLE-EXTRA | 73.02 ± 0.74 | 76.12 ± 0.44 | 77.67 ± 0.35 | 79.09 ± 0.36 | 79.61 ± 0.25 |
| | | $\mathcal{H} = 1$ | | | |
| SUN-DSBO-GT | **78.48 ± 0.19** | **81.21 ± 0.12** | 82.44 ± 0.10 | **83.42 ± 0.11** | 84.17 ± 0.10 |
| SUN-DSBO-SE | 78.48 ± 0.19 | 81.21 ± 0.12 | **82.45 ± 0.10** | 83.42 ± 0.10 | **84.18 ± 0.08** |
| DSGDA-GT | 74.39 ± 0.16 | 77.28 ± 0.18 | 78.86 ± 0.17 | 80.01 ± 0.09 | 80.68 ± 0.10 |
| SPARKLE-GT | 72.96 ± 0.67 | 76.11 ± 0.33 | 77.98 ± 0.36 | 79.09 ± 0.26 | 79.95 ± 0.33 |
| SLDBO | 72.99 ± 0.59 | 76.12 ± 0.25 | 78.00 ± 0.32 | 79.08 ± 0.22 | 79.94 ± 0.27 |
| D-SOBA | 72.95 ± 0.68 | 76.10 ± 0.33 | 77.98 ± 0.35 | 79.08 ± 0.29 | 79.93 ± 0.33 |
| SPARKLE-EXTRA | 72.95 ± 0.68 | 76.10 ± 0.33 | 77.97 ± 0.35 | 79.10 ± 0.28 | 79.96 ± 0.32 |

Table 6: Comparison of algorithms for data hyper-cleaning across different corruption rates (cr) with a heterogeneity parameter of $\mathcal{H} = 0.1$, corresponding to Figure 3.

| Algorithm / Iter | 100 | 200 | 300 | 400 | 500 |
|---|---|---|---|---|---|
| cr = 0.3 | | | | | |
| SUN-DSBO-GT | **73.32 ± 1.65** | **76.86 ± 1.06** | **78.92 ± 0.46** | **79.56 ± 0.94** | **79.89 ± 1.62** |
| SUN-DSBO-SE | 72.87 ± 1.79 | 76.48 ± 1.31 | 78.74 ± 0.37 | 79.38 ± 1.02 | 79.77 ± 1.15 |
| DSGDA-GT | 68.59 ± 2.27 | 72.26 ± 1.29 | 74.75 ± 1.23 | 76.09 ± 0.55 | 75.63 ± 2.32 |
| SPARKLE-GT | 69.67 ± 1.56 | 73.39 ± 1.36 | 74.98 ± 1.53 | 76.71 ± 0.30 | 76.73 ± 0.48 |
| SLDBO | 69.48 ± 1.35 | 73.31 ± 1.66 | 75.52 ± 1.53 | 76.80 ± 0.47 | 77.32 ± 1.09 |
| D-SOBA | 67.11 ± 3.29 | 71.68 ± 3.11 | 74.01 ± 2.12 | 75.43 ± 0.96 | 76.35 ± 1.03 |
| SPARKLE-EXTRA | 69.85 ± 1.24 | 73.42 ± 1.31 | 75.98 ± 0.28 | 76.84 ± 0.30 | 78.03 ± 0.59 |
| cr = 0.45 | | | | | |
| SUN-DSBO-GT | **71.98 ± 2.79** | **76.55 ± 1.39** | **78.94 ± 0.76** | **79.68 ± 0.77** | **80.36 ± 1.28** |
| SUN-DSBO-SE | 71.55 ± 3.29 | 76.31 ± 1.51 | 78.64 ± 0.70 | 79.48 ± 0.90 | 80.12 ± 1.30 |
| DSGDA-GT | 65.71 ± 4.90 | 71.07 ± 2.44 | 74.24 ± 1.41 | 75.49 ± 1.05 | 76.82 ± 0.69 |
| SPARKLE-EXTRA | 69.42 ± 0.99 | 72.82 ± 0.53 | 74.76 ± 0.58 | 76.08 ± 0.65 | 76.56 ± 0.46 |
| SPARKLE-GT | 68.26 ± 1.87 | 72.02 ± 1.34 | 74.34 ± 1.32 | 75.69 ± 0.82 | 76.27 ± 0.56 |
| SLDBO | 67.77 ± 1.81 | 71.96 ± 1.31 | 74.37 ± 1.40 | 75.68 ± 0.93 | 76.24 ± 0.64 |
| D-SOBA | 66.33 ± 2.40 | 70.77 ± 2.34 | 73.05 ± 2.15 | 75.07 ± 0.92 | 75.44 ± 1.40 |
| cr = 0.6 | | | | | |
| SUN-DSBO-GT | **71.42 ± 2.95** | **75.92 ± 0.99** | **77.61 ± 0.57** | **78.53 ± 0.61** | **79.41 ± 0.52** |
| SUN-DSBO-SE | 70.73 ± 3.20 | 75.55 ± 1.06 | 77.22 ± 0.62 | 78.28 ± 0.65 | 79.14 ± 0.53 |
| DSGDA-GT | 66.60 ± 3.34 | 71.04 ± 2.82 | 73.49 ± 1.81 | 74.13 ± 2.86 | 74.91 ± 2.08 |
| SPARKLE-EXTRA | 64.79 ± 3.31 | 70.13 ± 2.01 | 72.71 ± 1.37 | 73.89 ± 1.46 | 74.64 ± 0.85 |
| SPARKLE-GT | 63.42 ± 3.07 | 68.97 ± 1.30 | 72.15 ± 1.19 | 73.21 ± 1.37 | 74.11 ± 0.32 |
| SLDBO | 64.29 ± 3.45 | 69.85 ± 2.16 | 72.41 ± 1.61 | 73.69 ± 1.63 | 74.27 ± 1.25 |
| D-SOBA | 62.87 ± 3.84 | 68.61 ± 3.32 | 72.30 ± 1.75 | 72.73 ± 2.39 | 73.64 ± 1.41 |

Table 7: Comparison of algorithms for hyper-representation with varying values of the heterogeneity parameter $\mathcal{H}$, corresponding to Figures 4 (a), (b), and (c).

| Algorithm / Iter | 100 | 200 | 300 | 400 | 500 |
|---|---|---|---|---|---|
| $\mathcal{H} = 0.1$, MLP on MNIST | | | | | |
| SUN-DSBO-GT | **90.25 ± 1.96** | **92.98 ± 1.41** | **94.22 ± 0.90** | **94.76 ± 0.93** | **95.40 ± 0.61** |
| SUN-DSBO-SE | 89.64 ± 2.29 | 92.55 ± 1.63 | 94.02 ± 0.85 | 94.33 ± 0.86 | 94.96 ± 0.91 |
| DSGDA-GT | 73.70 ± 2.91 | 81.80 ± 2.71 | 82.84 ± 5.01 | 72.34 ± 13.77 | 61.69 ± 12.22 |
| SPARKLE-EXTRA | 89.51 ± 1.26 | 91.88 ± 0.91 | 93.18 ± 0.70 | 93.88 ± 0.67 | 94.43 ± 0.53 |
| SPARKLE-GT | 89.78 ± 1.08 | 92.24 ± 0.88 | 93.49 ± 0.59 | 94.13 ± 0.60 | 94.65 ± 0.51 |
| SLDBO | 89.61 ± 1.36 | 92.01 ± 0.97 | 93.23 ± 0.75 | 94.05 ± 0.66 | 94.48 ± 0.52 |
| D-SOBA | 87.06 ± 2.99 | 91.01 ± 1.44 | 92.70 ± 0.94 | 93.53 ± 0.73 | 94.22 ± 0.65 |
| $\mathcal{H} = 0.5$, MLP on MNIST | | | | | |
| SUN-DSBO-GT | **93.53 ± 0.23** | **95.24 ± 0.18** | **96.05 ± 0.19** | **96.59 ± 0.12** | **96.87 ± 0.13** |
| SUN-DSBO-SE | 93.42 ± 0.26 | 95.19 ± 0.21 | 96.00 ± 0.23 | 96.54 ± 0.14 | 96.81 ± 0.12 |
| DSGDA-GT | 78.96 ± 3.54 | 85.10 ± 1.33 | 87.52 ± 0.70 | 88.63 ± 0.47 | 87.15 ± 2.51 |
| SPARKLE-EXTRA | 91.63 ± 0.30 | 93.53 ± 0.22 | 94.48 ± 0.18 | 95.09 ± 0.16 | 95.55 ± 0.13 |
| SPARKLE-GT | 91.56 ± 0.27 | 93.49 ± 0.19 | 94.43 ± 0.18 | 95.07 ± 0.13 | 95.54 ± 0.15 |
| SLDBO | 91.69 ± 0.32 | 93.58 ± 0.23 | 94.46 ± 0.19 | 95.11 ± 0.09 | 95.55 ± 0.11 |
| D-SOBA | 91.29 ± 0.32 | 93.32 ± 0.23 | 94.29 ± 0.22 | 94.96 ± 0.14 | 95.46 ± 0.14 |
| $\mathcal{H} = 1$, MLP on MNIST | | | | | |
| SUN-DSBO-GT | **94.04 ± 0.11** | **95.66 ± 0.12** | **96.42 ± 0.10** | **96.83 ± 0.08** | **97.10 ± 0.12** |
| SUN-DSBO-SE | 94.01 ± 0.10 | 95.66 ± 0.11 | 96.39 ± 0.11 | 96.82 ± 0.09 | 97.11 ± 0.10 |
| DSGDA-GT | 81.47 ± 0.16 | 86.24 ± 0.18 | 88.12 ± 0.12 | 89.15 ± 0.11 | 89.60 ± 0.23 |
| SPARKLE-EXTRA | 91.91 ± 0.16 | 93.82 ± 0.13 | 94.72 ± 0.14 | 95.37 ± 0.11 | 95.74 ± 0.11 |
| SPARKLE-GT | 91.80 ± 0.20 | 93.76 ± 0.15 | 94.67 ± 0.11 | 95.29 ± 0.13 | 95.71 ± 0.09 |
| SLDBO | 92.03 ± 0.18 | 93.86 ± 0.16 | 94.75 ± 0.10 | 95.32 ± 0.10 | 95.77 ± 0.13 |
| D-SOBA | 91.79 ± 0.20 | 93.76 ± 0.15 | 94.67 ± 0.11 | 95.29 ± 0.13 | 95.71 ± 0.08 |

Table 8: Comparison of algorithms for hyper-representation with varying heterogeneity parameter $\mathcal{H}$, corresponding to Figures 4 (d), (e), and (f).

| Algorithm / Iter | 500 | 1000 | 1500 | 2000 | 3000 |
|---|---|---|---|---|---|
| $\mathcal{H} = 0.1$, CNN on CIFAR-10 | | | | | |
| SUN-DSBO-GT | $26.86 \pm 2.30$ | $\mathbf{32.79 \pm 2.55}$ | $36.96 \pm 2.63$ | $39.11 \pm 2.66$ | $40.72 \pm 2.93$ |
| SUN-DSBO-SE | $\mathbf{27.64 \pm 3.44}$ | $32.58 \pm 2.06$ | $\mathbf{37.35 \pm 2.63}$ | $\mathbf{40.04 \pm 2.83}$ | $\mathbf{42.97 \pm 4.37}$ |
| SPARKLE-EXTRA | $23.90 \pm 1.64$ | $29.38 \pm 2.67$ | $30.54 \pm 4.37$ | $33.85 \pm 3.81$ | $35.64 \pm 4.34$ |
| SPARKLE-GT | $23.39 \pm 2.72$ | $27.79 \pm 2.33$ | $29.75 \pm 3.41$ | $30.84 \pm 2.47$ | $36.40 \pm 2.98$ |
| SLDBO | $24.13 \pm 2.40$ | $30.50 \pm 2.15$ | $34.83 \pm 2.25$ | $36.65 \pm 1.96$ | $41.27 \pm 2.77$ |
| D-SOBA | $25.55 \pm 1.23$ | $28.76 \pm 3.81$ | $31.62 \pm 2.54$ | $31.89 \pm 3.01$ | $36.75 \pm 3.69$ |
| $\mathcal{H} = 0.5$, CNN on CIFAR-10 | | | | | |
| SUN-DSBO-GT | $35.09 \pm 4.43$ | $41.68 \pm 3.78$ | $45.77 \pm 3.23$ | $48.66 \pm 2.63$ | $52.19 \pm 2.30$ |
| SUN-DSBO-SE | $\mathbf{36.17 \pm 2.59}$ | $\mathbf{42.42 \pm 3.12}$ | $\mathbf{46.80 \pm 3.27}$ | $\mathbf{49.99 \pm 1.42}$ | $\mathbf{52.31 \pm 2.85}$ |
| SPARKLE-EXTRA | $33.13 \pm 2.93$ | $37.69 \pm 1.99$ | $39.66 \pm 1.68$ | $42.67 \pm 2.40$ | $46.19 \pm 1.66$ |
| SPARKLE-GT | $32.72 \pm 3.24$ | $37.14 \pm 2.73$ | $40.77 \pm 2.69$ | $42.75 \pm 2.64$ | $45.90 \pm 2.36$ |
| SLDBO | $32.87 \pm 3.00$ | $39.13 \pm 2.35$ | $43.17 \pm 2.08$ | $45.08 \pm 2.12$ | $47.97 \pm 1.95$ |
| D-SOBA | $32.64 \pm 0.74$ | $35.55 \pm 1.97$ | $38.68 \pm 2.20$ | $40.11 \pm 2.57$ | $43.94 \pm 2.13$ |
| $\mathcal{H} = 1$, CNN on CIFAR-10 | | | | | |
| SUN-DSBO-GT | $38.69 \pm 3.45$ | $49.19 \pm 1.38$ | $53.40 \pm 1.07$ | $56.22 \pm 1.61$ | $58.07 \pm 1.23$ |
| SUN-DSBO-SE | $\mathbf{39.64 \pm 3.07}$ | $\mathbf{49.70 \pm 1.00}$ | $\mathbf{54.56 \pm 1.14}$ | $\mathbf{57.34 \pm 1.36}$ | $\mathbf{58.65 \pm 1.52}$ |
| SPARKLE-EXTRA | $39.16 \pm 2.58$ | $47.32 \pm 1.31$ | $51.15 \pm 0.63$ | $53.75 \pm 0.80$ | $56.73 \pm 0.86$ |
| SPARKLE-GT | $39.56 \pm 2.22$ | $47.74 \pm 1.06$ | $51.69 \pm 0.62$ | $54.22 \pm 0.99$ | $57.05 \pm 0.80$ |
| SLDBO | $38.37 \pm 1.75$ | $45.79 \pm 1.09$ | $49.40 \pm 0.74$ | $51.84 \pm 1.32$ | $55.68 \pm 0.60$ |
| D-SOBA | $39.62 \pm 2.15$ | $47.59 \pm 1.10$ | $51.84 \pm 0.62$ | $54.21 \pm 0.90$ | $56.92 \pm 0.86$ |

Table 9: Data hyper-cleaning is performed with varying values of $\rho$, which quantifies the connectivity of the communication network, corresponding to Figure 5.

| $\rho$ / Iter | 100 | 200 | 300 | 400 | 500 |
|---|---|---|---|---|---|
| SUN-DSBO-SE | | | | | |
| $\rho = 0.798$ | $\mathbf{77.32 \pm 0.65}$ | $\mathbf{80.24 \pm 0.53}$ | $\mathbf{81.76 \pm 0.34}$ | $\mathbf{82.60 \pm 0.25}$ | $\mathbf{83.32 \pm 0.25}$ |
| $\rho = 0.824$ | $77.28 \pm 0.78$ | $80.22 \pm 0.54$ | $81.72 \pm 0.37$ | $82.60 \pm 0.26$ | $83.26 \pm 0.23$ |
| $\rho = 0.905$ | $77.11 \pm 0.92$ | $80.12 \pm 0.60$ | $81.75 \pm 0.47$ | $82.60 \pm 0.33$ | $\mathbf{83.32 \pm 0.28}$ |
| $\rho = 0.924$ | $76.95 \pm 1.02$ | $79.98 \pm 0.68$ | $81.66 \pm 0.47$ | $82.49 \pm 0.37$ | $83.22 \pm 0.29$ |
| $\rho = 0.970$ | $72.99 \pm 0.59$ | $76.12 \pm 0.25$ | $78.00 \pm 0.32$ | $79.08 \pm 0.22$ | $79.94 \pm 0.27$ |
| SUN-DSBO-GT | | | | | |
| $\rho = 0.798$ | $\mathbf{77.48 \pm 0.61}$ | $\mathbf{80.39 \pm 0.46}$ | $\mathbf{81.84 \pm 0.31}$ | $\mathbf{82.72 \pm 0.23}$ | $\mathbf{83.38 \pm 0.28}$ |
| $\rho = 0.824$ | $77.43 \pm 0.75$ | $80.38 \pm 0.50$ | $81.82 \pm 0.35$ | $82.72 \pm 0.22$ | $83.38 \pm 0.25$ |
| $\rho = 0.905$ | $77.24 \pm 0.91$ | $80.21 \pm 0.55$ | $81.79 \pm 0.46$ | $82.65 \pm 0.32$ | $83.36 \pm 0.26$ |
| $\rho = 0.924$ | $77.07 \pm 1.02$ | $80.12 \pm 0.61$ | $81.71 \pm 0.48$ | $82.58 \pm 0.34$ | $83.29 \pm 0.29$ |
| $\rho = 0.970$ | $76.62 \pm 1.04$ | $79.55 \pm 0.55$ | $81.01 \pm 0.57$ | $81.75 \pm 0.59$ | $82.33 \pm 0.59$ |

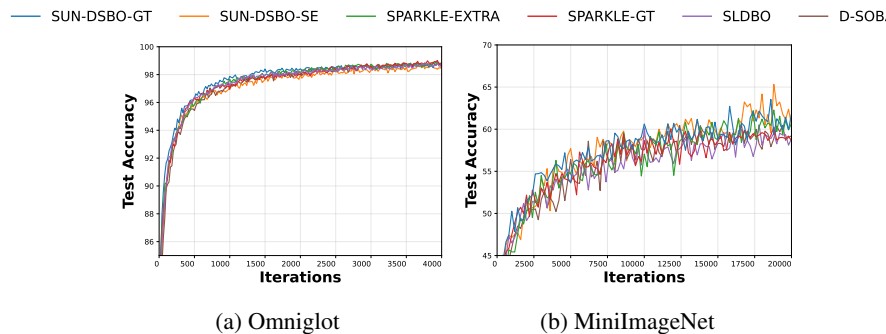

(a) Omniglot                    (b) MiniImageNet

Figure 16: Comparison of algorithms for meta-learning on the Omniglot and MiniImageNet datasets.

Table 10: Comparison of algorithms for meta-learning on Omniglot dataset.

| Method | 800 | 1600 | 2400 | 3200 | 4000 |
|---|---|---|---|---|---|
| SUN-DSBO-GT | **97.42** | **98.25** | 98.35 | 98.51 | 98.60 |
| SUN-DSBO-SE | 96.84 | 97.80 | 97.98 | 98.34 | 98.62 |
| SPARKLE-EXTRA | 97.19 | 98.21 | **98.39** | 98.62 | 98.81 |
| SPARKLE-GT | 97.04 | 98.06 | 98.36 | 98.64 | **98.88** |
| SLDBO | 97.21 | 98.20 | 98.31 | 98.45 | 98.84 |
| D-SOBA | 96.50 | 98.04 | 98.22 | **98.68** | 98.85 |

## B.5 META-LEARNING

We consider a decentralized meta-learning problem, inspired by Zhu et al. (2024), with the problem formulation and experimental setup detailed in Appendix C.5. The goal of the task is to learn a model that can generalize to new, unseen tasks by training on a variety of tasks with limited data. Our experiments are conducted on the following benchmark datasets: Omniglot (Lake et al., 2015) and MiniImageNet (Vinyals et al., 2016), both using a 4-layer CNN backbone, with different channel widths (64 for Omniglot, 32 for MiniImageNet). Omniglot is a few-shot learning dataset consisting of 1,623 handwritten characters from 50 different alphabets. MiniImageNet is a subset of the larger ImageNet dataset (Deng et al., 2009), containing 100 classes with 600 images per class.

In this section, we evaluate SUN-DSBO-GT/SE, SPARKLE-EXTRA, SPARKLE-GT, SLDBO, and D-SOBA using a heterogeneity parameter $\mathcal{H} = 1$ on ring topologies in a 5-way-5-shot setting. The task sampling strategy is as follows: for training, validation, and testing, we sample 20k, 200, and 200 tasks, respectively. Each task consists of 5-way 5-shot, with support and query sets containing 5×2 images each. During evaluation, the results are averaged over multiple tasks. As shown in Figure 16 (a) and Table 10, all algorithms perform similarly on the Omniglot dataset, with SUN-DSBO-GT being slightly faster in the early stages. However, neither SUN-DSBO-GT nor SUN-DSBO-SE achieves higher accuracy later on compared to the other algorithms. SPARKLE-GT achieves the highest accuracy, though the difference from the other algorithms is minimal, as most algorithms have relatively converged and stabilized. As shown in Figure 16 (b) and Table 11, SUN-DSBO-GT performs better overall on the MiniImageNet. However, SUN-DSBO-SE achieves the highest peak accuracy, with SUN-DSBO-GT securing the second-highest peak accuracy during the iterative process, as shown in Figure 16 (b).

Table 11: Comparison of algorithms for meta-learning on the MiniImageNet datasets.

| Method | 4000 | 8000 | 12000 | 16000 | 20000 |
|---|---|---|---|---|---|
| SUN-DSBO-GT | 55.88 | **59.54** | **60.70** | **60.42** | **61.82** |
| SUN-DSBO-SE | 55.94 | 56.16 | 60.30 | 59.42 | 60.54 |
| SPARKLE-EXTRA | **56.30** | 56.02 | 54.52 | 59.40 | 60.68 |
| SPARKLE-GT | 54.74 | 56.72 | 58.46 | 58.22 | 58.62 |
| SLDBO | 54.10 | 54.94 | 55.90 | 59.56 | 58.88 |
| D-SOBA | 50.22 | 54.84 | 58.76 | 58.44 | 59.20 |

## C  DETAILS OF EXPERIMENTS

### C.1  HYPERPARAMETER TUNING STRATEGY

*In our experiments, hyperparameter tuning is not overly complex and does not significantly exceed the complexity of hyperparameter adjustment in recent popular algorithms presented in Zhu et al. (2024); Wang et al. (2024).* Specifically, our hyperparameters include the proximal parameter $\gamma$, penalty parameter $\mu_k$, and the corresponding step-sizes $\lambda_x^k$, $\lambda_y^k$, $\lambda_\theta^k$. Our analysis reveals that different hyperparameters require varying levels of tuning precision. The proximal parameter $\gamma$ has minimal impact on algorithm performance, as demonstrated in item (i) of Subsection 2.1 and the corresponding experimental results in "Results with different $\gamma$ in (5)" (Subsection B.3). For the penalty parameter $\mu_k$, we observe that the initial value $\mu_0$ has significantly less influence than the power parameter $p$. Smaller values of $p$ tend to yield better experimental performance, which is related to the stationarity measure as discussed in Remark D.1. When $p$ is relatively small, its specific value has little effect on performance, a finding corroborated by our experimental results in "Results with different $p$" corresponding to Theorems 3.5 and 3.7. Based on these observations, we adopt a two-tier tuning strategy:

- *Coarse tuning*: For the proximal parameter $\gamma$ and penalty parameter $\mu_k$, we employ coarse grid search with large intervals, as these parameters are relatively insensitive.

- *Fine tuning*: For the step-sizes $\lambda_x^k$, $\lambda_y^k$, $\lambda_\theta^k$, we perform fine-grained grid search with small intervals. Similarly, the algorithms in Zhu et al. (2024); Wang et al. (2024) also require fine-tuning of three step-sizes. Additionally, the algorithm in Zhu et al. (2024) requires tuning three momentum coefficients, while the algorithm in Wang et al. (2024) may require tuning an inner loop iteration count.

The above analysis demonstrates that our algorithm's hyperparameter tuning is not excessively complex and maintains comparable tuning requirements to existing popular methods.

Next, we present the specific configurations used in the experiments outlined in Section 4 and B.

### C.2  DATA HYPER-CLEANING

We evaluate the proposed algorithms on the data hyper-cleaning task (Zhu et al., 2024) using Fashion-MNIST. The dataset contains 60,000 training and 10,000 test images. We randomly split the training set into 50,000 training and 10,000 validation samples. A corruption rate `cr` is applied to introduce noise into the training labels, while the validation set remains clean. The goal is to learn a weighting policy over the noisy training data such that the model trained on the weighted dataset generalizes well to the validation set.

This task can be formulated as a decentralized bilevel optimization (DBO) problem, where the upper level optimizes the data-cleaning policy, and the lower level solves the client-specific training objectives under the learned policy:

$$
\min_{\boldsymbol{\psi},\boldsymbol{w}} F(\boldsymbol{\psi},\boldsymbol{w}) = \frac{1}{n}\sum_{i=1}^{n}\frac{1}{|\mathcal{D}_{\text{val}}^i|}\sum_{(\mathbf{x}_j,y_j)\in\mathcal{D}_{\text{val}}^i}\mathcal{L}\big(h(\mathbf{x}_j^\top;\boldsymbol{w}),y_j\big),
$$
$$
\text{s.t. } \boldsymbol{w}\in\arg\min_{\boldsymbol{w}'}f(\boldsymbol{\psi},\boldsymbol{w}') = \frac{1}{n}\sum_{i=1}^{n}\frac{1}{|\mathcal{D}_{\text{tr}}^i|}\sum_{(\mathbf{x}_j,y_j)\in\mathcal{D}_{\text{tr}}^i}\sigma(\psi_j)\mathcal{L}\big(h(\mathbf{x}_j^\top;\boldsymbol{w}'),y_j\big)+\alpha\|\boldsymbol{w}'\|^2,
$$

(10)

where $\mathcal{D}_{\text{tr}}^i$ and $\mathcal{D}_{\text{val}}^i$ denote the local training and validation datasets of client $i$, respectively; $(\mathbf{x}_j,y_j)$ is the $j$-th data sample and label; $\sigma(\cdot)$ is the sigmoid function; $\mathcal{L}$ is the cross-entropy loss; $h$ is a two-layer MLP with a 300-dimensional hidden layer and ReLU activation; and $\boldsymbol{w}$ represents the model parameters. The regularization parameter is set to $\alpha = 0.001$, and the batch size is 50.

All experiments in this subsection were conducted using Python 3.7 on a machine with an Intel(R) Xeon(R) Gold 5218R CPU @ 2.10GHz and an NVIDIA A100 GPU (40GB memory).

**Settings under varying $\alpha$.** In Figure 1, we evaluate all methods using 10 agents arranged in a ring communication topology in Figure 17 (a). The weight matrix $W = (w_{ij})$ is doubly stochastic, with

$$w_{i,i} = a, \quad w_{i,i\pm1} = \tfrac{1-a}{2}, \quad \text{and } a = 0.5,$$

while all other entries are set to zero. We set the heterogeneity parameter of the Dirichlet distribution to $\mathcal{H} = 0.1$, the label corruption rate to $\mathtt{cr} = 0.3$, and vary the regularization parameter $\alpha \in \{0.001, 0.01, 0.1\}$.

For the algorithmic configurations, SPARKLE-GT incorporates a moving-average term with weight 0.2; DSGDA-GT uses a penalty parameter $\nu = 0.1$; and SUN-DSBO-GT adopts a decaying sequence $\mu_k := 2(1 + k)^{-0.001}$ and sets $1/\gamma = 0.015$. The specific step sizes used for each method are summarized in Table 12. For SUN-DSBO-GT, the step sizes are given as $[\lambda_x^k, \lambda_y^k, \lambda_\theta^k]$; for DSGDA-GT and SPARKLE-GT, the step sizes are given as $[\lambda_x^k, \lambda_y^k, \lambda_z^k]$.

Table 12: Step sizes used in the data hyper-cleaning task under different $\alpha$ values.

| $\alpha$ | SUN-DSBO-GT | DSGDA-GT | SPARKLE-GT |
|---|---|---|---|
| 0.001 | [0.03, 0.02, 0.01] | [0.02, 0.02, 0.01] | [0.03, 0.03, 0.03] |
| 0.01 | [0.02, 0.02, 0.01] | [0.02, 0.01, 0.01] | [0.02, 0.02, 0.03] |
| 0.1 | [0.02, 0.02, 0.01] | [0.02, 0.01, 0.01] | [0.02, 0.02, 0.03] |

**Settings under varying $\mathcal{H}$.** In Figures 2 and 6, we evaluate all methods using 10 agents connected via a ring communication topology. The weight matrix $W = (w_{ij})$ is doubly stochastic, defined as

$$w_{i,i} = a, \quad w_{i,i\pm1} = \tfrac{1-a}{2}, \quad \text{with } a = 0.5,$$

and all other entries are set to zero. The heterogeneity parameter of the Dirichlet distribution is varied as $\mathcal{H} \in \{0.001, 0.01, 0.1\}$, while the corruption rate is fixed to $\mathtt{cr} = 0.3$ and the regularization parameter to $\alpha = 0.001$.

The step sizes are configured as follows. For D-SOBA (with moving-average term 0.2), the steps are $[\lambda_x^k, \lambda_y^k, \lambda_z^k] = [0.03, 0.03, 0.03]$; for SLDBO, $[0.03, 0.03, 0.03]$; for SPARKLE-EXTRA/GT (with moving-average term 0.2), $[0.03, 0.03, 0.03]$; for DSGDA-GT (with penalty parameter $\nu = 0.1$), $[0.02, 0.02, 0.01]$; and for SUN-DSBO-GT/SE (with decaying coefficient $\mu_k := 2(1 + k)^{-0.001}$ and $\gamma = 0.015$), the step sizes are $[\lambda_x^k, \lambda_y^k, \lambda_\theta^k] = [0.03, 0.02, 0.01]$.

**Settings under varying $\mathtt{cr}$.** In Figures 3, 7, and 8, we evaluate the effect of different corruption rates. All methods are tested with 10 agents connected via a ring communication topology, where the weight matrix $W = (w_{ij})$ is doubly stochastic, defined as

$$w_{i,i} = a, \quad w_{i,i\pm1} = \tfrac{1-a}{2}, \quad \text{with } a = 0.5,$$

and all other entries are zero. The heterogeneity parameter of the Dirichlet distribution is fixed at $\mathcal{H} = 0.1$, while the corruption rate varies as $\mathtt{cr} \in \{0.3, 0.45, 0.6\}$ and the regularization parameter is set to $\alpha = 0.001$.

The step sizes are configured as follows. For D-SOBA (with moving-average term 0.2), the steps are $[\lambda_x^k, \lambda_y^k, \lambda_z^k] = [0.03, 0.03, 0.03]$; for SLDBO, $[0.03, 0.03, 0.03]$; for SPARKLE-EXTRA/GT (with moving-average term 0.2), $[0.03, 0.03, 0.03]$; for DSGDA-GT (with penalty parameter $\nu = 0.1$), $[0.02, 0.02, 0.01]$; and for SUN-DSBO-GT/SE (with decaying coefficient $\mu_k := 2(1 + k)^{-0.001}$ and $\gamma = 0.015$), the step sizes are $[\lambda_x^k, \lambda_y^k, \lambda_\theta^k] = [0.03, 0.02, 0.01]$. For D-PSGD and GNSD, the step size $\lambda_x^k$ is set to 0.1.

**Settings under different communication topologies.** All experiments are conducted with 10 agents under various communication topologies, including exponential, ring, and line graphs.

For the *exponential topology* in Figure 17 (a), the weight matrix $W = (w_{ij})$ is doubly stochastic and defined as

$$w_{i,m} = \begin{cases} \dfrac{1}{8}, & \text{if there exists } j \in \{1, \ldots, 4\} \text{ such that } m \equiv i \pm (2^j - 1) \pmod{n}, \\ \dfrac{1}{4}, & \text{if } m = i, \\ 0, & \text{otherwise}, \end{cases}$$

where $\pm$ denotes symmetric bidirectional connections. Each self-loop has weight $\frac{1}{4}$, and each pair of neighbors shares total weight $\frac{1}{4}$, split evenly as $\frac{1}{8}$ in each direction.

For the *ring topology* in Figure 17 (b), the weight matrix is defined as

$$w_{i,i} = a, \quad w_{i,i\pm1} = \frac{1-a}{2}, \quad a \in \{0.2, 0.4, 0.5\},$$

with all other entries set to zero.

For the *line topology* in Figure 17 (c), the communication matrix is given by

$$w_{i,j} = \begin{cases} 1 & \text{if } (i = 0 \wedge j = 1) \vee (i = n - 1 \wedge j = n - 2), \\ 0.5 & \text{if } 1 \leq i \leq n - 2 \text{ and } |j - i| = 1, \\ 0 & \text{otherwise}, \end{cases}$$

where $\wedge$ denotes logical AND and $\vee$ denotes logical OR. The corresponding connectivity parameter $\rho$ for these topologies takes values in $\{0.798, 0.824, 0.905, 0.924, 0.970\}$.

The heterogeneity parameter for the Dirichlet distribution is set to $\mathcal{H} = 0.5$, with corruption rate $\mathtt{cr} = 0.3$ and regularization parameter $\alpha = 0.001$. For SUN-DSBO-GT and SUN-DSBO-SE, we adopt a decaying coefficient $\mu_k := 2(1 + k)^{-0.001}$ and fixed $\gamma = 0.015$. The step sizes are $[\lambda_x^k, \lambda_y^k, \lambda_\theta^k] = [0.03, 0.02, 0.01]$.

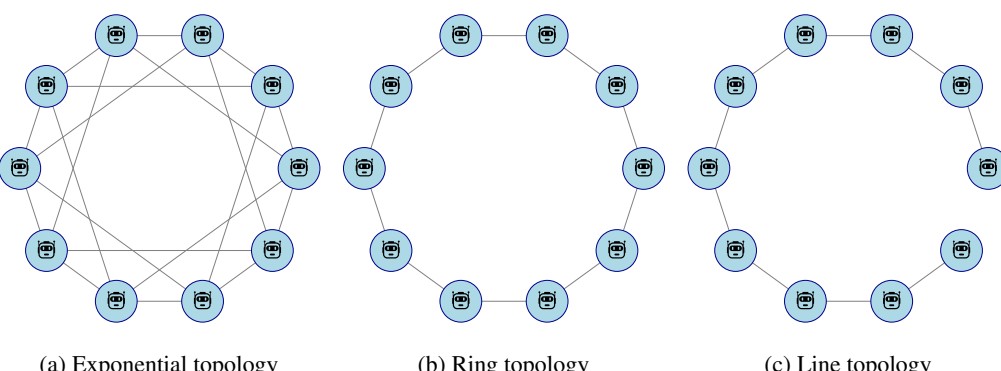

(a) Exponential topology      (b) Ring topology      (c) Line topology

Figure 17: Illustration of the communication topologies used in the experiments: (a) exponential graph with multi-hop connections, (b) ring graph with uniform nearest-neighbor links, and (c) line graph with fixed endpoints. All graphs are undirected and designed to ensure a doubly stochastic communication matrix.

**Settings with different number $n$ of agents.** In Figure 12 , we evaluate all methods using 10 agents connected via a ring communication topology. The weight matrix $W = (w_{ij})$ is doubly stochastic, defined as

$$w_{i,i} = a, \quad w_{i,i\pm1} = \frac{1-a}{2}, \quad \text{with } a = 0.5,$$

and all other entries are set to zero. We set the heterogeneity parameter of the Dirichlet distribution to $\mathcal{H} = 0.1$, while fixing the corruption rate to $\mathtt{cr} = 0.3$ and the regularization parameter to $\alpha = 0.001$. The number of agents is varied as $n \in \{10, 15, 20\}$. For Sun-DSBO-GT and Sun-DSBO-SE, we use a decaying penalty coefficient $\mu_k := 2(1 + k)^{-0.001}$ and set $\gamma = 50$. The step sizes are fixed as $[\lambda_x^k, \lambda_y^k, \lambda_\theta^k] = [0.03, 0.02, 0.01]$.

**Settings with SUN-DSBO Extended Algorithms.** The hyperparameter settings are the same as those in Figure 3.

### C.3 HYPER-REPRESENTATION

Inspired by Franceschi et al. (2018); Tarzanagh et al. (2022), we consider a hyper-representation problem designed to optimize a classification model in a two-stage process under a decentralized

setting. The outer-level objective learns a shared representation by minimizing validation loss, while the inner-level objective learns a local classifier on the training data. The bilevel formulation is given as:

$$\min_{x,\, y^*} \frac{1}{n} \sum_{i=1}^{n} f_{\text{ce}}(x, y^*(x);\, \mathcal{D}_{\text{val}}^i)$$

$$\text{s.t.} \quad y^*(x) \in \arg\min_{y} \frac{1}{n} \sum_{i=1}^{n} f_{\text{ce}}(x, y;\, \mathcal{D}_{\text{tr}}^i) + \alpha \|y\|^2, \tag{11}$$

where $x$ denotes the parameters of the representation (backbone) network, and $y$ the parameters of the classifier (head). Each client $i$ has its own local training dataset $\mathcal{D}_{\text{tr}}^i$ and validation dataset $\mathcal{D}_{\text{val}}^i$.

The cross-entropy loss is defined as:

$$f_{\text{ce}}(x, y; \mathcal{D}) := -\frac{1}{|\mathcal{D}|} \sum_{d_m \in \mathcal{D}} \log \frac{\exp\left(h_{l_m}(x, y; d_m)\right)}{\sum_{c=1}^{C} \exp\left(h_c(x, y; d_m)\right)},$$

where $C$ is the number of classes, $d_m$ is the $m$-th data point with label $l_m$, and $h(x, y; d_m) = [h_1(x, y; d_m), \ldots, h_C(x, y; d_m)]^\top \in \mathbb{R}^C$ denotes the model output (logits) for input $d_m$.

All experiments in this subsection were conducted using Python 3.7 on a machine equipped with an Intel(R) Xeon(R) Gold 5218R CPU @ 2.10GHz and an NVIDIA A100 GPU with 40GB of memory.

**Settings with MNIST and MLP.** In Figures 4(a)–(c) and 9(a)–(c), we evaluate the proposed algorithms on the hyper-representation problem using the MNIST dataset. The dataset contains 60,000 training and 10,000 test images. We randomly split the training set into 50,000 training and 10,000 validation samples. The model is a two-layer MLP with a 200-dimensional hidden layer and ReLU activation.

All methods are evaluated using 10 agents arranged in a ring communication topology. The batch size is set to 30. The weight matrix $W = (w_{ij})$ is doubly stochastic, with

$$w_{i,i} = a, \quad w_{i,i\pm1} = \frac{1-a}{2}, \quad \text{and } a = 0.5,$$

while all other entries are set to zero. We set the heterogeneity parameter of the Dirichlet distribution to $\mathcal{H} \in \{0.1, 0.5, 1\}$, the label corruption rate to $\text{cr} = 0.3$, and the regularization parameter to $\alpha = 0.001$.

The step sizes are configured as follows. For D-SOBA (with moving-average term 0.2), the steps are $[0.03, 0.03, 0.03]$; for SLDBO, $[0.03, 0.03, 0.03]$; for SPARKLE-EXTRA/GT (with moving-average term 0.2), $[0.03, 0.03, 0.03]$; for DSGDA-GT (with penalty parameter $\nu = 0.1$), $[0.02, 0.02, 0.01]$; and for SUN-DSBO-GT (with decaying coefficient $\mu_k := 2(1 + k)^{-0.001}$ and $\gamma = 50$), $[0.03, 0.02, 0.01]$.

**Settings with CIFAR and CNN.** In Figures 4(d)–(f) and 9(d)–(f), we evaluate the proposed algorithms on the hyper-representation problem using the CIFAR-10 dataset. The dataset contains 60,000 training and 10,000 test images. We randomly split the training set into 50,000 training and 10,000 validation samples. The model is a 7-layer CNN (LeCun et al., 1998).

All methods are evaluated using 10 agents arranged in a ring communication topology. The batch size is set to 30. The weight matrix $W = (w_{ij})$ is doubly stochastic, with

$$w_{i,i} = a, \quad w_{i,i\pm1} = \frac{1-a}{2}, \quad \text{and } a = 0.5,$$

while all other entries are set to zero. We set the heterogeneity parameter of the Dirichlet distribution to $\mathcal{H} \in \{0.1, 0.5, 1\}$, the label corruption rate to $\text{cr} = 0.3$, and the regularization parameter to $\alpha = 0.001$.

The step sizes are configured as follows. For D-SOBA (with moving-average term 0.2), the steps are $[0.1, 0.1, 0.05]$; for SLDBO, $[0.1, 0.1, 0.05]$; for SPARKLE-EXTRA/GT (with moving-average term 0.2), $[0.1, 0.1, 0.05]$; and for SUN-DSBO-GT/SE (with decaying coefficient $\mu_k := (1 + k)^{-0.001}$ and $\gamma = 0.02$), $[0.05, 0.05, 0.05]$. For the ablation study on the effect of the hyperparameter $p$ (Table 2), we set the decaying coefficient as $\mu_k := (1 + k)^{-0.001}$ and $\gamma = 0.02$, and use step sizes $[0.05, 0.05, 0.05]$ for the hyper-representation task. For the ablation study on the effect of the hyperparameter $\gamma$ (Table 3), we consider the following step sizes and $\mu_k$ schedules in the hyper-representation task:

Table 13: Step sizes used in the hyper-representation task under different $\gamma$ values.

| $\gamma$ | 5 | 0.5 | 0.05 | 0.005 |
|---|---|---|---|---|
| $\mu_k$ | $2(1+k)^{-0.001}$ | $2(1+k)^{-0.001}$ | $2(1+k)^{-0.001}$ | $(1+k)^{-0.001}$ |
| Step sizes | $[0.15, 0.15, 0.15]$ | $[0.07, 0.07, 0.07]$ | $[0.07, 0.015, 0.015]$ | $[0.07, 0.015, 0.015]$ |

## C.4 ABLATION STUDIES

Following Chen et al. (2023; 2025), we study decentralized hyperparameter optimization for $\ell^2$-regularized logistic regression. The objective is to identify the optimal regularization coefficient $\pi$ under the constraint that the lower-level model parameters $\tau^*(\pi)$ minimize the training loss.

**Settings with synthetic data.** We first evaluate the performance on heterogeneous synthetic data (Pedregosa, 2016; Grazzi et al., 2020). On node $i$, the upper- and lower-level objectives are defined as

$$f_i\big(\pi, \tau^*(\pi)\big) = \sum_{(x_e, y_e) \in \mathcal{D}_i^{\mathrm{val}}} \mathcal{L}\big(y_e\, x_e^\mathsf{T} \tau^*(\pi)\big),$$

$$g_i(\pi, \tau) = \sum_{(x_e, y_e) \in \mathcal{D}_i^{\mathrm{tr}}} \mathcal{L}\big(y_e\, x_e^\mathsf{T} \tau\big) + \tfrac{1}{2}\, \tau^\mathsf{T} \mathrm{diag}\big(e^\pi\big)\, \tau,$$

where $\mathcal{L}(x) = \log(1 + e^{-x})$, and $\mathrm{diag}(v)$ denotes a diagonal matrix formed from vector $v$. The lower-level optimal parameter is given by $\tau^*(\pi) = \arg\min_\tau g_i(\pi, \tau)$.

We generate a ground-truth parameter $\tau^* \in \mathbb{R}^s$ and a noise vector $\beta \in \mathbb{R}^t$. Each feature vector $x_e$ on node $i$ is drawn from the Gaussian distribution $\mathcal{N}(0, i^2)$, and the label is assigned as $y_e = \mathrm{sign}(x_e^\mathsf{T} \tau^* + 0.1\, \beta)$, with a noise rate of $\beta = 0.1$.

In Figures 10, all methods are evaluated using 8 agents under various communication topologies. For the ring topology, the weight matrix $W = (w_{ij})$ is doubly stochastic, defined by

$$w_{i,i} = a, \quad w_{i,i\pm 1} = \tfrac{1-a}{2}, \quad \text{with } a = 0.4,$$

and all other entries are set to zero.

All experiments are conducted on a machine with Intel(R) Core(TM) i7-10510U CPU @ 1.80GHz. The step sizes are configured as follows. For SUN-DSBO-GT/SE (with coefficient $\mu_k := 1.5(1 + k)^{-0.001}$ and $\gamma = 0.9$), the steps are $[0.01, 0.02, 0.03]$.

**Settings with MNIST.** We evaluate all algorithms on a hyperparameter optimization task for logistic regression using the MNIST dataset. We select 20,000 training samples and 5,000 test samples from the dataset. On node $i$, the bilevel objectives are defined as

$$f_i\big(\pi, \tau^*(\pi)\big) = \frac{1}{|\mathcal{D}_{\mathrm{val}}^{(i)}|} \sum_{(x_e, y_e) \in \mathcal{D}_{\mathrm{val}}^{(i)}} L\big(x_e^\mathsf{T} \tau^*(\pi), y_e\big),$$

$$g_i(\pi, \tau) = \frac{1}{|\mathcal{D}_{\mathrm{tr}}^{(i)}|} \sum_{(x_e, y_e) \in \mathcal{D}_{\mathrm{tr}}^{(i)}} L\big(x_e^\mathsf{T} \tau, y_e\big) + \frac{1}{c\,s} \sum_{j=1}^s e^{\pi_j} \sum_{k=1}^c \tau_{kj}^2,$$

where $c = 10$ is the number of classes, $s = 784$ is the feature dimension, $L(\cdot, \cdot)$ denotes the cross-entropy loss, and $\mathcal{D}_{\mathrm{tr}}^{(i)}$ and $\mathcal{D}_{\mathrm{val}}^{(i)}$ are the training and validation sets on node $i$, respectively.

In Figure 11, all methods are evaluated using 8 agents under various communication topologies. For the ring topology, the weight matrix $W = (w_{ij})$ is doubly stochastic and defined as

$$w_{i,i} = a, \quad w_{i,i\pm 1} = \tfrac{1-a}{2}, \quad \text{with } a = 0.4,$$

and all other entries are set to zero.

All experiments are conducted on a machine equipped with an Intel(R) Xeon(R) Gold 5218R CPU @ 2.10GHz. The step sizes are configured as follows. For SUN-DSBO-GT/SE (with decaying coefficient $\mu_k := (1 + k)^{-0.001}$ and $1/\gamma = 0.02$), the step sizes are set to $[0.05, 0.1, 0.1]$.

**Settings with Dynamic Topology.** The hyperparameter settings are the same as those in Figure 4 (b) and (c), corresponding to the heterogeneity parameters $\mathcal{H}$ for each case.

## C.5 META-LEARNING

In the context of decentralized meta-learning, the problem can be formulated as a decentralized stochastic bilevel optimization (SBO) problem (Zhu et al., 2024). We assume that the data for each task is distributed across $N$ nodes. Each node $i$ (for $i = 1, 2, \ldots, N$) holds a local training dataset $D_i^{\text{Train}}$ and a local validation dataset $D_i^{\text{Val}}$, which are subsets of the full task-specific datasets.

There are $R$ tasks $\{T_s\}_{s=1}^R$, where each task $T_s$ has its corresponding loss function $L(x, y_s, \xi)$, with $\xi$ being a stochastic sample drawn from the data distribution $D_s$. In this context, $y_s$ represents task-specific parameters, while $x$ represents global parameters shared by all tasks.

The decentralized formulation seeks to optimize the global parameters $x$ by coordinating the task-specific parameters $y_1, y_2, \ldots, y_R$ across all nodes while minimizing the following objective function:

$$f_i(x, y) = \frac{1}{R} \sum_{s=1}^R \mathbb{E}_{\xi \sim D_{i,s}^{\text{Train}}} [L(x, y_s, \xi)] + R(y_s)$$

where $R(y_s)$ represents a regularization term applied to the task-specific parameters $y_s$, such as $R(y_s) = C_r \|y_s\|^2$, ensuring that the task parameters remain small or smooth.

Each node $i$ aims to compute the task-specific parameters $y_s$ that minimize the expected loss over the local training data, while respecting the regularization:

$$g_i(x, y) = \frac{1}{R} \sum_{s=1}^R \mathbb{E}_{\xi \sim D_{i,s}^{\text{Train}}} [L(x, y_s, \xi)] + R(y_s)$$

The goal of meta-learning is to find the global parameters $x$ that minimize the average loss across all tasks. The overall objective can be written as:

$$\min_{x, y_1, \ldots, y_R} \frac{1}{R} \sum_{s=1}^R \mathbb{E}_{\xi \sim D_s} [L(x, y_s, \xi)]$$

where $L(x, y_s, \xi)$ represents the loss function for the task $T_s$ with given global parameters $x$ and task-specific parameters $y_s$, evaluated on the sample $\xi$ drawn from the distribution $D_s$.

This decentralized meta-learning framework ensures that each node learns the task-specific parameters collaboratively while maintaining local privacy constraints.

All methods are evaluated with 10 agents for Omniglot and 5 agents arranged in a ring topology. The batch size (meta-batch) is set to 16. The mixing matrix $W$ is doubly stochastic, with self, left, and right weights defined as:

$$w_{i,i} = w_{i,i\pm1} = \frac{1}{3}, \quad \text{and all other entries are set to 0.}$$

Tasks are 5-way 5-shot, with 20k, 200, and 200 tasks sampled for training, validation, and testing, respectively. The evaluation is capped at 1,000 tasks. Unless otherwise stated, the data is uniformly split (Dirichlet heterogeneity is off by default).

The models use a 4-layer CNN backbone consisting of 3×3 convolutions, batch normalization (BN), ReLU activation, and 2×2 pooling per layer. The channel width is 64 for Omniglot, and 32 for MiniImageNet, followed by a task-specific linear head.

The following are the hyperparameter settings:

For Omniglot: SPARKLE-GT/EXTRA, SLDBO D-SOBA: step sizes = [0.01, 0.1, 0.1]; SUN-DSBO-SE/GT: decaying coefficient $\mu_k := 4(1 + k)^{-0.01}$, $\gamma = 6.0$, and step sizes = [0.02, 0.1, 0.1].

For MiniImageNet: SPARKLE-GT/EXTRA, SLDBO D-SOBA: step sizes = [0.01, 0.05, 0.05]; SUN-DSBO-SE/GT: decaying coefficient $\mu_k := (1 + k)^{-0.01}$, $\gamma = 1.0$, and step sizes = [0.01, 0.15, 0.15].

The momentum for task-specific SGD is set to 0.85, with a regularization term of 0.001 for MiniImageNet and Omniglot.

# D  ADDITIONAL THEORETICAL RESULTS

## D.1  THE RELATIONSHIP BETWEEN PROBLEM (1) AND PROBLEM (4)

To clearly demonstrate the effectiveness of solving the reconstruction problem (4) in this work, we first describe the relationship between problem (1) and problem (2), and subsequently clarify the connection between problem (2) and problem (4).

**Relationship between problem (1) and problem (2).** Since it is difficult to work directly with implicit gradients for problem (1) when the lower-level problem is nonconvex, researchers often consider the following equivalent value function-based reformulation of problem (1):

$$\min_{x,y} F(x,y) \quad \text{s.t.} \quad G(x,y) - V(x) \leq 0, \quad \text{where} \quad V(x) = \min_y G(x,y), \tag{12}$$

where $V(x)$ is referred to as the value function. However, when the lower-level problem is nonconvex, $V(x)$ becomes *nonsmooth*, and conventional constraint qualifications fail to hold, see, e.g., Ye & Zhu (1995, Proposition 3.2). As a result, the classical Karush–Kuhn–Tucker (KKT) condition of problem (12) are overly complex—due to the computation of the subgradient of $V(x)$—and are no longer suitable as an effective metric for problem (1).

To address the nonsmooth issue, we apply a *smooth relaxation* using the Moreau envelope, leading to problem (2):

$$\min_{(x,y)} F(x,y) \quad \text{s.t.} \quad G(x,y) - V_\gamma(x,y) \leq 0,$$

where

$$V_\gamma(x,y) := \min_{\theta \in \mathbb{R}^{d_y}} \left\{ G(x,\theta) + \frac{1}{2\gamma} \|\theta - y\|^2 \right\}, \quad \gamma > 0,$$

is the Moreau envelope of $G$. Notably, $V_\gamma(x,y)$ is smooth and satisfies $G(x,y) - V_\gamma(x,y) \geq 0$. Moreover, as discussed in item (i) in Subsection 2.1, under mild conditions, problem (2) is equivalent to the relaxed version of problem (1):

$$\min_{x,y} F(x,y) \quad \text{s.t.} \quad \nabla_y G(x,y) = 0. \tag{13}$$

It is worth noting that problem (13) is independent of $\gamma$ and *remains equivalent to problem (1)* when the lower-level objective satisfies either a convexity condition or the Polyak–Łojasiewicz (PL) condition. Therefore, it is reasonable to consider problem (2) as a surrogate for problem (1).

**Connection between problem (2) and problem (4).** Although problem (2) is smooth, conventional constraint qualifications (e.g., MFCQ) do not hold for its inequality constraint $G(x,y) - V_\gamma(x,y) \leq 0$. Hence, we apply the *penalty method*, which is widely used in such problems; see, e.g., the penalty approaches for nonconvex bilevel optimization discussed in Kwon et al. (2024). Thanks to the property $G(x,y) - V_\gamma(x,y) \geq 0$, the penalty formulation can be expressed as problem (4):

$$\min_{(x,y) \in \mathbb{R}^{d_x} \times \mathbb{R}^{d_y}} \Psi_\mu(x,y) := \mu F(x,y) + G(x,y) - V_\gamma(x,y).$$

Depending on whether the exact penalty property holds, the penalty parameter $\mu$ is chosen to be either sufficiently small or gradually diminishing. In particular, if an *error bound condition holds* for problem (2), then $\mu$ can be fixed as a sufficiently small constant. In this case, problem 4 is equivalent to problem 2. As noted, the theoretical analysis and results remain valid for fixed $\mu$, i.e., when $p = 0$. On the other hand, if the exact penalty property does not hold for problem (2), then $\mu$ should gradually diminish to ensure that problem (4) serves as an effective approximation of problem (2).

Next, we discuss the relationship between the metric in problem (1) and problem (4).

*Remark* D.1. We have discussed the relationship between problem (1) and problem (2) in the previous discussion, so the connection between the metric in problem (1) and problem (2) follows analogously. If an *error bound condition* holds for problem (2), $\mu_k$ can be fixed as a sufficiently small constant. In this case, the stationarity measure $\nabla \Psi_{\mu_k}(x,y)$ with $\mu_k$ being a sufficiently small constant corresponds to the classical Karush–Kuhn–Tucker (KKT) condition of problem (2). Otherwise, if the exact penalty property does not hold, $\mu_k$ should gradually diminish so that problem (4) approximates problem (2); then, $\nabla \Psi_{\mu_k}(x,y)$ with a varying sequence $\mu_k$ corresponds to an approximate KKT condition. In both scenarios, $\nabla \Psi_{\mu_k}(x,y)$ effectively measures optimality in problem (2).

## D.2    ADDITIONAL CONVERGENCE RATE RESULTS

**Results with $\mu_k$ including $p = 0$.**    When the penalty parameter $\mu_k$ is fixed at a constant value $\mu_0$, setting $p = 0$ in Theorems 3.5 and 3.7 leads to the corresponding convergence rate result for the approximation problem (5), where $\mu_k = \mu_0$, derived from its stationarity measure.

**Proposition D.2.** *By replacing $p \in (0, 1/4)$ in Theorem 3.5 with $p \in [0, 1/4)$, we obtain the following result:*

$$\frac{1}{K} \sum_{k=0}^{K-1} \mathbb{E}\|\nabla \Psi_{\mu_k}(\bar{x}^k, \bar{y}^k)\|^2 = \mathcal{O}\left( \frac{\mathcal{C}_1}{n^{1/2} K^{1/2}} \right) + \mathcal{O}\left( \frac{n(\delta_f^2 + \delta_g^2)}{(1-\rho)^2 K} \right) + \mathcal{O}\left( \frac{n(\sigma_f^2 + \sigma_g^2)}{(1-\rho)^2 K} \right),$$

*where $\mathcal{C}_1$ is a positive constant depending on $\delta_f^2$, $\delta_g^2$, $\sigma_f^2$, and $\sigma_g^2$, but independent of $n$ and $(1-\rho)$.*

**Proposition D.3.** *By replacing $p \in (0, 1/4)$ in Theorem 3.7 with $p \in [0, 1/4)$, we obtain the following result:*

$$\frac{1}{K} \sum_{k=0}^{K-1} \mathbb{E}\|\nabla \Psi_{\mu_k}(\bar{x}^k, \bar{y}^k)\|^2 = \mathcal{O}\left( \frac{\mathcal{C}_2}{n^{1/2} K^{1/2}} \right) + \mathcal{O}\left( \frac{n(\delta_f^2 + \delta_g^2)}{(1-\rho)^4 K} \right),$$

*where $\mathcal{C}_2$ is a positive constant depending on $\delta_f^2$, and $\delta_g^2$, but independent of $n$ and $(1-\rho)$.*

The proofs of Propositions D.2 and D.3 can be found in Theorems F.11 and G.8.

**Results illustrating the effectiveness of the stationarity measure $\nabla \Psi_{\mu_k}$.**    Based on Propositions D.2 and D.3, we further analyze the convergence behavior of SUN-DSBO-SE and SUN-DSBO-GT, illustrating the effectiveness of our results with $\nabla \Psi_{\mu_k}$:

**Corollary D.4.** *Under the conditions of Proposition D.2, for SUN-DSBO-SE, it holds that*

$$\min_{\lfloor (K-1)/2 \rfloor \leq k \leq K-1} \mathbb{E}\|\nabla \Psi_{\mu_k}(\bar{x}^k, \bar{y}^k)\|^2 = \mathcal{O}\left( \frac{\mathcal{C}_1}{n^{1/2} K^{1/2}} \right) + \mathcal{O}\left( \frac{n(\delta_f^2 + \delta_g^2)}{(1-\rho)^2 K} \right) + \mathcal{O}\left( \frac{n(\sigma_f^2 + \sigma_g^2)}{(1-\rho)^2 K} \right),$$

*where $\mathcal{C}_1$ is a positive constant depending on $\delta_f^2$, $\delta_g^2$, $\sigma_f^2$, and $\sigma_g^2$, but independent of $n$ and $(1-\rho)$. Here, $\lfloor (K-1)/2 \rfloor$ denotes the floor function of $(K-1)/2$.*

*Proof.* From Proposition D.2, we know that:

$$\frac{1}{K} \sum_{k=0}^{K-1} \mathbb{E}\|\nabla \Psi_{\mu_k}(\bar{x}^k, \bar{y}^k)\|^2 = \mathcal{O}\left( \frac{\mathcal{C}_1}{n^{1/2} K^{1/2}} \right) + \mathcal{O}\left( \frac{n(\delta_f^2 + \delta_g^2)}{(1-\rho)^2 K} \right) + \mathcal{O}\left( \frac{n(\sigma_f^2 + \sigma_g^2)}{(1-\rho)^2 K} \right).$$

From here, we can derive the following bounds:

$$\frac{1}{K} \sum_{k=\lfloor (K-1)/2 \rfloor}^{K-1} \mathbb{E}\|\nabla \Psi_{\mu_k}(\bar{x}^k, \bar{y}^k)\|^2 = \mathcal{O}\left( \frac{\mathcal{C}_1}{n^{1/2} K^{1/2}} \right) + \mathcal{O}\left( \frac{n(\delta_f^2 + \delta_g^2)}{(1-\rho)^2 K} \right) + \mathcal{O}\left( \frac{n(\sigma_f^2 + \sigma_g^2)}{(1-\rho)^2 K} \right).$$

We can then establish the following relationship:

$$\min_{\lfloor (K-1)/2 \rfloor \leq k \leq K-1} \mathbb{E}\|\nabla \Psi_{\mu_k}(\bar{x}^k, \bar{y}^k)\|^2 \leq \frac{1}{K - \lfloor (K-1)/2 \rfloor} \sum_{k=\lfloor (K-1)/2 \rfloor}^{K-1} \mathbb{E}\|\nabla \Psi_{\mu_k}(\bar{x}^k, \bar{y}^k)\|^2$$

$$\leq \frac{K}{K - \lfloor (K-1)/2 \rfloor} \frac{1}{K} \sum_{k=\lfloor (K-1)/2 \rfloor}^{K-1} \mathbb{E}\|\nabla \Psi_{\mu_k}(\bar{x}^k, \bar{y}^k)\|^2$$

$$\leq 2 \frac{1}{K} \sum_{k=\lfloor (K-1)/2 \rfloor}^{K-1} \mathbb{E}\|\nabla \Psi_{\mu_k}(\bar{x}^k, \bar{y}^k)\|^2.$$

Then we can obtain the above conclusion.    $\square$

**Corollary D.5.** *Under the conditions of Proposition D.3, for SUN-DSBO-GT, it holds that*

$$\min_{\lfloor (K-1)/2 \rfloor \le k \le K-1} \mathbb{E}\big\| \nabla \Psi_{\mu_k}(\bar{x}^k, \bar{y}^k) \big\|^2 = \mathcal{O}\Big( \frac{\mathcal{C}_2}{n^{1/2} K^{1/2}} \Big) + \mathcal{O}\Big( \frac{n(\delta_f^2 + \delta_g^2)}{(1-\rho)^4 K} \Big),$$

*where $\mathcal{C}_2$ is a positive constant depending on $\delta_f^2$ and $\delta_g^2$, but independent of $n$ and $(1-\rho)$.*

This proof is similar to the proof of Proposition D.2.

By definition, if

$$\min_{\lfloor (K-1)/2 \rfloor \le k \le K-1} \mathbb{E}\big\| \nabla \Psi_{\mu_k}(\bar{x}^k, \bar{y}^k) \big\|^2 \le \epsilon,$$

then there exists some $\hat{k} \in [\lfloor (K-1)/2 \rfloor, K-1]$ such that

$$\mathbb{E}\big\| \nabla \Psi_{\mu_{\hat{k}}}(\bar{x}^{\hat{k}}, \bar{y}^{\hat{k}}) \big\|^2 \le \epsilon.$$

When $K$ is sufficiently large, $\hat{k}$ is also large, ensuring that $\mu_{\hat{k}} \to 0$. Consequently, $\nabla \Psi_{\mu_{\hat{k}}}(\bar{x}^{\hat{k}}, \bar{y}^{\hat{k}})$ approximates a stationary point of the problem (13).

# E  PROOF SKETCH

In this section, we present the corresponding challenges and provide a macroscopic overview of the solution approaches, summarizing the relevant analytical challenges. We then outline the key steps and subsequently discuss the non-trivial analytical aspects.

## E.1  MAIN CHALLENGES

Below, we elaborate on the specific challenges encountered in our study:

*Compared to single-agent bilevel optimization with nonconvex lower-level objectives, the main difference lies in the need to address issues introduced by our decentralized algorithm, such as consensus constraints and data heterogeneity.* To resolve the consensus constraint, we adopt a network-consensus framework and propose two single-loop algorithms which maintain communication efficiency. Additionally, we incorporate a heterogeneity correction technique, gradient tracking, to mitigate the effects of data heterogeneity. Notably, compared to methods in Liu et al. (2024) that rely on Moreau envelope reformulation, our approach also accounts for stochasticity, which requires analyzing the impact of stochastic errors and the complexity introduced by this aspect.

*Compared to existing global DSBO approaches in Table 1, our assumptions are weaker, primarily in two aspects.* On one hand, we do not assume that the gradient norm of the upper-level objective function is bounded, which necessitates additional steps for gradient estimation. To estimate the gradient, we need to introduce the gradient similarity assumption (Assumption 3.4) or the heterogeneity correction technique (GT). Moreover, by incorporating GT, we can eliminate the gradient similarity assumption for both the upper and lower-level objective functions. Consequently, further analysis is needed to address the resulting complexities in the convergence analysis. On the other hand, we assume that the lower-level objective is nonconvex, which means we cannot guarantee the uniqueness of the lower-level solution. As a result, we cannot use algorithms based on hyper-gradients, as hyper-gradients are not well-defined in this case, nor can we directly use value function-based algorithms, since the smoothness of the value functions relies on the uniqueness of the lower-level solution (Wang et al., 2024). Similar to existing decentralized or centralized bilevel algorithms for non-strongly convex or nonconvex LL objectives, We adopt a feasible relaxation-based approach.

In decentralized bilevel optimization, no prior work, to the best of our knowledge, has explored DSBO with nonconvex lower-level objectives. Recently, Qin et al. (2025) has studied DSBO with non-strongly convex lower-level objectives, introducing gradually diminishing quadratic regularization to address the non-uniqueness challenge of the lower-level solution. However, it assumes the lower-level objective is convex in the personalized DSBO setting, where "personalized" refers to the absence of consensus constraints on the lower-level variables. This approach cannot be directly applied to our problem (1) due to the lack of a uniqueness guarantee for the lower-level solution caused by nonconvexity when gradually diminishing quadratic regularization is introduced. Additionally,

compared to most personalized DSBO works, the challenge in global DSBO lies in the inability of single-agent bilevel optimization methods to adapt to the distributed setting, when relaxed to lower-level strong convexity, as discussed in Chen et al. (2023, Section 1.2). To address the challenge of non-uniqueness of the lower-level solution, we employ a Moreau envelope-based penalty method to smooth and relax the original problem (1) into problem 4. The relationship between problem (1) and problem (4) is discussed in Appendix D.1. Furthermore, in the subsequent steps, decentralized coordination for the lower-level variables is required, and feedback is incorporated into the upper-level network-consensus process.

From the above discussion, we can conclude that the main objective of our analysis is to conduct convergence analysis under weaker assumptions—particularly without bounded gradient assumptions or even bounded gradient similarity assumptions for the upper-level objective function—while considering the consensus error and heterogeneity effects introduced by applying Moreau envelope-based penalty to DSBO with nonconvex lower-level objectives, coupled with the impact of stochastic errors. In the next subsection, we focus on the key steps of the proof, which are accompanied by the challenges discussed earlier.

### E.2 KEY STEPS

In Section 3, we establish the theoretical convergence guarantees for the proposed algorithms through Theorems 3.5 and 3.7. This section outlines the key ideas underlying the convergence analysis of SUN-DSBO-SE and SUN-DSBO-GT . The analysis follows two main steps: (i) deriving an upper bound on the residual function, and (ii) bounding the corresponding terms using a Lyapunov-based argument. Together, these steps yield rigorous convergence guarantees under appropriately chosen step sizes. While the sketches follow a unified structure, the steps highlighted in yellow correspond to SUN-DSBO-SE , and those in green correspond to SUN-DSBO-GT .

**Step 1. Bound the residual $\|\nabla\Psi_{\mu_k}(\bar{x}^k,\bar{y}^k)\|^2$ in terms of the step sizes.**
By exploiting Assumption 3.1, the $L$-smoothness of $\Phi_{\mu_k}$ (Lemma F.3), and choosing step sizes as $\lambda_x^k = \lambda_y^k = c_\lambda \lambda_\theta^k$ for some $c_\lambda > 0$, we obtain:

$$\mathbb{E}\|\nabla\Psi_{\mu_k}(\bar{x}^k,\bar{y}^k)\|^2 \le \frac{1}{\lambda_\theta^k} R_s^k, \tag{14}$$

where

$$R_s^k := \mathcal{O}\left(\frac{1}{\lambda_\theta^k}\right)\|\mathbb{E}[\bar{x}^{k+1}-\bar{x}^k]\|^2 + \mathcal{O}\left(\frac{1}{\lambda_\theta^k}\right)\|\mathbb{E}[\bar{y}^{k+1}-\bar{y}^k]\|^2$$
$$+ \mathcal{O}(\lambda_\theta^k)\mathbb{E}\|\bar{\theta}^k - \theta_\gamma^*(\bar{x}^k,\bar{y}^k)\|^2 + \mathcal{O}(\lambda_\theta^k)\Delta^k.$$

Here, $\theta_\gamma^*(x,y)$ denotes the unique solution to problem (3), and $\Delta^k$ captures the cumulative consensus error over $x$, $y$, and $\theta$, as defined in Section 3. This upper bound reveals three main sources of residual error: (i) the drift between successive averaged iterates $(\bar{x}^k,\bar{y}^k)$, (ii) the mismatch between $\bar{\theta}^k$ and the smoothed lower-level solution $\theta_\gamma^*(\bar{x}^k,\bar{y}^k)$, and (iii) the global consensus error $\Delta^k$, which distinguishes it from the single-agent setting. Under properly chosen step sizes, $R_s^k$ remains bounded, thereby ensuring convergence of the residual.

**Step 2. Control $R_s^k$ using a Lyapunov descent argument.**
To establish the descent of $R_s^k$ in (14), we introduce the following Lyapunov functions.

$$\mathcal{L}_{\text{se}}^k := \mathbb{E}\left[\Phi_{\mu_k}(\bar{x}^k,\bar{y}^k)\right] + a_1\mathbb{E}\|\bar{\theta}^k - \theta_\gamma^*(\bar{x}^k,\bar{y}^k)\|^2$$
$$+ \frac{a_2^k}{n}\sum_{i=1}^n \mathbb{E}\|x_i^k - \bar{x}^k\|^2 + \frac{a_3^k}{n}\sum_{i=1}^n \mathbb{E}\|y_i^k - \bar{y}^k\|^2 + \frac{a_4^k}{n}\sum_{i=1}^n \mathbb{E}\|\theta_i^k - \bar{\theta}^k\|^2, \tag{15}$$

where

$$a_1 := \frac{1}{2L_\theta}\sqrt{L_2^2 + \frac{1}{\gamma^2}}, \quad a_2^k := \tau_2 c_\lambda \lambda_\theta^k, \quad a_3^k := \tau_3 c_\lambda \lambda_\theta^k, \quad a_4^k := \tau_4 \lambda_\theta^k,$$

with $\tau_2$–$\tau_4$ from (51) ensuring the descent of $\mathcal{L}_{\text{se}}^k$, and $L_\theta$ a constant from Lemma F.2.

$$\mathcal{L}_{\text{gt}}^k := \mathbb{E}\big[\Phi_{\mu_k}(\bar{x}^k, \bar{y}^k)\big] + a_1 \mathbb{E}\|\bar{\theta}^k - \theta_\gamma^*(\bar{x}^k, \bar{y}^k)\|^2$$

$$+ \frac{a_2^k}{n}\sum_{i=1}^n \mathbb{E}\|x_i^k - \bar{x}^k\|^2 + \frac{a_3^k}{n}\sum_{i=1}^n \mathbb{E}\|y_i^k - \bar{y}^k\|^2 + \frac{a_4^k}{n}\sum_{i=1}^n \mathbb{E}\|\theta_i^k - \bar{\theta}^k\|^2 \tag{16}$$

$$+ \frac{a_5^k}{n}\sum_{i=1}^n \mathbb{E}\|t_{x,i}^k - \bar{t}_x^k\|^2 + \frac{a_6^k}{n}\sum_{i=1}^n \mathbb{E}\|t_{y,i}^k - \bar{t}_y^k\|^2 + \frac{a_7^k}{n}\sum_{i=1}^n \mathbb{E}\|t_{\theta,i}^k - \bar{t}_\theta^k\|^2,$$

where $t_{\diamond,i}^k$ and $\bar{t}_\diamond^k$ for $\diamond \in \{x, y, \theta\}$ are auxiliary gradient tracking variables in (60), and the coefficients are given by:

$$a_1 = \frac{1}{2L_\theta}\sqrt{L_2^2 + \frac{1}{\gamma^2}}, \quad a_2^k = \tau_2 c_\lambda \lambda_\theta^k, \quad a_3^k = \tau_3 c_\lambda \lambda_\theta^k,$$

$$a_4^k = \tau_4 c_\lambda \lambda_\theta^k, \quad a_5^k = \tau_5 c_\lambda^2 (\lambda_\theta^k)^2, \quad a_6^k = \tau_6 c_\lambda^2 (\lambda_\theta^k)^2, \quad a_7^k = \tau_7 c_\lambda^2 (\lambda_\theta^k)^2,$$

with $\tau_2$–$\tau_7$ ensuring the descent of $\mathcal{L}_{\text{gt}}^k$ and $L_\theta$ is some positive constant defined in Lemma F.2.

Here, $\Phi_{\mu_k}(x, y) := \mu_k(F(x,y) - \underline{F}) + G(x,y) - V_\gamma(x,y) \geq 0$, and satisfies $\nabla\Psi_{\mu_k}(x,y) = \nabla\Phi_{\mu_k}(x,y)$. The descent of $R_s^k$ follows under a step size regime that is jointly tailored with the penalty decay schedule $\{\mu_k\}$, in order to accommodate both the hierarchical structure and the approximation behavior induced by the vanishing penalty.

$$R_s^k \leq \mathcal{L}_{\text{se}}^k - \mathcal{L}_{\text{se}}^{k+1} + \epsilon_{\text{sto,se}}^k + \epsilon_{\text{dh}}^k, \tag{17}$$

where $\epsilon_{\text{sto,se}}^k$ and $\epsilon_{\text{dh}}^k$ are defined in (54), representing the errors due to stochastic estimation and data heterogeneity, respectively. More details can be found in Lemma F.9.

$$R_s^k \leq \mathcal{L}_{\text{gt}}^k - \mathcal{L}_{\text{gt}}^{k+1} + \epsilon_{\text{sto,gt}}^k, \tag{18}$$

where $\epsilon_{\text{sto,gt}}^k$ is defined in (83), representing the error due to stochastic estimation. More details can be found in Lemma G.6.

These inequalities characterize the descent of $R_s^k$ through Lyapunov difference terms and additive error components, where the contributions of stochastic noise and gradient dissimilarity (as introduced in Assumption 3.4) are explicitly maintained to preserve analytical clarity, and will be used to establish the convergence bounds that follow.

Substituting (17) into (14), summing from $k = 0$ to $K - 1$, and utilizing the monotonicity of $\mu_k$ and non-negativity of $\Phi_{\mu_k}(x, y)$, we obtain:

$$\sum_{k=0}^{K-1} \lambda_\theta^k \mathbb{E}\|\nabla\Psi_{\mu_k}(\bar{x}^k, \bar{y}^k)\|^2 \leq \mathcal{O}\left(\sum_{k=0}^{K-1} \frac{(\lambda_\theta^k)^2}{n}(\delta_f^2 + \delta_g^2)\right) + \mathcal{O}\left(\sum_{k=0}^{K-1} \frac{(\lambda_\theta^k)^3}{(1-\rho)^2}(\delta_f^2 + \delta_g^2)\right)$$

$$+ \mathcal{O}\left(\sum_{k=0}^{K-1} \frac{(\lambda_\theta^k)^3}{(1-\rho)^2}(\sigma_f^2 + \sigma_g^2)\right) + \mathcal{O}(\mathcal{L}_{\text{se}}^0). \tag{19}$$

As detailed in (58), the second and third terms in (19) correspond to the accumulated stochastic noise $\sum_k \epsilon_{\text{sto}}^k$, while the last term reflects the cumulative effect of data heterogeneity $\sum_k \epsilon_{\text{dh}}^k$, as defined in (17). By selecting step sizes according to Theorem 3.5, the convergence of $\mathbb{E}\|\nabla\Psi_{\mu_k}(\bar{x}^k, \bar{y}^k)\|^2$ is ensured, completing the proof of Theorem 3.5.

Substituting (18) into (14) and summing over $k$, we have:

$$\sum_{k=0}^{K-1} \lambda_\theta^k \, \mathbb{E}\|\nabla \Psi_{\mu_k}(\bar{x}^k, \bar{y}^k)\|^2 \leq \mathcal{O}\left(\sum_{k=0}^{K-1} \frac{(\lambda_\theta^k)^2}{n}(\delta_f^2 + \delta_g^2)\right) + \mathcal{O}\left(\sum_{k=0}^{K-1} \frac{(\lambda_\theta^k)^3}{(1-\rho)^4}(\delta_f^2 + \delta_g^2)\right)$$
$$+ \mathcal{O}(\mathcal{L}_{\mathrm{gt}}^0). \tag{20}$$

The proof follows a similar line of reasoning as the derivation leading to Equation (19). The second term in (20) captures the accumulated stochastic error $\sum_k \epsilon_{\mathrm{sto}}^k$, while the third term quantifies the residual error determined by the initial Lyapunov gap $\mathcal{L}_{\mathrm{gt}}^0$, as defined in (18). With constant step sizes specified in Theorem 3.7, convergence is guaranteed, thereby completing the proof of Theorem 3.7.

Next, we elaborate on the *non-trivial aspects* of our analysis, which distinguish it from previous works.

- *The special gradient form with dynamic $\mu_k$ under stochastic and heterogeneous settings.* If we choose $\mu_k$ as a sufficiently small constant, it may result in slower convergence, as discussed in Kwon et al. (2023; 2024) for centralized bilevel optimization. Gradually decreasing the penalty parameters $\{\mu_k\}$, with $\mu_k \to 0$ as the iteration index $k$ increases, is a better choice. Varying $\mu_k$ leads to a model that evolves with the iteration index $k$, introducing unique challenges under weaker conditions, particularly in gradient estimation and heterogeneity analysis. The structure of our problem yields a *special gradient form* (see Eq. (6)), which may involve stochastic estimates with a *time-varying $\mu_k$*, and may even incorporate heterogeneity correction techniques without assuming bounded upper-level gradient norms. These factors complicate gradient estimation and heterogeneity analysis. On one hand, certain coefficients become dynamic, such as the heterogeneity levels $\sigma_f$ and $\sigma_g$ in Assumption 3.4 and the stochastic error, requiring additional handling (e.g., step-size restrictions). On the other hand, when heterogeneity correction techniques are incorporated, analyzing GT methods under dynamic gradients with stochastic error becomes necessary, which is not conventional.

- *Design of Lyapunov Functions*: The iterative structure of the proposed stochastic bilevel optimization algorithm requires the construction of Lyapunov functions with several undetermined coefficients. These functions must simultaneously capture upper-level optimality, lower-level error, consensus error, and auxiliary error, making their design analytically challenging.

  - For *SUN-DSBO-SE* (without gradient tracking), the analysis relies on the assumption of *bounded gradient similarity*, which is weaker than the commonly used bounded gradient norm assumption. The main difficulty lies in controlling error accumulation and variance throughout the iterations. This requires carefully combining upper-level optimality, lower-level error, consensus error, and auxiliary error terms with appropriately chosen coefficients to establish convergence.
  - For *SUN-DSBO-GT* (with gradient tracking), the analysis no longer depends on the bounded gradient similarity assumption. Instead, the primary challenge is to handle the coupling between *local variable bias* and *gradient tracking errors*. This demands a precise selection of Lyapunov terms to manage these interactions and ensure system stability.

  The corresponding form of the Lyapunov function can be found in Step 2 of the next section.

- *Step-size (learning rate) selection.* Selecting appropriate step-sizes is crucial for ensuring the convergence of our algorithm. First, we must control the step-size to ensure convergence while mitigating the effects of stochastic errors and heterogeneity. Second, this task becomes particularly challenging due to the complex interactions between the error term coefficients and the time-varying $\mu_k$, including stochastic and consensus errors. Therefore, we need to carefully consider the range of the step-size to accommodate the dynamic nature of $\mu_k$.

# F    PROOFS OF SUN-DSBO-SE

We first introduce some notations that will be used throughout the analysis.

## F.1    NOTATIONS

We recall the following averaged variables across agents:

$$\bar{x}^k := \frac{1}{n}\sum_{i=1}^{n} x_i^k, \quad \bar{y}^k := \frac{1}{n}\sum_{i=1}^{n} y_i^k, \quad \bar{\theta}^k := \frac{1}{n}\sum_{i=1}^{n} \theta_i^k, \tag{21}$$

$$\bar{d}_x^k := \frac{1}{n}\sum_{i=1}^{n} d_{x,i,k}^{(\zeta_i^k)}, \quad \bar{d}_y^k := \frac{1}{n}\sum_{i=1}^{n} d_{y,i,k}^{(\zeta_i^k)}, \quad \bar{d}_\theta^k := \frac{1}{n}\sum_{i=1}^{n} d_{\theta,i,k}^{(\zeta_i^k)}, \tag{22}$$

where $d_{\theta,i,k}^{(\zeta_i^k)} = \hat{D}_{\theta,i}^k, d_{x,i,k}^{(\zeta_i^k)} = \hat{D}_{x,i}^k$, and $d_{y,i,k}^{(\zeta_i^k)} = \hat{D}_{y,i}^k$, with $\zeta_i^k = [\xi_{f,i}^k, \xi_{g,i}^k]$ denoting the stochastic sample used in Algorithm 1. The explicit expressions of the gradient estimators are given as:

$$d_{\theta,i,k}^{(\zeta_i^k)} = \nabla_y g_i(x_i^k, \theta_i^k; \xi_{g,i}^k) + \frac{1}{\gamma}(\theta_i^k - y_i^k),$$

$$d_{x,i,k}^{(\zeta_i^k)} = \mu_k \nabla_x f_i(x_i^k, y_i^k; \xi_{f,i}^k) + \nabla_x g_i(x_i^k, y_i^k; \xi_{g,i}^k) - \nabla_x g_i(x_i^k, \theta_i^k; \xi_{g,i}^k), \tag{23}$$

$$d_{y,i,k}^{(\zeta_i^k)} = \mu_k \nabla_y f_i(x_i^k, y_i^k; \xi_{f,i}^k) + \nabla_y g_i(x_i^k, y_i^k; \xi_{g,i}^k) - \frac{1}{\gamma}(y_i^k - \theta_i^k).$$

We denote the corresponding expectations as

$$\tilde{d}_{\theta,i}^k := \mathbb{E}[d_{\theta,i,k}^{(\zeta_i^k)}], \quad \tilde{d}_{x,i}^k := \mathbb{E}[d_{x,i,k}^{(\zeta_i^k)}], \quad \tilde{d}_{y,i}^k := \mathbb{E}[d_{y,i,k}^{(\zeta_i^k)}],$$

where the expectation is taken with respect to the sampling $\zeta_i^k$.

We further define the averaged expected gradients across all nodes:

$$\tilde{\bar{d}}_\theta^k := \frac{1}{n}\sum_{i=1}^{n} \tilde{d}_{\theta,i}^k, \quad \tilde{\bar{d}}_x^k := \frac{1}{n}\sum_{i=1}^{n} \tilde{d}_{x,i}^k, \quad \tilde{\bar{d}}_y^k := \frac{1}{n}\sum_{i=1}^{n} \tilde{d}_{y,i}^k. \tag{24}$$

Note that the averaged iterates evolve as

$$\bar{x}^{k+1} = \bar{x}^k - \lambda_x^k \bar{d}_x^k, \quad \bar{y}^{k+1} = \bar{y}^k - \lambda_y^k \bar{d}_y^k, \quad \bar{\theta}^{k+1} = \bar{\theta}^k - \lambda_\theta^k \bar{d}_\theta^k. \tag{25}$$

To quantify the consensus error among agents, we define

$$\Delta_x^k := \frac{1}{n}\sum_{i=1}^{n} \mathbb{E}\|x_i^k - \bar{x}^k\|^2, \quad \Delta_y^k := \frac{1}{n}\sum_{i=1}^{n} \mathbb{E}\|y_i^k - \bar{y}^k\|^2, \quad \Delta_\theta^k := \frac{1}{n}\sum_{i=1}^{n} \mathbb{E}\|\theta_i^k - \bar{\theta}^k\|^2. \tag{26}$$

We further define the following auxiliary function at each node $i$, which will be used in our analysis of the lower-level updates:

$$\phi_{\mu_k}^i(x,y)[\theta] := \mu_k f_i(x,y) + g_i(x,y) - g_i(x,\theta) - \frac{1}{2\gamma}\|\theta - y\|^2. \tag{27}$$

To establish the convergence results, we first illustrate the decreasing property of the auxiliary function defined as:

$$\min_{(x,y)\in\mathbb{R}^{d_x}\times\mathbb{R}^{d_y}} \Phi_{\mu_k}(x,y) := \mu_k\Big(F(x,y) - \underline{F}\Big) + G(x,y) - V_\gamma(x,y). \tag{28}$$

Finally, to formalize the randomness introduced by sampling, we define the natural filtration associated with the algorithm as

$$\mathcal{F}_0 := \{\Omega, \varnothing\}, \quad \mathcal{F}_k := \sigma\left(\{\zeta_i^0, \zeta_i^1, \dots, \zeta_i^{k-1} : i \in \mathcal{G}\}\right), \quad \forall k \geq 1,$$

where $\varnothing$ denotes the empty set.

### F.2 PRELIMINARY LEMMAS

Define $\theta_\gamma^*(x, y)$ is the unique solution to problem (3), owing to the strong convexity of its objective function.

**Lemma F.1** (Properties of Moreau envelope(Liu et al., 2024)). *Suppose that $g_i(x, y)$ is $L_2$-smooth on $\mathbb{R}^{d_x} \times \mathbb{R}^{d_y}$. Then for $\gamma \in (0, \frac{1}{2L_2})$, $\rho_{v_1} \geq L_2$ and $\rho_{v_2} \geq \frac{1}{\gamma}$, the function $V_\gamma(x, y) + \frac{\rho_{v_1}}{2}\|x\|^2 + \frac{\rho_{v_2}}{2}\|y\|^2$ is convex on $\mathbb{R}^{d_x} \times \mathbb{R}^{d_y}$. Furthermore, for $\gamma \in (0, \frac{1}{2L_2})$, the function $V_\gamma(x, y)$ is differentiable, and its gradient is given by: $\nabla V_\gamma(x, y) := \left(\nabla_x G(x, \theta_\gamma^*(x, y)), \frac{y - \theta_\gamma^*(x, y)}{\gamma}\right)^T$. In addition, the following inequality holds:*

$$-V_\gamma(x, y) \leq -V_\gamma(\bar{x}, \bar{y}) - \left\langle \nabla V_\gamma(\bar{x}, \bar{y}), (x, y) - (\bar{x}, \bar{y}) \right\rangle + \frac{\rho_{v_1}}{2}\|x - \bar{x}\|^2 + \frac{\rho_{v_2}}{2}\|y - \bar{y}\|^2, \quad (29)$$

*for $(\bar{x}, \bar{y}) \in \mathbb{R}^{d_x} \times \mathbb{R}^{d_y}$.*

By Liu et al. (2024, Lemma A.9), we can easily derive the following lemma.

**Lemma F.2** (Property of $\theta_\gamma^*(x, y)$ (Liu et al., 2024)). *Let $\gamma \in (0, \frac{1}{2L_2})$. Then, there exists $L_\theta > 0$ such that for any $(x, y), (x', y') \in \mathbb{R}^{d_x} \times \mathbb{R}^{d_y}$, the following inequality holds:*

$$\|\theta_\gamma^*(x, y) - \theta_\gamma^*(x', y')\| \leq L_\theta \|(x, y) - (x', y')\|, \quad (30)$$

*which $L_\theta$ is some positive constant.*

We next characterize the smoothness of the regularized objective $\Phi_{\mu_k}(x, y)$, which plays a key role in our convergence analysis.

**Lemma F.3** (Property of $\Phi_{\mu_k}(x, y)$). *Under Assumption 3.1, if $\gamma \in (0, \frac{1}{2L_2})$, then $\Phi_{\mu_k}(x, y)$ is $L_{\Phi_k}$-smooth with respect to $(x, y)$, where $L_{\Phi_k} := \mu_k L_1 + L_2 + \max\{L_2, 1/\gamma\}$.*

Note that $L_{\Phi_k}$ has both a positive lower and upper bound, i.e., $L_{\Phi_0} \geq L_{\Phi_k} \geq L_{\Phi_\infty} := L_2 + \min\{L_2, 1/\gamma\}$, due to the decay of the penalty parameter $\mu_k$.

**Lemma F.4.** *Consider the mixing matrix $\mathbf{W} = (w_{ij}) \in \mathbb{R}^{n \times n}$ defined in Assumption 3.3, for any $x_1, \ldots, x_n \in \mathbb{R}^{d_x}$, let $\bar{x} = \frac{1}{n}\sum_{i=1}^n x_i$, we have*

$$
\begin{aligned}
&(a) \quad \sum_{i=1}^n \left\| \sum_{j=1}^n w_{ij} x_j \right\|^2 \leq \sum_{j=1}^n \|x_j\|^2, \\
&(b) \quad \sum_{i=1}^n \left\| \sum_{j=1}^n w_{ij}(x_j - \bar{x}) \right\|^2 \leq \rho^2 \sum_{i=1}^n \|x_i - \bar{x}\|^2,
\end{aligned}
\quad (31)
$$

*where $\rho$ is defined in Assumption 3.3.*

### F.3 CONVERGENCE ANALYSIS

**Lemma F.5.** *The sequence $\{\theta_i^k\}$, $\{x_i^k\}$ and $\{y_i^k\}$ generated by Algorithm SUN-DSBO-SE satisfies*

$$
\begin{aligned}
\sum_{i=1}^n \mathbb{E}\|x_i^{k+1} - \bar{x}^{k+1}\|^2 \leq{}& \rho \sum_{i=1}^n \mathbb{E}\|x_i^k - \bar{x}^k\|^2 + 6\frac{\rho^2 \lambda_x^{k^2}}{1 - \rho} 3n(\mu_k^2 \delta_f^2 + 2\delta_g^2) \\
&+ 3\frac{\rho^2 \lambda_x^{k^2}}{1 - \rho} 3(\mu_k^2 \sigma_f^2 + 2\sigma_g^2),
\end{aligned}
$$

$$
\begin{aligned}
\sum_{i=1}^n \mathbb{E}\|y_i^{k+1} - \bar{y}^{k+1}\|^2 \leq{}& \rho \sum_{i=1}^n \mathbb{E}\|y_i^k - \bar{y}^k\|^2 + 6\frac{\rho^2 \lambda_y^{k^2}}{1 - \rho} 3n(\mu_k^2 \delta_f^2 + \delta_g^2) \\
&+ 3\frac{\rho^2 \lambda_y^{k^2}}{1 - \rho} 3(\mu_k^2 \sigma_f^2 + \sigma_g^2),
\end{aligned}
$$

$$
\sum_{i=1}^n \mathbb{E}\|\theta_i^{k+1} - \bar{\theta}^{k+1}\|^2 \leq \rho \sum_{i=1}^n \mathbb{E}\|\theta_i^k - \bar{\theta}^k\|^2 + 6\frac{\rho^2 \lambda_\theta^{k^2}}{1 - \rho} 3n\delta_g^2 + 3\frac{\rho^2 \lambda_\theta^{k^2}}{1 - \rho} 3\sigma_g^2.
$$

*Proof.* First, we consider the term $\sum_{i=1}^{n} \mathbb{E}\|x_i^{k+1} - \bar{x}^{k+1}\|^2$,

$$\sum_{i=1}^{n} \mathbb{E}\big[\|x_i^{k+1} - \bar{x}^{k+1}\|^2 \big| \mathcal{F}_k\big]$$

$$= \sum_{i=1}^{n} \mathbb{E}\bigg[\Big\| \sum_{j=1}^{n} w_{ij}(x_j^k - \lambda_x^k d_{x,j,k}^{(\zeta_i^k)}) - \frac{1}{n}\sum_{s=1}^{n}\sum_{j=1}^{n} w_{sj}(x_j^k - \lambda_x^k d_{x,j,k}^{(\zeta_i^k)})\Big\|^2 \bigg| \mathcal{F}_k\bigg]$$

$$\leq \Big(1 + \frac{1-\rho}{\rho}\Big) \sum_{i=1}^{n} \Big\| \sum_{j=1}^{n} w_{ij} x_j^k - \bar{x}^k \Big\|^2$$

$$+ \Big(1 + \frac{\rho}{1-\rho}\Big) \lambda_x^{k2} \sum_{i=1}^{n} \mathbb{E}\bigg[\Big\| \sum_{j=1}^{n} w_{ij}(d_{x,i,k}^{(\zeta_i^k)} - \bar{d}_x^k)\Big\|^2 \bigg| \mathcal{F}_k\bigg].$$

Taking expectations on both sides, we obtain

$$\sum_{i=1}^{n} \mathbb{E}\|x_i^{k+1} - \bar{x}^{k+1}\|^2$$

$$\leq \rho \sum_{i=1}^{n} \mathbb{E}\|x_i^k - \bar{x}^k\|^2 + \frac{\rho^2 \lambda_x^{k2}}{1-\rho} \sum_{i=1}^{n} \mathbb{E}\|d_{x,i,k}^{(\zeta_i^k)} - \bar{d}_x^k\|^2$$

$$\leq \rho \sum_{i=1}^{n} \mathbb{E}\|x_i^k - \bar{x}^k\|^2 + 3\frac{\rho^2 \lambda_x^{k2}}{1-\rho} \sum_{i=1}^{n} \Big(\mathbb{E}\|\tilde{d}_{x,i}^k - d_{x,i,k}^{(\zeta_i^k)}\|^2 + \|\tilde{d}_{x,i}^k - \tilde{\bar{d}}_x^k\|^2 + \|\tilde{\bar{d}}_x^k - \bar{d}_x^k\|^2\Big)$$

$$\leq \rho \sum_{i=1}^{n} \mathbb{E}\|x_i^k - \bar{x}^k\|^2 + 6\frac{\rho^2 \lambda_x^{k2}}{1-\rho} 3n\big(\mu_k^2 \delta_f^2 + 2\delta_g^2\big) + 3\frac{\rho^2 \lambda_x^{k2}}{1-\rho} 3\big(\mu_k^2 \sigma_f^2 + 2\sigma_g^2\big).$$

Similarly, we have

$$\sum_{i=1}^{n} \mathbb{E}\|y_i^{k+1} - \bar{y}^{k+1}\|^2$$

$$\leq \rho \sum_{i=1}^{n} \mathbb{E}\|y_i^k - \bar{y}^k\|^2 + 6\frac{\rho^2 \lambda_y^{k2}}{1-\rho} 3n\big(\mu_k^2 \delta_f^2 + \delta_g^2\big) + 3\frac{\rho^2 \lambda_y^{k2}}{1-\rho} 3\big(\mu_k^2 \sigma_f^2 + \sigma_g^2\big),$$

and

$$\sum_{i=1}^{n} \mathbb{E}\|\theta_i^{k+1} - \bar{\theta}^{k+1}\|^2 \leq \rho \sum_{i=1}^{n} \mathbb{E}\|\theta_i^k - \bar{\theta}^k\|^2 + 6\frac{\rho^2 \lambda_\theta^{k2}}{1-\rho} 3n\delta_g^2 + 3\frac{\rho^2 \lambda_\theta^{k2}}{1-\rho} 3\sigma_g^2. \tag{32}$$

$\square$

**Lemma F.6** (Descent in $\Phi_{\mu_k}(x,y)$). *Under Assumptions 3.1, $\gamma \in (0, \frac{1}{2L_2})$, the sequence of $(\bar{x}^k, \bar{y}^k, \bar{\theta}^k)$ generated by SUN-DSBO-SE satisfies*

$$\mathbb{E}\Phi_{\mu_k}(\bar{x}^{k+1}, \bar{y}^{k+1}) - \mathbb{E}\Phi_{\mu_k}(\bar{x}^k, \bar{y}^k)$$

$$\leq -\frac{1}{2\lambda_x^k}\|\mathbb{E}[\bar{x}^{k+1} - \bar{x}^k]\|^2 + \frac{L_{\Phi_k}}{2}\mathbb{E}\|\bar{x}^{k+1} - \bar{x}^k\|^2 - \frac{1}{2\lambda_y^k}\|\mathbb{E}[\bar{y}^{k+1} - \bar{y}^k]\|^2 + \frac{L_{\Phi_k}}{2}\mathbb{E}\|\bar{y}^{k+1} - \bar{y}^k\|^2$$

$$+ \big(\lambda_x^k L_2^2 + \lambda_y^k \frac{1}{\gamma^2}\big)\mathbb{E}\|\bar{\theta}^k - \theta_\gamma^*(\bar{x}^k, \bar{y}^k)\|^2 + \big(3\lambda_x^k\big(\mu_k^2 L_1^2 + 2L_2^2\big) + 4\lambda_y^k\big(\mu_k^2 L_1^2 + L_2^2\big)\big)\mathbb{E}\Delta_x^k$$

$$+ \Big(3\lambda_x^k L_2^2 + 4\lambda_y^k\big(\mu_k^2 L_1^2 + L_2^2 + \frac{1}{\gamma^2}\big)\Big)\mathbb{E}\Delta_y^k + \Big(3\lambda_x^k L_2^2 + \frac{4}{\gamma^2}\lambda_y^k\Big)\mathbb{E}\Delta_\theta^k,$$

*where $\Phi_{\mu_k}(x,y) := \mu_k\big(F(x,y) - \underline{F}\big) + G(x,y) - V_\gamma(x,y)$.*

*Proof.* Considering the update rule for the variable $x$ as defined in (1) in server and leveraging the property of the projection operator $\text{Proj}_X$, it follows that

$$\langle \bar{x}^k - \lambda_x^k \bar{d}_x^k - \bar{x}^{k+1}, \bar{x}^k - \bar{x}^{k+1}\rangle \leq 0, \tag{33}$$

which leading to

$$\langle \tilde{\bar{d}}_x^k, \mathbb{E}[\bar{x}^{k+1} - \bar{x}^k] \rangle \leq -\frac{1}{\lambda_x^k} \|\mathbb{E}[\bar{x}^{k+1} - \bar{x}^k]\|^2. \tag{34}$$

Similarly,

$$\langle \tilde{\bar{d}}_y^k, \mathbb{E}[\bar{y}^{k+1} - \bar{y}^k] \rangle \leq -\frac{1}{\lambda_y^k} \|\mathbb{E}[\bar{y}^{k+1} - \bar{y}^k]\|^2. \tag{35}$$

By the Lemma F.3, we have

$$\mathbb{E}\Phi_{\mu_k}(\bar{x}^{k+1}, \bar{y}^{k+1}) - \mathbb{E}\Phi_{\mu_k}(\bar{x}^k, \bar{y}^k)$$

$$\leq \mathbb{E}\langle \nabla_x \Phi_{\mu_k}(\bar{x}^k, \bar{y}^k) - \tilde{\bar{d}}_x^k + \tilde{\bar{d}}_x^k, \bar{x}^{k+1} - \bar{x}^k \rangle + \mathbb{E}\langle \nabla_y \Phi_{\mu_k}(\bar{x}^k, \bar{y}^k) - \tilde{\bar{d}}_y^k + \tilde{\bar{d}}_y^k, \bar{y}^{k+1} - \bar{y}^k \rangle$$

$$\quad + \frac{L_{\Phi_k}}{2}(\mathbb{E}\|\bar{x}^{k+1} - \bar{x}^k\|^2 + \mathbb{E}\|\bar{y}^{k+1} - \bar{y}^k\|^2)$$

$$\leq \mathbb{E}\langle \nabla_x \Phi_{\mu_k}(\bar{x}^k, \bar{x}^k) - \tilde{\bar{d}}_x^k, \bar{x}^{k+1} - \bar{x}^k \rangle + \mathbb{E}\langle \nabla_y \Phi_{\mu_k}(\bar{x}^k, \bar{y}^k) - \tilde{\bar{d}}_y^k, \bar{y}^{k+1} - \bar{y}^k \rangle$$

$$\quad - \frac{1}{\lambda_x^k}\|\mathbb{E}[\bar{x}^{k+1} - \bar{x}^k]\|^2 - \frac{1}{\lambda_y^k}\|\mathbb{E}[\bar{y}^{k+1} - \bar{y}^k]\|^2 + \frac{L_{\Phi_k}}{2}(\mathbb{E}\|\bar{x}^{k+1} - \bar{x}^k\|^2 + \mathbb{E}\|\bar{y}^{k+1} - \bar{y}^k\|^2).$$

For $\mathbb{E}\langle \nabla_x \Phi_{\mu_k}(\bar{x}^k, \bar{y}^k) - \tilde{\bar{d}}_x^k, \bar{x}^{k+1} - \bar{x}^k \rangle$, we have

$$\mathbb{E}\langle \nabla_x \Phi_{\mu_k}(\bar{x}^k, \bar{y}^k) - \tilde{\bar{d}}_x^k, \bar{x}^{k+1} - \bar{x}^k \rangle$$

$$\leq \frac{\lambda_x^k}{2} \mathbb{E}\left\|\nabla_x \Phi_{\mu_k}(\bar{x}^k, \bar{y}^k) - \tilde{\bar{d}}_x^k\right\|^2 + \frac{1}{2\lambda_x^k}\|\mathbb{E}[\bar{x}^{k+1} - \bar{x}^k]\|^2. \tag{36}$$

According to the definition of $\tilde{\bar{d}}_x^k$ in (24), we can analyze the term $\left\|\nabla_x \Phi_{\mu_k}(\bar{x}^k, \bar{y}^k) - \tilde{\bar{d}}_x^k\right\|^2$ in (36) as follows:

$$\mathbb{E}\left\|\nabla_x \Phi_{\mu_k}(\bar{x}^k, \bar{y}^k) - \frac{1}{n}\sum_{i=1}^n \tilde{d}_{x,i}^k\right\|^2$$

$$= \mathbb{E}\left\|\frac{1}{n}\sum_{i=1}^n \left[\nabla_x \phi_{\mu_k}^i(\bar{x}^k, \bar{y}^k)[\theta_\gamma^*(\bar{x}^k, \bar{y}^k)] - \nabla_x \phi_{\mu_k}^i(\bar{x}^k, \bar{y}^k)[\bar{\theta}^k] + \nabla_x \phi_{\mu_k}^i(\bar{x}^k, \bar{y}^k)[\bar{\theta}^k] - \tilde{d}_{x,i}^k\right]\right\|^2$$

$$\leq 2L_2^2 \mathbb{E}\left\|\theta_\gamma^*(\bar{x}^k, \bar{y}^k) - \bar{\theta}^k\right\|^2 + 2\mathbb{E}\left\|\frac{1}{n}\sum_{i=1}^n \left(\nabla_x \phi_{\mu_k}^i(\bar{x}^k, \bar{y}^k)[\bar{\theta}^k] - \tilde{d}_{x,i}^k\right)\right\|^2, \tag{37}$$

where $\phi_{\mu_k}^i(x, y)[\theta]$ is defined in (27).

For the term $\mathbb{E}\left\|\frac{1}{n}\sum_{i=1}^n \nabla_x \phi_{\mu_k}^i(\bar{x}^k, \bar{y}^k)[\bar{\theta}^k] - \tilde{d}_{x,i}^k\right\|^2$, according to the definition, we have

$$\mathbb{E}\left\|\frac{1}{n}\sum_{i=1}^n \left[\nabla_x \phi_{\mu_k}^i(\bar{x}^k, \bar{y}^k)[\bar{\theta}^k] - \tilde{d}_{x,i}^k\right]\right\|^2$$

$$\overset{(a)}{\leq} \frac{1}{n}\sum_{i=1}^n \mathbb{E}\left\|\nabla_x \phi_{\mu_k}^i(\bar{x}^k, \bar{y}^k)[\bar{\theta}^k] - \tilde{d}_{x,i}^k\right\|^2$$

$$\overset{(b)}{\leq} 3(\mu_k^2 L_1^2 + 2L_2^2) \cdot \frac{1}{n}\sum_{i=1}^n \mathbb{E}\|x_i^k - \bar{x}^k\|^2 + 3(\mu_k^2 L_1^2 + L_2^2) \cdot \frac{1}{n}\sum_{i=1}^n \mathbb{E}\|y_i^k - \bar{y}^k\|^2$$

$$\quad + L_2^2 \cdot \frac{1}{n}\sum_{i=1}^n \mathbb{E}\|\theta_i^k - \bar{\theta}^k\|^2, \tag{38}$$

where $(a)$ comes from Jensen's inequality, $(b)$ comes from the Assumption 3.1.

Combining the inequalities (37) and (38), we have

$$\mathbb{E}\left\|\nabla_x \Phi_{\mu_k}(\bar{x}^k, \bar{y}^k) - \frac{1}{n}\sum_{i=1}^n \tilde{d}_{x,i}^k\right\|^2$$

$$\leq 2L_2^2 \mathbb{E}\|\theta_\gamma^*(\bar{x}^k, \bar{y}^k) - \bar{\theta}^k\|^2 + 6\big(\mu_k^2 L_1^2 + 2L_2^2\big)\frac{1}{n}\sum_{i=1}^n \mathbb{E}\|x_i^k - \bar{x}^k\|^2 \tag{39}$$

$$+ 6\big(\mu_k^2 L_1^2 + L_2^2\big)\frac{1}{n}\sum_{i=1}^n \mathbb{E}\|y_i^k - \bar{y}^k\|^2 + 6L_2^2 \frac{1}{n}\sum_{i=1}^n \mathbb{E}\|\theta_i^k - \bar{\theta}^k\|^2.$$

Similarly,

$$\mathbb{E}\left\|\nabla_y \Phi_{\mu_k}(\bar{x}^k, \bar{y}^k) - \bar{d}_y^k\right\|^2$$

$$\leq \frac{2}{\gamma^2}\mathbb{E}\|\theta_\gamma^*(\bar{x}^k, \bar{y}^k) - \bar{\theta}^k\|^2 + 8\big(\mu_k^2 L_1^2 + L_2^2\big)\frac{1}{n}\sum_{i=1}^n \mathbb{E}\|x_i^k - \bar{x}^k\|^2 \tag{40}$$

$$+ 8\big(\mu_k^2 L_1^2 + L_2^2 + \frac{1}{\gamma^2}\big)\frac{1}{n}\sum_{i=1}^n \mathbb{E}\|y_i^k - \bar{y}^k\|^2 + 8\frac{1}{\gamma^2}\frac{1}{n}\sum_{i=1}^n \mathbb{E}\|\theta_i^k - \bar{\theta}^k\|^2.$$

Based on the above,

$$\mathbb{E}\Phi_{\mu_k}(\bar{x}^{k+1}, \bar{y}^{k+1}) - \mathbb{E}\Phi_{\mu_k}(\bar{x}^k, \bar{y}^k)$$

$$\leq -\frac{1}{2\lambda_x^k}\|\mathbb{E}[\bar{x}^{k+1} - \bar{x}^k]\|^2 + \frac{L_{\Phi_k}}{2}\mathbb{E}\|\bar{x}^{k+1} - \bar{x}^k\|^2 - \frac{1}{2\lambda_y^k}\|\mathbb{E}[\bar{y}^{k+1} - \bar{y}^k]\|^2 + \frac{L_{\Phi_k}}{2}\mathbb{E}\|\bar{y}^{k+1} - \bar{y}^k\|^2$$

$$+ \big(\lambda_x^k L_2^2 + \lambda_y^k \frac{1}{\gamma^2}\big)\mathbb{E}\|\bar{\theta}^k - \theta_\gamma^*(\bar{x}^k, \bar{y}^k)\|^2 + \big(3\lambda_x^k\big(\mu_k^2 L_1^2 + 2L_2^2\big) + 4\lambda_y^k\big(\mu_k^2 L_1^2 + L_2^2\big)\big)\mathbb{E}\Delta_x^k$$

$$+ \left(3\lambda_x^k L_2^2 + 4\lambda_y^k\big(\mu_k^2 L_1^2 + L_2^2 + \frac{1}{\gamma^2}\big)\right)\mathbb{E}\Delta_y^k + \left(3\lambda_x^k L_2^2 + \frac{4}{\gamma^2}\lambda_y^k\right)\mathbb{E}\Delta_\theta^k.$$

$$\square$$

**Lemma F.7.** *Under Assumptions 3.1 and 3.2, let $\bar{\theta}^k$ be the sequence generated by Algorithm SUN-DSBO-SE. Then the error $\|\bar{\theta}^k - \theta_\gamma^*(\bar{x}^k, \bar{y}^k)\|^2$ satisfies the following recursion:*

$$\mathbb{E}\|\bar{\theta}^{k+1} - \theta_\gamma^*(\bar{x}^{k+1}, \bar{y}^{k+1})\|^2$$

$$\leq (1+\delta_k)\Big(\mathbb{E}\|\bar{\theta}^k - \theta_\gamma^*(\bar{x}^k, \bar{y}^k)\|^2 + \lambda_\theta^{k^2}\mathbb{E}\|\bar{d}_\theta^k\|^2 + \lambda_\theta^k \frac{6}{\eta}L_2^2\mathbb{E}\Delta_x^k + \lambda_\theta^k\frac{6}{\eta}\big(L_2^2 + \frac{1}{\gamma^2}\big)\mathbb{E}\Delta_\theta^k$$

$$+ \lambda_\theta^k \frac{6}{\eta}\frac{1}{\gamma^2}\mathbb{E}\Delta_y^k - \rho\lambda_\theta^k\mathbb{E}\|\bar{\theta}^k - \theta_\gamma^*(\bar{x}^k, \bar{y}^k)\|^2\Big)$$

$$+ 2L_\theta^2\big(1 + \frac{1}{\delta_k}\big)\big(\mathbb{E}\|\bar{x}^{k+1} - \bar{x}^k\|^2 + \mathbb{E}\|\bar{y}^{k+1} - \bar{y}^k\|^2\big), \tag{41}$$

*where $\delta_k > 0$, and $\eta := \frac{1}{\gamma} - L_2$.*

*Proof.* For the gap of $\|\bar{\theta}^k - \theta_\gamma^*(\bar{x}^k, \bar{y}^k)\|^2$ , we have

$$\mathbb{E}\|\bar{\theta}^{k+1} - \theta_\gamma^*(\bar{x}^{k+1}, \bar{y}^{k+1})\|^2$$

$$= \mathbb{E}\|\bar{\theta}^{k+1} - \theta_\gamma^*(\bar{x}^k, \bar{y}^k)\|^2 + \mathbb{E}\|\theta_\gamma^*(\bar{x}^{k+1}, \bar{y}^{k+1}) - \theta_\gamma^*(\bar{x}^k, \bar{y}^k)\|^2$$

$$+ 2\mathbb{E}\langle \bar{\theta}^{k+1} - \theta_\gamma^*(\bar{x}^k, \bar{y}^k), \theta_\gamma^*(\bar{x}^{k+1}, \bar{y}^{k+1}) - \theta_\gamma^*(\bar{x}^k, \bar{y}^k)\rangle. \tag{42}$$

For the last term in( 42), we have

$$2\mathbb{E}\langle \bar{\theta}^{k+1} - \theta_\gamma^*(\bar{x}^k, \bar{y}^k), \theta_\gamma^*(\bar{x}^{k+1}, \bar{y}^{k+1}) - \theta_\gamma^*(\bar{x}^k, \bar{y}^k)\rangle$$

$$\overset{(a)}{\leq} \delta_k\mathbb{E}\|\bar{\theta}^{k+1} - \theta_\gamma^*(\bar{x}^k, \bar{y}^k)\|^2 + \frac{1}{\delta_k}\mathbb{E}\|\theta_\gamma^*(\bar{x}^{k+1}, \bar{y}^{k+1}) - \theta_\gamma^*(\bar{x}^k, \bar{y}^k)\|^2$$

$$\overset{(b)}{\leq} \delta_k\mathbb{E}\|\bar{\theta}^{k+1} - \theta_\gamma^*(\bar{x}^k, \bar{y}^k)\|^2 + \frac{2L_\theta^2}{\delta_k}\mathbb{E}\|(\bar{x}^{k+1}, \bar{y}^{k+1}) - (\bar{x}^k, \bar{y}^k)\|^2,$$

where $(a)$ can be derived from Young's inequality, $(b)$ comes from the Lemma F.2. Then the inequality can be reformulated as

$$\mathbb{E}\left\|\bar{\theta}^{k+1} - \theta_\gamma^*(\bar{x}^{k+1}, \bar{y}^{k+1})\right\|^2 \leq (1 + \delta_k)\,\mathbb{E}\left\|\bar{\theta}^{k+1} - \theta_\gamma^*(\bar{x}^k, \bar{y}^k)\right\|^2$$
$$+ 2L_\theta^2\left(1 + \frac{1}{\delta_k}\right)\mathbb{E}\left\|(\bar{x}^{k+1}, \bar{y}^{k+1}) - (\bar{x}^k, \bar{y}^k)\right\|^2. \quad (43)$$

For the term $\mathbb{E}\|\bar{\theta}^{k+1} - \theta_\gamma^*(\bar{x}^k, \bar{y}^k)\|^2$ in Eq. (43),

$$\mathbb{E}\|\bar{\theta}^{k+1} - \theta_\gamma^*(\bar{x}^k, \bar{y}^k)\|^2 \overset{(a)}{\leq} \mathbb{E}\|\bar{\theta}^k - \theta_\gamma^*(\bar{x}^k, \bar{y}^k) - \lambda_\theta^k \bar{d}_\theta^k\|^2$$
$$= \mathbb{E}\|\bar{\theta}^k - \theta_\gamma^*(\bar{x}^k, \bar{y}^k)\|^2 + \lambda_\theta^{k\,2}\mathbb{E}\|\bar{d}_\theta^k\|^2 - 2\lambda_\theta^k\mathbb{E}\left\langle\bar{\theta}^k - \theta_\gamma^*(\bar{x}^k, \bar{y}^k), \bar{d}_\theta^k\right\rangle,$$
$$(44)$$

where (a) comes from the iteration of variable $\bar{\theta}^k$ in (25).

For the term $-\mathbb{E}\left\langle\bar{\theta}^k - \theta_\gamma^*(\bar{x}^k, \bar{y}^k), \bar{d}_\theta^k\right\rangle$ in (44),

$$-\mathbb{E}\left\langle\bar{\theta}^k - \theta_\gamma^*(\bar{x}^k, \bar{y}^k), \bar{d}_\theta^k\right\rangle$$

$$= -\mathbb{E}\left\langle\bar{\theta}^k - \theta_\gamma^*(\bar{x}^k, \bar{y}^k), \frac{1}{n}\sum_{i=1}^n d_{\theta,i,k}^{(\zeta_i^k)}\right\rangle$$

$$= -\mathbb{E}\left\langle\bar{\theta}^k - \theta_\gamma^*(\bar{x}^k, \bar{y}^k), \frac{1}{n}\sum_{i=1}^n d_{\theta,i,k}^{(\zeta_i^k)} - \frac{1}{n}\sum_{i=1}^n[\nabla_y g_i(\bar{x}^k, \bar{\theta}^k) + \frac{1}{\gamma}(\bar{\theta}^k - \bar{y}^k)]\right\rangle$$

$$- \mathbb{E}\left\langle\bar{\theta}^k - \theta_\gamma^*(\bar{x}^k, \bar{y}^k), \frac{1}{n}\sum_{i=1}^n[\nabla_y g_i(\bar{x}^k, \bar{\theta}^k) + \frac{1}{\gamma}(\bar{\theta}^k - \bar{y}^k)] - \frac{1}{n}\sum_{i=1}^n[\nabla_y g_i(\bar{x}^k, \theta_\gamma^*(\bar{x}^k, \bar{y}^k))\right.$$

$$\left.+ \frac{1}{\gamma}(\theta_\gamma^*(\bar{x}^k, \bar{y}^k) - \bar{y}^k)]\right\rangle$$

$$\overset{(a)}{\leq} \frac{\eta}{2}\mathbb{E}\|\bar{\theta}^k - \theta_\gamma^*(\bar{x}^k, \bar{y}^k)\|^2 + \frac{1}{\eta}\frac{1}{n}\sum_{i=1}^n\mathbb{E}\left\|d_{\theta,i,k}^{(\zeta_i^k)} - \nabla_y g_i(\bar{x}^k, \bar{\theta}^k) + \frac{1}{\gamma}(\bar{\theta}^k - \bar{y}^k)\right\|^2$$

$$- \eta\mathbb{E}\|\bar{\theta}^k - \theta_\gamma^*(\bar{x}^k, \bar{y}^k)\|^2$$

$$\overset{(b)}{\leq} \frac{3}{\eta}L_2^2\mathbb{E}\Delta_x^k + \frac{3}{\eta}\left(L_2^2 + \frac{1}{\gamma^2}\right)\mathbb{E}\Delta_\theta^k + \frac{3}{\eta}\frac{1}{\gamma^2}\mathbb{E}\Delta_y^k - \frac{\eta}{2}\mathbb{E}\|\bar{\theta}^k - \theta_\gamma^*(\bar{x}^k, \bar{y}^k)\|^2, \quad (45)$$

where the first two terms in (a) comes from the Young's inequality and the last term in (a) comes from the strong convexity of $G(x, \theta) + \frac{1}{2\gamma}\|\theta - y\|^2$ w.r.t. $\theta$ which can be derived from the Lemma F.1, where (b) comes from the L-smoothness of $f_i(x, y)$ and $g_i(x, y)$ in Assumption 3.1. Then we have

$$\mathbb{E}\|\bar{\theta}^{k+1} - \theta_\gamma^*(\bar{x}^{k+1}, \bar{y}^{k+1})\|^2$$

$$\leq (1 + \delta_k)\left(\mathbb{E}\|\bar{\theta}^k - \theta_\gamma^*(\bar{x}^k, \bar{y}^k)\|^2 + \lambda_\theta^{k\,2}\mathbb{E}\|\bar{d}_\theta^k\|^2 + \lambda_\theta^k\frac{6}{\eta}L_2^2\mathbb{E}\Delta_x^k + \lambda_\theta^k\frac{6}{\eta}\left(L_2^2 + \frac{1}{\gamma^2}\right)\mathbb{E}\Delta_\theta^k\right.$$

$$\left.+ \lambda_\theta^k\frac{6}{\eta}\frac{1}{\gamma^2}\mathbb{E}\Delta_y^k - \eta\lambda_\theta^k\mathbb{E}\|\bar{\theta}^k - \theta_\gamma^*(\bar{x}^k, \bar{y}^k)\|^2\right) \quad (46)$$

$$+ 2L_\theta^2\left(1 + \frac{1}{\delta_k}\right)\left(\mathbb{E}\|\bar{x}^{k+1} - \bar{x}^k\|^2 + \mathbb{E}\|\bar{y}^{k+1} - \bar{y}^k\|^2\right).$$

$$\square$$

**Lemma F.8.** *The sequences $\{d_{\theta,i}^k\}$ generated by Algorithm SUN-DSBO-SE satisfy*

$$\mathbb{E}\|\bar{d}_\theta^k\|^2 \leq 2\frac{\sigma_g^2}{n} + 12L_2^2\frac{1}{n}\sum_{i=1}^n\mathbb{E}\|x_i^k - \bar{x}^k\|^2 + 12\left(L_2^2 + \frac{1}{\gamma^2}\right)\frac{1}{n}\sum_{i=1}^n\mathbb{E}\|\theta_i^k - \bar{\theta}^k\|^2$$

$$+ 12\frac{1}{\gamma^2}\frac{1}{n}\sum_{i=1}^n\mathbb{E}\|y_i^k - \bar{y}^k\|^2 + 4\left(L_2^2 + \frac{1}{\gamma^2}\right)\mathbb{E}\|\theta_\gamma^*(\bar{x}^k, \bar{y}^k) - \bar{\theta}^k\|^2.$$

*Proof.* We will analyze the term $\mathbb{E}\|\bar{d}_\theta^k\|^2$:

$$\mathbb{E}\|\bar{d}_\theta^k\|^2 \leq 2\mathbb{E}\|\bar{d}_\theta^k - \tilde{\bar{d}}_\theta^k\|^2 + 2\mathbb{E}\|\tilde{\bar{d}}_\theta^k\|^2$$

$$\overset{(a)}{\leq} 2\frac{\sigma_g^2}{n} + 2\mathbb{E}\left\|\frac{1}{n}\sum_{i=1}^n \left(\nabla_2 g_i(x_i^k, \theta_i^k) + \frac{1}{\gamma}(\theta_i^k - y_i^k) - \nabla_2 g_i(\bar{x}^k, \bar{\theta}^k) - \frac{1}{\gamma}(\bar{\theta}^k - \bar{y}^k)\right.\right.$$

$$\left.\left. + \nabla_2 g_i(\bar{x}^k, \bar{\theta}^k) + \frac{1}{\gamma}(\bar{\theta}^k - \bar{y}^k) - \nabla_2 g_i(\bar{x}^k, \theta_\gamma^*(\bar{x}^k, \bar{y}^k)) - \frac{1}{\gamma}(\theta_\gamma^*(\bar{x}^k, \bar{y}^k) - \bar{y}^k)\right)\right\|^2$$

$$\leq 2\frac{\sigma_g^2}{n} + 12L_2^2\frac{1}{n}\sum_{i=1}^n \mathbb{E}\|x_i^k - \bar{x}^k\|^2 + 12\left(L_2^2 + \frac{1}{\gamma^2}\right)\frac{1}{n}\sum_{i=1}^n \mathbb{E}\|\theta_i^k - \bar{\theta}^k\|^2$$

$$+ 12\frac{1}{\gamma^2}\frac{1}{n}\sum_{i=1}^n \mathbb{E}\|y_i^k - \bar{y}^k\|^2 + 4\left(L_2^2 + \frac{1}{\gamma^2}\right)\mathbb{E}\|\theta_\gamma^*(\bar{x}^k, \bar{y}^k) - \bar{\theta}^k\|^2,$$

where (a) comes from the Assumption 3.2 and $\theta_\gamma^*(x, y)$ is the unique solution of (3). $\qquad\square$

We define the Lyapunov function

$$\mathcal{L}_{\text{se}}^k := \mathbb{E}\Phi_{\mu_k}(\bar{x}^k, \bar{y}^k) + a_1\mathbb{E}\|\bar{\theta}^k - \theta_\gamma^*(\bar{x}^k, \bar{y}^k)\|^2$$

$$+ \frac{a_2^k}{n}\sum_{i=1}^n \mathbb{E}\|x_i^k - \bar{x}^k\|^2 + \frac{a_3^k}{n}\sum_{i=1}^n \mathbb{E}\|y_i^k - \bar{y}^k\|^2 + \frac{a_4^k}{n}\sum_{i=1}^n \mathbb{E}\|\theta_i^k - \bar{\theta}^k\|^2,$$

where $a_2^k := \tau_2\lambda_x^k$, $a_3^k := \tau_3\lambda_y^k$ and $a_4^k := \tau_4\lambda_\theta^k$.

Combining Lemmas F.5, F.6, F.7, and F.8, we obtain

$$\mathcal{L}_{\text{se}}^{k+1} - \mathcal{L}_{\text{se}}^k$$

$$\leq -\frac{1}{2\lambda_x^k}\|\mathbb{E}[\bar{x}^{k+1} - \bar{x}^k]\|^2 - \frac{1}{2\lambda_y^k}\|\mathbb{E}[\bar{y}^{k+1} - \bar{y}^k]\|^2 + \frac{L_{\Phi_k}}{2}\mathbb{E}\left(\|\bar{x}^{k+1} - \bar{x}^k\|^2 + \mathbb{E}\|\bar{y}^{k+1} - \bar{y}^k\|^2\right)$$

$$+ \left(\lambda_x^k L_2^2 + \lambda_y^k\frac{1}{\gamma^2}\right)\mathbb{E}\|\bar{\theta}^k - \theta_\gamma^*(\bar{x}^k, \bar{y}^k)\|^2 + \left(3\lambda_x^k\left(\mu_k^2 L_1^2 + 2L_2^2\right) + 4\lambda_y^k\left(\mu_k^2 L_1^2 + L_2^2\right)\right)\mathbb{E}\Delta_x^k$$

$$+ \left(3\lambda_x^k\left(\mu_k^2 L_1^2 + L_2^2\right) + 4\lambda_y^k\left(\mu_k^2 L_1^2 + L_2^2 + \frac{1}{\gamma^2}\right)\right)\mathbb{E}\Delta_y^k + \left(3\lambda_x^k L_2^2 + \frac{4}{\gamma^2}\lambda_y^k\right)\mathbb{E}\Delta_\theta^k$$

$$+ a_1(1 + \delta_k)\left(\mathbb{E}\|\bar{\theta}^k - \theta_\gamma^*(\bar{x}^k, \bar{y}^k)\|^2 + \lambda_\theta^{k^2}\mathbb{E}\|\bar{d}_\theta^k\|^2 + \lambda_\theta^k\frac{6}{\eta}L_2^2\mathbb{E}\Delta_x^k + \lambda_\theta^k\frac{6}{\eta}\left(L_2^2 + \frac{1}{\gamma^2}\right)\mathbb{E}\Delta_\theta^k\right.$$

$$\left. + \lambda_\theta^k\frac{6}{\eta}\frac{1}{\gamma^2}\mathbb{E}\Delta_y^k - \eta\lambda_\theta^k\mathbb{E}\|\bar{\theta}^k - \theta_\gamma^*(\bar{x}^k, \bar{y}^k)\|^2\right)$$

$$+ 2a_1 L_\theta^2\left(1 + \frac{1}{\delta_k}\right)\left(\mathbb{E}\|\bar{x}^{k+1} - \bar{x}^k\|^2 + \mathbb{E}\|\bar{y}^{k+1} - \bar{y}^k\|^2\right) - a_1\mathbb{E}\|\bar{\theta}^k - \theta_\gamma^*(\bar{x}^k, \bar{y}^k)\|^2$$

$$+ a_2^k(\rho - 1)\mathbb{E}\Delta_x^k + a_2^k 6\frac{\rho^2\lambda_x^{k^2}}{1 - \rho}3\left(\mu_k^2\delta_f^2 + 2\delta_g^2\right) + a_2^k 3\frac{\rho^2\lambda_x^{k^2}}{1 - \rho}3\frac{1}{n}\left(\mu_k^2\sigma_f^2 + 2\sigma_g^2\right)$$

$$+ a_3^k(\rho - 1)\mathbb{E}\Delta_y^k + a_3^k 6\frac{\rho^2\lambda_y^{k^2}}{1 - \rho}3\left(\mu_k^2\delta_f^2 + \delta_g^2\right) + a_3^k 3\frac{1}{n}\frac{\rho^2\lambda_y^{k^2}}{1 - \rho}3\left(\mu_k^2\sigma_f^2 + \sigma_g^2\right)$$

$$+ a_4^k(\rho - 1)\mathbb{E}\Delta_\theta^k + a_4^k 6\frac{\rho^2\lambda_\theta^{k^2}}{1 - \rho}3\delta_g^2 + a_3^k 3\frac{\rho^2\lambda_x^{k^2}}{1 - \rho}3\frac{1}{n}\sigma_g^2.$$

We now analyze the coefficients in the above inequality. In particular, we assume that the step sizes follow a proportional relation:

$$\lambda_x^k = \lambda_y^k = c_\lambda\lambda_\theta^k, \tag{47}$$

where $c_\lambda > 0$ is a fixed constant. For the coefficient of term $\|\mathbb{E}[\bar{x}^{k+1} - \bar{x}^k]\|^2$ is

$$-\left(\frac{1}{2\lambda_x^k} - \frac{L_{\Phi_k}}{2}\right) + 2a_1 L_\theta^2\left(1 + \frac{1}{\delta_k}\right).$$

If we take $\lambda_\theta^k \leq \frac{4}{3\eta}$ such that $1 + \frac{1}{\delta_k} \leq 2$, the coefficient of term $\|\mathbb{E}[\bar{x}^{k+1} - \bar{x}^k]\|^2$ is $-\frac{1}{8c_\lambda \lambda_\theta^k}$.

For the coefficient of term $\|\mathbb{E}[\bar{y}^{k+1} - \bar{y}^k]\|^2$ is

$$- \big(\frac{1}{2\lambda_y^k} - \frac{L_{\Phi_k}}{2}\big) + 2a_1 L_\theta^2 \big(1 + \frac{1}{\delta_k}\big).$$

Then we take the coefficient of term $\|\mathbb{E}[\bar{x}^{k+1} - \bar{x}^k]\|^2$ is $-\frac{1}{8c_\lambda \lambda_\theta^k}$.

For the coefficient of term $\mathbb{E}\|\bar{\theta}^k - \theta_\gamma^*(\bar{x}^k, \bar{y}^k)\|^2$ is

$$\lambda_x^k L_2^2 + \lambda_y^k \frac{1}{\gamma^2} + a_1\big((1 + \delta_k)(1 - \eta\lambda_\theta^k) - 1\big) + 4\lambda_\theta^{k^2}\Big(L_2^2 + \frac{1}{\gamma^2}\Big)a_1(1 + \delta_k)$$

$$+ 4\Big(9\tau_5 \lambda_x^{k^3} L_2^2 + 12\tau_6 \lambda_y^{k^3} \frac{1}{\gamma^2} + 9\tau_7 \lambda_\theta^{k^3}\big(L_2^2 + \frac{1}{\gamma^2}\big)\Big)\frac{1}{1-\rho}\Big).$$

If we take $a_1 := \frac{1}{2L_\theta}\sqrt{L_2^2 + \frac{1}{\gamma^2}}$, $\delta_k := \frac{\lambda_\theta^k \eta}{4(1 - \frac{\lambda_\theta^k \eta}{2})}$, where $c_\lambda \leq \frac{\eta}{32a_1 L_\theta^2}$, $\eta := \frac{1}{\gamma} - L_2$, $\lambda_\theta^k \leq \frac{4a_1 L_\theta^2}{\rho L_{\Phi\infty}}$, the coefficient of term $\mathbb{E}\|\bar{\theta}^k - \theta_\gamma^*(\bar{x}^k, \bar{y}^k)\|^2$ is $-\frac{1}{16a_1 L_\theta^2}\big(L_2^2 + \frac{1}{\gamma^2}\big)\lambda_\theta^k$.

For the coefficient of term $\Delta_x^k$ is

$$\Big(3\lambda_x^k\big(\mu_k^2 L_1^2 + 2L_2^2\big) + 4\lambda_y^k\big(\mu_k^2 L_1^2 + L_2^2\big)\Big) + a_1(1 + \delta_k)\lambda_\theta^k \frac{6}{\eta}L_2^2$$

$$+ \tau_2 \lambda_x^k(\rho - 1) + 12L_2^2 \lambda_\theta^{k^2} a_1(1 + \delta_k).$$

If we take $\tau_2 := 2\frac{\Big(3\big(\mu_0 L_1^2 + 2L_2^2\big)c_\lambda + 4\big(\mu_0 L_1^2 + L_2^2\big)c_\lambda\Big) + 2a_1 \frac{6}{\eta}L_2^2}{c_\lambda(1-\rho)}$ and

$$\lambda_\theta^k \leq \frac{\tau_2 c_\lambda(1 - \rho)}{96L_2^2\Big(2a_1 + 4\big(9\tau_5 c_\lambda^3 L_2^2 + 12\tau_6 c_\lambda^3 \frac{1}{\gamma^2} + 9\tau_7\big(L_2^2 + \frac{1}{\gamma^2}\big)\big)\Big)\frac{1}{1-\rho}},$$

the coefficient of term $\Delta_x^k$ is $-\frac{\tau_2 c_\lambda(1-\rho)}{8}\lambda_\theta^k$.

For the coefficient of term $\Delta_y^k$ is

$$\Big(3\lambda_x^k\big(\mu_0 L_1^2 + L_2^2\big) + 4\lambda_y^k\big(\mu_0 L_1^2 + L_2^2 + \frac{1}{\gamma^2}\big)\Big) + a_1(1 + \delta_k)\lambda_\theta^k \frac{6}{\eta}\frac{1}{\gamma^2} + \tau_3 \lambda_y^k(\rho - 1)$$

$$+ 12\frac{1}{\gamma^2}\lambda_\theta^{k^2} a_1(1 + \delta_k).$$

If we take $\tau_3 := 2\frac{\Big(3\big(\mu_0 L_1^2 + L_2^2\big)c_\lambda + 4\big(\mu_0 L_1^2 + L_2^2 + \frac{1}{\gamma^2}\big)c_\lambda\Big) + 2a_1 \frac{6}{\eta}\frac{1}{\gamma^2}}{c_\lambda(1-\rho)}$ and $\lambda_\theta^k \leq \frac{\tau_3 c_\lambda(1-\rho)}{96\frac{1}{\gamma^2}2a_1\frac{1}{1-\rho}}$, the coefficient of term $\Delta_y^k$ is $-\frac{\tau_3 c_\lambda(1-\rho)}{8}\lambda_\theta^k$.

For the coefficient of term $\Delta_\theta^k$ is

$$\Big(3\lambda_x^k L_2^2 + \frac{4}{\gamma^2}\lambda_y^k\Big) + a_1(1 + \delta_k)\lambda_\theta^k \frac{6}{\eta}\big(L_2^2 + \frac{1}{\gamma^2}\big) + \tau_4 \lambda_\theta^k(\rho - 1) + 12\big(\frac{1}{\gamma^2} + L_2^2\big)\lambda_\theta^{k^2} a_1(1 + \delta_k).$$

If we take $\tau_4 := 2\frac{\left(3L_2^2 c_\lambda + \frac{4}{\gamma^2}c_\lambda\right) + 2a_1\frac{6}{\eta}\left(\frac{1}{\gamma^2} + L_2^2\right)}{c_\lambda(1-\rho)}$ and $\lambda_\theta^k \leq \frac{\tau_4 c_\lambda(1-\rho)}{96\left(L_2^2 + \frac{1}{\gamma^2}\right)2a_1\frac{1}{1-\rho}}$, the coefficient of term

$\Delta_\theta^k$ is $-\frac{\tau_4(1-\rho)}{8}\lambda_\theta^k$. Since $\mu_k$ is a decaying sequence, it follows that

$$
\begin{aligned}
\mathcal{L}_{\text{se}}^{k+1} - \mathcal{L}_{\text{se}}^k \leq &- \frac{1}{8c_\lambda\lambda_\theta^k}\|\mathbb{E}[\bar{x}^{k+1} - \bar{x}^k]\|^2 - \frac{1}{8c_\lambda\lambda_\theta^k}\|\mathbb{E}[\bar{y}^{k+1} - \bar{y}^k]\|^2 \\
&- \frac{1}{16a_1 L_\theta^2}\left(L_2^2 + \frac{1}{\gamma^2}\right)\lambda_\theta{}^k\mathbb{E}\|\bar{\theta}^k - \theta_\gamma^*(\bar{x}^k, \bar{y}^k)\|^2 \\
&- \frac{\tau_2 c_\lambda(1-\rho)}{8}c_\lambda\lambda_\theta^k\Delta_x^k - \frac{\tau_3 c_\lambda(1-\rho)}{8}c_\lambda\lambda_\theta^k\Delta_y^k - \frac{\tau_4(1-\rho)}{8}\lambda_\theta^k\Delta_\theta^k \\
&+ \lambda_\theta^k{}^2 M_1\frac{1}{n}\left(\mu_0\delta_f^2 + 2\delta_g^2\right) + \lambda_\theta^k{}^3 M_2\frac{1}{1-\rho}\left(\mu_0\delta_f^2 + 2\delta_g^2\right) \\
&+ \lambda_\theta^k{}^3 M_3\frac{1}{1-\rho}\left(\mu_0\sigma_f^2 + 2\sigma_g^2\right),
\end{aligned}
$$

where

$$
M_1 := 4a_1 + 6\left(\frac{L_{\Phi_0}}{2} + 4a_1 L_\theta^2\right)c_\lambda^2, \ M_2 := 18\left(\tau_2 + \tau_3\right)c_\lambda^3 + \tau_4, \ M_3 := 9\left(\tau_2 + \tau_3\right)c_\lambda^3 + \tau_4. \tag{48}
$$

Building upon the preceding discussions, we present the following lemma:

**Lemma F.9** (Descent in Lyapunov function $\mathcal{L}_{\text{se}}^k$). *Fix the number of communication rounds $K$. Define the Lyapunov function as*

$$
\begin{aligned}
\mathcal{L}_{se}^k := &\mathbb{E}\Phi_{\mu_k}(\bar{x}^k, \bar{y}^k) + a_1\mathbb{E}\|\bar{\theta}^k - \theta_\gamma^*(\bar{x}^k, \bar{y}^k)\|^2 \\
&+ \frac{a_2^k}{n}\sum_{i=1}^n\mathbb{E}\|x_i^k - \bar{x}^k\|^2 + \frac{a_3^k}{n}\sum_{i=1}^n\mathbb{E}\|y_i^k - \bar{y}^k\|^2 + \frac{a_4^k}{n}\sum_{i=1}^n\mathbb{E}\|\theta_i^k - \bar{\theta}^k\|^2,
\end{aligned} \tag{49}
$$

*where*

$$
a_1 := \frac{1}{2L_\theta}\sqrt{L_2^2 + \frac{1}{\gamma^2}}, \quad a_2^k := \tau_2\lambda_x^k, \quad a_3^k := \tau_3\lambda_y^k, \quad a_4^k := \tau_4\lambda_\theta^k, \tag{50}
$$

*with*

$$
\begin{aligned}
\tau_2 &:= 2\frac{\left(3\left(\mu_0 L_1^2 + 2L_2^2\right)c_\lambda + 4\left(\mu_0 L_1^2 + L_2^2\right)c_\lambda\right) + 2a_1\frac{6}{\eta}L_2^2}{c_\lambda(1-\rho)}, \\
\tau_3 &:= 2\frac{\left(3\left(\mu_0 L_1 + L_2^2\right)c_\lambda + 4\left(\mu_0 L_1^2 + L_2^2 + \frac{1}{\gamma^2}\right)c_\lambda\right) + 2a_1\frac{6}{\eta}\frac{1}{\gamma^2}}{c_\lambda(1-\rho)}, \\
\tau_4 &:= 2\frac{\left(3L_2^2 c_\lambda + \frac{4}{\gamma^2}c_\lambda\right) + 2a_1\frac{6}{\eta}\left(\frac{1}{\gamma^2} + L_2^2\right)}{c_\lambda(1-\rho)}.
\end{aligned} \tag{51}
$$

*Under Assumptions 3.1–3.4, let the learning rates as follows: $\lambda_\theta^k = c_\theta n^{1/2}K^{-1/2}$, $\lambda_x^k = \lambda_y^k = c_\lambda\lambda_\theta^k$, where $c_\lambda \leq \frac{\eta}{32a_1 L_\theta^2}$, $c_\theta$ are positive constants that ensure the learning rate satisfies the following:*

$$
\lambda_\theta^k \leq \frac{4}{3\eta}, \quad \lambda_\theta^k \leq \frac{4a_1 L_\theta^2}{\rho L_{\Phi_\infty}},
$$

$$
\lambda_\theta^k \leq \frac{\tau_2 c_\lambda(1-\rho)}{96L_2^2\left(2a_1 + 4\left(9\tau_5 c_\lambda^3 L_2^2 + 12\tau_6 c_\lambda^3\frac{1}{\gamma^2} + 9\tau_7\left(L_2^2 + \frac{1}{\gamma^2}\right)\right)\right)\frac{1}{1-\rho}}, \tag{52}
$$

$$
\lambda_\theta^k \leq \frac{\tau_3 c_\lambda(1-\rho)}{96\frac{1}{\gamma^2}2a_1\frac{1}{1-\rho}}, \quad \lambda_\theta^k \leq \frac{\tau_4 c_\lambda(1-\rho)}{96\left(L_2^2 + \frac{1}{\gamma^2}\right)2a_1\frac{1}{1-\rho}},
$$

where $\delta_k := \frac{\lambda_\theta^k \eta}{4(1 - \frac{\lambda_\theta^k \eta}{2})}$, $\rho := \frac{1}{\gamma} - L_2$. *Taking into account that $\mu_k$ in Algorithm 1 is a decaying sequence, we then obtain the following descent property:*

$$
\begin{aligned}
\mathcal{L}_{se}^{k+1} - \mathcal{L}_{se}^k \leq &- \frac{1}{8 c_\lambda \lambda_\theta^k} \left\| \mathbb{E}[\bar{x}^{k+1} - \bar{x}^k] \right\|^2 - \frac{1}{8 c_\lambda \lambda_\theta^k} \left\| \mathbb{E}[\bar{y}^{k+1} - \bar{y}^k] \right\|^2 \\
&- \frac{1}{16 \, a_1 L_\theta^2} \left( L_2^2 + \frac{1}{\gamma^2} \right) \lambda_\theta^k \, \mathbb{E} \left\| \bar{\theta}^k - \theta_\gamma^*(\bar{x}^k, \bar{y}^k) \right\|^2 - \frac{\tau_2 c_\lambda (1 - \rho)}{8} c_\lambda \lambda_\theta^k \Delta_x^k \\
&- \frac{\tau_3 c_\lambda (1 - \rho)}{8} c_\lambda \lambda_\theta^k \Delta_y^k - \frac{\tau_4 (1 - \rho)}{8} \lambda_\theta^k \Delta_\theta^k + \epsilon_{sto,\, se}^k + \epsilon_{dh}^k,
\end{aligned}
\tag{53}
$$

*where $\epsilon_{sto,\, se}^k$ and $\epsilon_{dh}^k$ denote the stochastic error and data heterogeneity error terms, respectively. Specifically, they are defined as:*

$$
\begin{aligned}
\epsilon_{sto,\, se}^k &:= (\lambda_\theta^k)^2 M_1 \cdot \frac{1}{n} \left( \mu_0^2 \delta_f^2 + 2 \delta_g^2 \right) + (\lambda_\theta^k)^3 M_2 \cdot \frac{1}{1 - \rho} \left( \mu_0^2 \delta_f^2 + 2 \delta_g^2 \right), \\
\epsilon_{dh}^k &:= (\lambda_\theta^k)^3 M_3 \cdot \frac{1}{1 - \rho} \left( \mu_0^2 \sigma_f^2 + 2 \sigma_g^2 \right),
\end{aligned}
\tag{54}
$$

*where $M_1$, $M_2$, and $M_3$ are constants defined in (48).*

*Remark* F.10. Since all terms on the right-hand side of the inequalities in the conditions 52 are non-negative, the step size $\lambda_\theta^k$ on the left-hand side can be chosen sufficiently small to satisfy the conditions in 52.

Based on Lemma F.9, we establish the convergence rate of the SUN-DSBO-SE algorithm. Note that here $p \in [0, 1/4)$ includes the case where $p \in (0, 1/4)$ in Theorem 3.5.

---

**Theorem F.11.** *Under Assumptions 3.1–3.4, let $\mu_k = \mu_0(k+1)^{-p}$, where $\mu_0 > 0$, $p \in [0, 1/4)$, and $\gamma \in (0, \frac{1}{2L_2})$. Choose learning rates as follows: $\lambda_\theta^k = c_\theta n^{1/2} K^{-1/2}$, $\lambda_x^k = \lambda_y^k = c_\lambda \lambda_\theta^k$, where $c_\theta, c_\lambda$ are positive constants that satisfy the conditions in Lemma F.9. Then, for SUN-DSBO-SE, we have*

$$
\begin{aligned}
\frac{1}{K} \sum_{k=0}^{K-1} \mathbb{E} \| \nabla \Psi_{\mu_k}(\bar{x}^k, \bar{y}^k) \|^2 =& \, \mathcal{O} \left( \frac{\mathcal{C}_1}{n^{1/2} K^{1/2}} \right) + \mathcal{O} \left( \frac{n M_2 (\mu_0 \delta_f^2 + \delta_g^2)}{(1 - \rho) K} \right) \\
&+ \mathcal{O} \left( \frac{n M_3 (\mu_0 \sigma_f^2 + \sigma_g^2)}{(1 - \rho) K} \right),
\end{aligned}
\tag{55}
$$

*and*

$$
\begin{aligned}
\frac{1}{K} \sum_{k=0}^{K-1} \Delta^k =& \, \mathcal{O} \left( \frac{\mathcal{C}_1}{K^{1/2}} \right) + \mathcal{O} \left( \frac{n^{3/2} M_2 (\mu_0 \delta_f^2 + \delta_g^2)}{(1 - \rho) K} \right) \\
&+ \mathcal{O} \left( \frac{n^{3/2} M_3 (\mu_0 \sigma_f^2 + \sigma_g^2)}{(1 - \rho) K} \right),
\end{aligned}
$$

*where*

$$
\mathcal{C}_1 := \mathcal{L}_{se}^0 + M_1 \left( \mu_0 \delta_f^2 + 2 \delta_g^2 \right), \quad \Delta^k := \frac{1}{n} \sum_{i=1}^n \mathbb{E} \left( \| x_i^k - \bar{x}^k \|^2 + \| y_i^k - \bar{y}^k \|^2 + \| \theta_i^k - \bar{\theta}^k \|^2 \right),
$$

*and $M_1$, $M_2$, and $M_3$ are defined in (48).*

---

*Proof.* Given that $\nabla_x \Psi_{\mu_k}(\bar{x}^k, \bar{y}^k) = \nabla_x \Phi_{\mu_k}(\bar{x}^k, \bar{y}^k), \nabla_y \Psi_{\mu_k}(\bar{x}^k, \bar{y}^k) = \nabla_y \Phi_{\mu_k}(\bar{x}^k, \bar{y}^k)$, we estimate the squared norm of the gradients as follows:

$$\mathbb{E}\|\nabla_x \Psi_{\mu_k}(\bar{x}^k, \bar{y}^k)\|^2 \le 2\mathbb{E}\|\nabla_x \Phi_{\mu_k}(\bar{x}^k, \bar{y}^k) - \tilde{\bar{d}}_x^k\|^2 + \frac{2}{(\lambda_x^k)^2}\left\|\mathbb{E}[\bar{x}^{k+1} - \bar{x}^k]\right\|^2,$$

$$\mathbb{E}\|\nabla_y \Psi_{\mu_k}(\bar{x}^k, \bar{y}^k)\|^2 \le 2\mathbb{E}\|\nabla_y \Phi_{\mu_k}(\bar{x}^k, \bar{y}^k) - \tilde{\bar{d}}_y^k\|^2 + \frac{2}{(\lambda_y^k)^2}\left\|\mathbb{E}[\bar{y}^{k+1} - \bar{y}^k]\right\|^2.$$

By utilizing the inequality mentioned above and (39) and (40) and performing left and right multiplication by $\lambda_\theta^k$, we establish the existence of $C_R > 0$ such that

$$
\begin{aligned}
\lambda_\theta^k \mathbb{E}\|\nabla \Psi_{\mu_k}(\bar{x}^k, \bar{y}^k)\|^2 \le C_R \Big( &-\frac{1}{8c_\lambda \lambda_\theta^k}\|\mathbb{E}[\bar{x}^{k+1} - \bar{x}^k]\|^2 - \frac{1}{8c_\lambda \lambda_\theta^k}\|\mathbb{E}[\bar{y}^{k+1} - \bar{y}^k]\|^2 \\
&-\frac{1}{16a_1 L_\theta^2}\Big(L_2^2 + \frac{1}{\gamma^2}\Big)\lambda_\theta^k \mathbb{E}\|\bar{\theta}^k - \theta_\gamma^*(\bar{x}^k, \bar{y}^k)\|^2 \\
&-\frac{\tau_2 c_\lambda(1-\rho)}{8}c_\lambda \lambda_\theta^k \Delta_x^k - \frac{\tau_3 c_\lambda(1-\rho)}{8}c_\lambda \lambda_\theta^k \Delta_y^k - \frac{\tau_4(1-\rho)}{8}\lambda_\theta^k \Delta_\theta^k \Big).
\end{aligned}
\tag{56}
$$

By using the (53),

$$
\begin{aligned}
&\lambda_\theta^k \mathbb{E}\|\nabla \Psi_{\mu_k}(\bar{x}^k, \bar{y}^k)\|^2 \\
&\le C_R\big(\mathcal{L}_{\text{se}}^k - \mathcal{L}_{\text{se}}^{k+1}\big) + \lambda_\theta^{k\,2} C_R M_1 \frac{1}{n}\Big(\mu_0^2 \delta_f^2 + 2\delta_g^2\Big) \\
&\quad + \lambda_\theta^{k\,3} M_2 C_R \frac{1}{1-\rho}\big(\mu_0^2 \delta_f^2 + 2\delta_g^2\big) + \lambda_\theta^{k\,3} M_3 C_R \frac{1}{1-\rho}\big(\mu_0^2 \sigma_f^2 + 2\sigma_g^2\big),
\end{aligned}
\tag{57}
$$

where $M_1$, $M_2$ and $M_3$ are defined in (48).

Summing both sides of the equality from $k = 0$ to $K - 1$ yields,

$$
\begin{aligned}
&\sum_{k=0}^{K-1} \lambda_\theta^k \mathbb{E}\|\nabla \Psi_{\mu_k}(\bar{x}^k, \bar{y}^k)\|^2 \\
&= \mathcal{O}\left(\sum_{k=0}^{K-1} \lambda_\theta^{k\,2} M_1 \frac{1}{n}\big(\mu_0 \delta_f^2 + \delta_g^2\big)\right) + \mathcal{O}\left(\sum_{k=0}^{K-1} \lambda_\theta^{k\,3} M_2 \frac{1}{1-\rho}\big(\mu_0 \delta_f^2 + \delta_g^2\big)\right) \\
&\quad + \mathcal{O}\left(\sum_{k=0}^{K-1} \lambda_\theta^{k\,3} M_3 \frac{1}{1-\rho}\big(\mu_0 \sigma_f^2 + \sigma_g^2\big)\right) + \mathcal{O}\big(\mathcal{L}_{\text{se}}^0\big).
\end{aligned}
\tag{58}
$$

If we take $\lambda_\theta^k = \mathcal{O}\big(\frac{1}{n^{1/2}K^{1/2}}\big)$, then we have

$$
\begin{aligned}
\frac{1}{K}\sum_{k=0}^{K-1} \mathbb{E}\|\nabla \Psi_{\mu_k}(\bar{x}^k, \bar{y}^k)\|^2 = &\mathcal{O}\left(\frac{\mathcal{L}_{\text{se}}^0 + M_1\big(\mu_0 \delta_f^2 + 2\delta_g^2\big)}{n^{1/2}K^{1/2}}\right) + \mathcal{O}\left(\frac{nM_2(\mu_0 \delta_f^2 + \delta_g^2)}{(1-\rho)K}\right) \\
&+ \mathcal{O}\left(\frac{nM_3(\mu_0 \sigma_f^2 + \sigma_g^2)}{(1-\rho)K}\right).
\end{aligned}
$$

Based on (56), and following a similar line of analysis as above, we obtain

$$
\begin{aligned}
\frac{1}{K}\sum_{k=0}^{K-1} \Delta^k = &\mathcal{O}\left(\frac{\mathcal{L}_{\text{se}}^0 + M_1\big(\mu_0 \delta_f^2 + 2\delta_g^2\big)}{K^{1/2}}\right) + \mathcal{O}\left(\frac{n^{3/2}M_2(\mu_0 \delta_f^2 + \delta_g^2)}{(1-\rho)K}\right) \\
&+ \mathcal{O}\left(\frac{n^{3/2}M_3(\mu_0 \sigma_f^2 + \sigma_g^2)}{(1-\rho)K}\right).
\end{aligned}
$$

$\square$

**Corollary F.12** (Complexity of **SUN-DSBO-SE**). *Under the same assumptions as in Theorem 3.5, the sample complexity of **SUN-DSBO-SE** is $\mathcal{O}\left(\epsilon^{-2}\right)$. Its transient iteration complexity is* $\mathcal{O}\left(\max\left\{n^3/(1-\rho)^4, n^3(\sigma_f^2 + \sigma_g^2)^2/(1-\rho)^4\right\}\right)$.

*Proof.* From Theorem F.11, when $K \gg n$, we have $\min_{0 \le k \le K-1} \mathbb{E}\|\nabla_x \Psi_{\mu_k}(\bar{x}^k, \bar{y}^k)\|^2 = \mathcal{O}\left(\frac{1}{K^{1/2}}\right) \le \epsilon$. Thus, to achieve an $\epsilon$-accurate solution, it suffices to take $K = \mathcal{O}\left(\epsilon^{-2}\right)$, which corresponds to the total sample complexity is $\mathcal{O}(\epsilon^{-2})$.

Moreover, according to Theorem F.11, if the term $\frac{1}{\sqrt{nK}}$ dominates the convergence rate, then

$$\frac{1}{n^{1/2}K^{1/2}} \gtrsim \max\left\{n \cdot \frac{1}{(1-\rho)^2 K}, n \cdot \frac{(\sigma_f^2 + \sigma_g^2)}{(1-\rho)^2 K}\right\}. \tag{59}$$

Simplifying (59) yields $K \gtrsim \max\left\{\frac{n^3}{(1-\rho)^4}, \frac{n^3(\sigma_f^2 + \sigma_g^2)^2}{(1-\rho)^4}\right\}$. Therefore, the transient iteration complexity is $\mathcal{O}\left(\frac{n^3}{(1-\rho)^4}\right)$. $\qquad\square$

# G    PROOFS OF SUN-DSBO-GT

## G.1    NOTATIONS

We first introduce some additional notations that will be used throughout the analysis:

$$\bar{t}_x^k := \frac{1}{n}\sum_{i=1}^n t_{x,i}^k, \quad \bar{t}_y^k := \frac{1}{n}\sum_{i=1}^n t_{y,i}^k, \quad \bar{t}_\theta^k := \frac{1}{n}\sum_{i=1}^n t_{\theta,i}^k, \tag{60}$$

where $t_{\theta,i}^k = D_{\theta,i}^k$, $t_{x,i}^k = D_{x,i}^k$, and $t_{y,i}^k = D_{y,i}^k$, where $D_{\theta,i}^k$, $D_{x,i}^k$ and $D_{y,i}^k$ in defined in Algorithm 2.

We define the conditional expectations as

$$\tilde{t}_{\theta,i}^k := \mathbb{E}[t_{\theta,i}^k], \quad \tilde{t}_{x,i}^k := \mathbb{E}[t_{x,i}^k], \quad \tilde{t}_{y,i}^k := \mathbb{E}[t_{y,i}^k], \tag{61}$$

with the initialization satisfying $\tilde{t}_{x,i}^{-1} = t_{x,i}^{-1}$, $\tilde{t}_{y,i}^{-1} = t_{y,i}^{-1}$, and $\tilde{t}_{\theta,i}^{-1} = t_{\theta,i}^{-1}$.

Similarly, we define the average expected values as

$$\widetilde{\bar{t}}_\theta^k := \frac{1}{n}\sum_{i=1}^n \tilde{t}_{\theta,i}^k, \quad \widetilde{\bar{t}}_x^k := \frac{1}{n}\sum_{i=1}^n \tilde{t}_{x,i}^k, \quad \widetilde{\bar{t}}_y^k := \frac{1}{n}\sum_{i=1}^n \tilde{t}_{y,i}^k. \tag{62}$$

Note that $\bar{d}_x^k = \bar{t}_x^k$, $\bar{d}_y^k = \bar{t}_y^k$, and $\bar{d}_\theta^k = \bar{t}_\theta^k$, so the averaged updates in Algorithm 2 follow

$$\bar{x}^{k+1} = \bar{x}^k - \lambda_x^k \bar{d}_x^k, \quad \bar{y}^{k+1} = \bar{y}^k - \lambda_y^k \bar{d}_y^k, \quad \bar{\theta}^{k+1} = \bar{\theta}^k - \lambda_\theta^k \bar{d}_\theta^k.$$

## G.2    PRELIMINARY LEMMAS

In order to further control the consensus errors among local variables $\theta_i^k$, $x_i^k$, and $y_i^k$, we state the following result that characterizes the evolution of their disagreement across iterations:

**Lemma G.1.** *The sequences $\{\theta_i^k\}$, $\{x_i^k\}$, and $\{y_i^k\}$ generated by Algorithm SUN-DSBO-GT satisfy the following recursion:*

$$\sum_{i=1}^n \mathbb{E}\|\theta_i^{k+1} - \bar{\theta}^{k+1}\|^2 \le \rho \sum_{i=1}^n \mathbb{E}\|\theta_i^k - \bar{\theta}^k\|^2 + \frac{\rho^2(\lambda_\theta^k)^2}{1-\rho}\sum_{i=1}^n \mathbb{E}\|t_{\theta,i}^k - \bar{t}_\theta^k\|^2,$$

$$\sum_{i=1}^n \mathbb{E}\|x_i^{k+1} - \bar{x}^{k+1}\|^2 \le \rho \sum_{i=1}^n \mathbb{E}\|x_i^k - \bar{x}^k\|^2 + \frac{\rho^2(\lambda_x^k)^2}{1-\rho}\sum_{i=1}^n \mathbb{E}\|t_{x,i}^k - \bar{t}_x^k\|^2,$$

$$\sum_{i=1}^n \mathbb{E}\|y_i^{k+1} - \bar{y}^{k+1}\|^2 \le \rho \sum_{i=1}^n \mathbb{E}\|y_i^k - \bar{y}^k\|^2 + \frac{\rho^2(\lambda_y^k)^2}{1-\rho}\sum_{i=1}^n \mathbb{E}\|t_{y,i}^k - \bar{t}_y^k\|^2.$$

*Proof.* First, we consider the term $\sum_{i=1}^{n} \mathbb{E}\left[\|x_i^{k+1} - \bar{x}^{k+1}\|^2\right]$. Similar to the analysis in Lemma A.3 in Dong et al. (2023), we have

$$
\sum_{i=1}^{n}\left[\|x_i^{k+1} - \bar{x}^{k+1}\|^2\right]
$$

$$
= \sum_{i=1}^{n}\left\|\sum_{j=1}^{n} w_{ij}(x_j^k - \lambda_x^k t_{x,j}^k) - \frac{1}{n}\sum_{s=1}^{n}\sum_{j=1}^{n} w_{sj}(x_j^k - \lambda_x^k t_{x,j}^k)\right\|^2
$$

$$
\leq \left(1 + \frac{1-\rho}{\rho}\right)\sum_{i=1}^{n}\left\|\sum_{j=1}^{n} w_{ij}x_j^k - \bar{x}^k\right\|^2 + \left(1 + \frac{\rho}{1-\rho}\right)(\lambda_x^k)^2\sum_{i=1}^{n}\left\|\sum_{j=1}^{n} w_{ij}(t_{x,j}^k - \bar{t}_x^k)\right\|^2
$$

$$
\leq \rho\sum_{i=1}^{n}\left[\|x_i^k - \bar{x}^k\|^2\right] + \frac{\rho^2(\lambda_x^k)^2}{1-\rho}\sum_{i=1}^{n}\left[\|t_{x,i}^k - \bar{t}_x^k\|^2\right].
$$

Combined with the analysis in Lemma F.5, we have

$$
\sum_{i=1}^{n}\mathbb{E}\|x_i^{k+1} - \bar{x}^{k+1}\|^2 \leq \rho\sum_{i=1}^{n}\mathbb{E}\|x_i^k - \bar{x}^k\|^2 + \frac{\rho^2\lambda_x^{k^2}}{1-\rho}\sum_{i=1}^{n}\mathbb{E}\|t_{x,i}^k - \bar{t}_x^k\|^2.
$$

Similarly, we obtain

$$
\sum_{i=1}^{n}\mathbb{E}\left[\|y_i^{k+1} - \bar{y}^{k+1}\|^2\right] \leq \rho\sum_{i=1}^{n}\mathbb{E}\left[\|y_i^k - \bar{y}^k\|^2\right] + \frac{\rho^2(\lambda_y^k)^2}{1-\rho}\sum_{i=1}^{n}\mathbb{E}\left[\|t_{y,i}^k - \bar{t}_y^k\|^2\right],
$$

$$
\sum_{i=1}^{n}\mathbb{E}\left[\|\theta_i^{k+1} - \bar{\theta}^{k+1}\|^2\right] \leq \rho\sum_{i=1}^{n}\mathbb{E}\left[\|\theta_i^k - \bar{\theta}^k\|^2\right] + \frac{\rho^2(\lambda_\theta^k)^2}{1-\rho}\sum_{i=1}^{n}\mathbb{E}\left[\|t_{\theta,i}^k - \bar{t}_\theta^k\|^2\right].
$$

$\square$

**Lemma G.2.** *(Dong et al., 2023) The sequences $\{x_i^k\}, \{y_i^k\}$ and $\{\theta_i^k\}$ generated by Algorithm SUN-DSBO-GT satisfies*

$$
\sum_{i=1}^{n}\mathbb{E}\|\theta_i^{k+1} - \theta_i^k\|^2 \leq 8\sum_{i=1}^{n}\mathbb{E}\|\theta_i^k - \bar{\theta}^k\|^2 + 4\lambda_\theta^{k^2}\sum_{i=1}^{n}\mathbb{E}\|t_{\theta,i}^k - \bar{t}_\theta^k\|^2 + 4n\lambda_\theta^{k^2}\mathbb{E}\|\bar{d}_\theta^k\|^2,
$$

$$
\sum_{i=1}^{n}\mathbb{E}\|x_i^{k+1} - x_i^k\|^2 \leq 8\sum_{i=1}^{n}\mathbb{E}\|x_i^k - \bar{x}^k\|^2 + 4\lambda_x^{k^2}\sum_{i=1}^{n}\mathbb{E}\|t_{x,i}^k - \bar{t}_x^k\|^2 + 4n\mathbb{E}\|\bar{x}^{k+1} - \bar{x}^k\|^2,
$$

$$
\sum_{i=1}^{n}\mathbb{E}\|y_i^{k+1} - y_i^k\|^2 \leq 8\sum_{i=1}^{n}\mathbb{E}\|y_i^k - \bar{y}^k\|^2 + 4\lambda_y^{k^2}\sum_{i=1}^{n}\mathbb{E}\|t_{y,i}^k - \bar{t}_y^k\|^2 + 4n\mathbb{E}\|\bar{y}^{k+1} - \bar{y}^k\|^2.
$$

## G.3 CONVERGENCE ANALYSIS

Following a similar line of analysis as in Lemma F.6, we can establish a same result.

**Lemma G.3** (Descent in $\Phi_{\mu_k}(x,y)$)**.** *Under Assumptions 3.1–3.3, $\gamma \in (0, \frac{1}{2L_2})$, the sequence of $(\bar{x}^k, \bar{y}^k, \bar{\theta}^k)$ generated by SUN-DSBO-GT satisfies*

$$
\mathbb{E}\Phi_{\mu_k}(\bar{x}^{k+1}, \bar{y}^{k+1}) - \mathbb{E}\Phi_{\mu_k}(\bar{x}^k, \bar{y}^k)
$$

$$
\leq -\frac{1}{2\lambda_x^k}\|\mathbb{E}[\bar{x}^{k+1} - \bar{x}^k]\|^2 + \frac{L_{\Phi_k}}{2}\mathbb{E}\|\bar{x}^{k+1} - \bar{x}^k\|^2 - \frac{1}{2\lambda_y^k}\|\mathbb{E}[\bar{y}^{k+1} - \bar{y}^k]\|^2 + \frac{L_{\Phi_k}}{2}\mathbb{E}\|\bar{y}^{k+1} - \bar{y}^k\|^2
$$

$$
+ \left(\lambda_x^k L_2^2 + \lambda_y^k\frac{1}{\gamma^2}\right)\mathbb{E}\|\bar{\theta}^k - \theta_\gamma^*(\bar{x}^k, \bar{y}^k)\|^2 + \left(3\lambda_x^k\left(\mu_k^2 L_1^2 + 2L_2^2\right) + 4\lambda_y^k\left(\mu_k^2 L_1^2 + L_2^2\right)\right)\mathbb{E}\Delta_x^k
$$

$$
+ \left(3\lambda_x^k L_2^2 + 4\lambda_y^k\left(\mu_k^2 L_1^2 + L_2^2 + \frac{1}{\gamma^2}\right)\right)\mathbb{E}\Delta_y^k + \left(3\lambda_x^k L_2^2 + \frac{4}{\gamma^2}\lambda_y^k\right)\mathbb{E}\Delta_\theta^k,
$$

*where $\Phi_{\mu_k}(x,y) := \mu_k\left(F(x,y) - \underline{F}\right) + G(x,y) - V_\gamma(x,y)$.*

**Lemma G.4.** *Under Assumptions 3.1 and 3.2, let $\bar{\theta}^k$ be the sequence generated by Algorithm SUN-DSBO-SE. Then the error $\|\bar{\theta}^k - \theta_\gamma^*(\bar{x}^k, \bar{y}^k)\|^2$ satisfies the following recursion:*

$$
\mathbb{E}\|\bar{\theta}^{k+1} - \theta_\gamma^*(\bar{x}^{k+1}, \bar{y}^{k+1})\|^2
$$

$$
\leq (1 + \delta_k)\Big(\mathbb{E}\|\bar{\theta}^k - \theta_\gamma^*(\bar{x}^k, \bar{y}^k)\|^2 + \lambda_\theta^{k\,2}\mathbb{E}\|\bar{t}_\theta^k\|^2 + \lambda_\theta^k \frac{6}{\eta}L_2^2 \mathbb{E}\Delta_x^k + \lambda_\theta^k \frac{6}{\eta}\Big(L_2^2 + \frac{1}{\gamma^2}\Big)\mathbb{E}\Delta_\theta^k
$$

$$
+ \lambda_\theta^k \frac{6}{\eta}\frac{1}{\gamma^2}\mathbb{E}\Delta_y^k - \rho\lambda_\theta^k\mathbb{E}\|\bar{\theta}^k - \theta_\gamma^*(\bar{x}^k, \bar{y}^k)\|^2\Big) \tag{63}
$$

$$
+ 2L_\theta^2\Big(1 + \frac{1}{\delta_k}\Big)\big(\mathbb{E}\|\bar{x}^{k+1} - \bar{x}^k\|^2 + \mathbb{E}\|\bar{y}^{k+1} - \bar{y}^k\|^2\big),
$$

*where $\delta_k > 0, \eta := \frac{1}{\gamma} - L_2$.*

Given that $\bar{t}_\theta^k = \bar{d}_\theta^k$, we invoke Lemma F.8 to obtain the following bound on the averaged sequence $\{\bar{t}_\theta^k\}$ generated by Algorithm SUN-DSBO-GT:

**Lemma G.5.** *The averaged sequence $\{\bar{t}_\theta^k\}$ of $\{t_{\theta,i}^k\}, i \in [n]$ satisfies the following inequality:*

$$
\|\bar{t}_\theta^k\|^2 \leq 2\frac{\sigma_g^2}{n} + 12L_2^2 \cdot \frac{1}{n}\sum_{i=1}^n \mathbb{E}\|x_i^k - \bar{x}^k\|^2 + 12\left(L_2^2 + \frac{1}{\gamma^2}\right) \cdot \frac{1}{n}\sum_{i=1}^n \mathbb{E}\|\theta_i^k - \bar{\theta}^k\|^2
$$

$$
+ 12 \cdot \frac{1}{\gamma^2} \cdot \frac{1}{n}\sum_{i=1}^n \mathbb{E}\|y_i^k - \bar{y}^k\|^2 + 4\left(L_2^2 + \frac{1}{\gamma^2}\right) \cdot \mathbb{E}\|\theta_\gamma^*(\bar{x}^k, \bar{y}^k) - \bar{\theta}^k\|^2.
$$

We define the Lyapunov function

$$
\mathcal{L}_{\text{gt}}^k := \mathbb{E}\Phi_{\mu_k}(\bar{x}^k, \bar{y}^k) + a_1\mathbb{E}\|\bar{\theta}^k - \theta_\gamma^*(\bar{x}^k, \bar{y}^k)\|^2
$$

$$
+ \frac{a_2^k}{n}\sum_{i=1}^n \mathbb{E}\|x_i^k - \bar{x}^k\|^2 + \frac{a_3^k}{n}\sum_{i=1}^n \mathbb{E}\|y_i^k - \bar{y}^k\|^2 + \frac{a_4^k}{n}\sum_{i=1}^n \mathbb{E}\|\theta_i^k - \bar{\theta}^k\|^2 \tag{64}
$$

$$
+ \frac{a_5^k}{n}\sum_{i=1}^n \mathbb{E}\|t_{x,i}^k - \bar{t}_x^k\|^2 + \frac{a_6^k}{n}\sum_{i=1}^n \mathbb{E}\|t_{y,i}^k - \bar{t}_y^k\|^2 + \frac{a_7^k}{n}\sum_{i=1}^n \mathbb{E}\|t_{\theta,i}^k - \bar{t}_\theta^k\|^2.
$$

Then we have

$$\mathcal{L}_{\text{gt}}^{k+1} - \mathcal{L}_{\text{gt}}^{k}$$

$$\leq -\frac{1}{2\lambda_x^k}\|\mathbb{E}[\bar{x}^{k+1} - \bar{x}^k]\|^2 - \frac{1}{2\lambda_y^k}\|\mathbb{E}[\bar{y}^{k+1} - \bar{y}^k]\|^2 + \frac{L_{\Phi_k}}{2}\mathbb{E}\Big(\|\bar{x}^{k+1} - \bar{x}^k\|^2 + \mathbb{E}\|\bar{y}^{k+1} - \bar{y}^k\|^2\Big)$$

$$+ \big(\lambda_x^k L_2^2 + \lambda_y^k \frac{1}{\gamma^2}\big)\mathbb{E}\|\bar{\theta}^k - \theta_\gamma^*(\bar{x}^k, \bar{y}^k)\|^2 + \big(3\lambda_x^k\big(\mu_k^2 L_1^2 + 2L_2^2\big) + 4\lambda_y^k\big(\mu_k^2 L_1^2 + L_2^2\big)\big)\mathbb{E}\Delta_x^k$$

$$+ \Big(3\lambda_x^k\big(\mu_k^2 L_1^2 + L_2^2\big) + 4\lambda_y^k\big(\mu_k^2 L_1^2 + L_2^2 + \frac{1}{\gamma^2}\big)\Big)\mathbb{E}\Delta_y^k + \Big(3\lambda_x^k L_2^2 + \frac{4}{\gamma^2}\lambda_y^k\Big)\mathbb{E}\Delta_\theta^k$$

$$+ a_1(1+\delta_k)\Big(\mathbb{E}\|\bar{\theta}^k - \theta_\gamma^*(\bar{x}^k, \bar{y}^k)\|^2 + \lambda_\theta^{k\,2}\mathbb{E}\|\bar{d}_\theta^k\|^2 + \lambda_\theta^k\frac{6}{\eta}L_2^2\mathbb{E}\Delta_x^k + \lambda_\theta^k\frac{6}{\eta}\big(L_2^2 + \frac{1}{\gamma^2}\big)\mathbb{E}\Delta_\theta^k$$

$$+ \lambda_\theta^k\frac{6}{\eta}\frac{1}{\gamma^2}\mathbb{E}\Delta_y^k - \eta\lambda_\theta^k\mathbb{E}\|\bar{\theta}^k - \theta_\gamma^*(\bar{x}^k, \bar{y}^k)\|^2\Big)$$

$$+ 2a_1 L_\theta^2\big(1 + \frac{1}{\delta_k}\big)\big(\mathbb{E}\|\bar{x}^{k+1} - \bar{x}^k\|^2 + \mathbb{E}\|\bar{y}^{k+1} - \bar{y}^k\|^2\big) - a_1\mathbb{E}\|\bar{\theta}^k - \theta_\gamma^*(\bar{x}^k, \bar{y}^k)\|^2$$

$$+ a_2^k(\rho - 1)\mathbb{E}\Delta_x^k + 3a_2^k\frac{\rho^2\lambda_x^{k\,2}}{1-\rho}\frac{1}{n}\sum_{i=1}^n \mathbb{E}\|t_{x,i}^k - \bar{t}_x^k\|^2$$

$$+ a_3^k(\rho - 1)\mathbb{E}\Delta_y^k + 3a_3^k\frac{\rho^2\lambda_y^{k\,2}}{1-\rho}\frac{1}{n}\sum_{i=1}^n \mathbb{E}\|t_{y,i}^k - \bar{t}_y^k\|^2$$

$$+ a_4^k(\rho - 1)\mathbb{E}\Delta_\theta^k + 3a_4^k\frac{\rho^2\lambda_\theta^{k\,2}}{1-\rho}\frac{1}{n}\sum_{i=1}^n \mathbb{E}\|t_{\theta,i}^k - \bar{t}_\theta^k\|^2$$

$$+ a_5^k(\rho - 1)\frac{1}{n}\sum_{i=1}^n \mathbb{E}\|t_{x,i}^k - \bar{t}_x^k\|^2 + a_5^k\frac{1}{1-\rho}\frac{1}{n}\sum_{i=1}^n \mathbb{E}\|d_{x,i,k+1}^{(\zeta_i^{k+1})} - d_{x,i,k}^{(\zeta_i^k)}\|^2$$

$$+ a_6^k(\rho - 1)\frac{1}{n}\sum_{i=1}^n \mathbb{E}\|t_{y,i}^k - \bar{t}_y^k\|^2 + a_6^k\frac{1}{1-\rho}\frac{1}{n}\sum_{i=1}^n \mathbb{E}\|d_{y,i,k+1}^{(\zeta_i^{k+1})} - d_{y,i,k}^{(\zeta_i^k)}\|^2$$

$$+ a_7^k(\rho - 1)\frac{1}{n}\sum_{i=1}^n \mathbb{E}\|t_{\theta,i}^k - \bar{t}_\theta^k\|^2 + a_7^k\frac{1}{1-\rho}\frac{1}{n}\sum_{i=1}^n \mathbb{E}\|d_{\theta,i,k+1}^{(\zeta_i^{k+1})} - d_{\theta,i,k}^{(\zeta_i^k)}\|^2.$$

For the coefficient of term $\|\mathbb{E}[\bar{x}^{k+1} - \bar{x}^k]\|^2$ is

$$\bigg(-\big(\frac{1}{2\lambda_y^k} - \frac{L_{\Phi_k}}{2}\big) + 2a_1 L_\theta^2\big(1 + \frac{1}{\delta_k}\big)$$

$$+ 4\Big(9a_5\lambda_x^{k\,2}\big(\mu_k^2 L_1^2 + L_2^2\big) + 12a_6\lambda_y^{k\,2}\big(\mu_k^2 L_1^2 + L_2^2 + \frac{1}{\gamma^2}\big) + 9a_7\lambda_\theta^{k\,2}\frac{1}{\gamma^2}\Big)\frac{1}{1-\rho}\bigg).$$

If we take $\lambda_\theta^k \leq \frac{4}{3\eta}$ such that $1 + \frac{1}{\delta_k} \leq 2$ and

$$\lambda_\theta^k \leq \frac{1-\rho}{64c_\lambda\Big(\frac{L_{\Phi\infty}}{2} + 4a_1 L_\theta^2 + 8\Big(9\tau_5\big(\mu_0^2 L_1^2 + 2L_2^2\big) + 12\tau_6\big(\mu_0^2 L_1^2 + L_2^2\big) + 9\tau_7 L_2^2\Big)\Big)}, \qquad (65)$$

the coefficient of term $\|\mathbb{E}[\bar{x}^{k+1} - \bar{x}^k]\|^2$ is $-\frac{1}{16c_\lambda\lambda_\theta^k}$.

For the coefficient of term $\|\mathbb{E}[\bar{y}^{k+1} - \bar{y}^k]\|^2$ is

$$-\big(\frac{1}{2\lambda_y^k} - \frac{L_{\Phi_k}}{2}\big) + 2a_1 L_\theta^2\big(1 + \frac{1}{\delta_k}\big)$$

$$+ 4\Big(9a_5\lambda_x^{k\,2}\big(\mu_k^2 L_1^2 + L_2^2\big) + 12a_6\lambda_y^{k\,2}\big(\mu_k^2 L_1^2 + L_2^2 + \frac{1}{\gamma^2}\big) + 9a_7\lambda_\theta^{k\,2}\frac{1}{\gamma^2}\Big)\frac{1}{1-\rho}.$$

If we take

$$\lambda_\theta^k \leq \frac{1-\rho}{64c_\lambda\left(\frac{L_{\Phi\infty}}{2} + 4a_1L_\theta^2 + 8\left(9\tau_5\left(\mu_0^2L_1^2 + L_2^2\right) + 12\tau_6\left(\mu_0^2L_1^2 + L_2^2 + \frac{1}{\gamma^2}\right) + 9\tau_7\frac{1}{\gamma^2}\right)\right)}, \quad (66)$$

the coefficient of term $\|\mathbb{E}[\bar{x}^{k+1} - \bar{x}^k]\|^2$ is $-\frac{1}{16c_\lambda\lambda_\theta^k}$.

For the coefficient of term $\mathbb{E}\|\bar{\theta}^k - \theta_\gamma^*(\bar{x}^k, \bar{y}^k)\|^2$ is

$$\lambda_x^k L_2^2 + \lambda_y^k\frac{1}{\gamma^2} + a_1\left((1+\delta_k)(1-\eta\lambda_\theta^k) - 1\right)$$

$$+4\lambda_\theta^{k^2}\left(L_2^2 + \frac{1}{\gamma^2}\right)\left(a_1(1+\delta_k) + 4\left(9\tau_5\lambda_x^{k^3}L_2^2 + 12\tau_6\lambda_y^{k^3}\frac{1}{\gamma^2} + 9\tau_7\lambda_\theta^{k^3}\left(L_2^2 + \frac{1}{\gamma^2}\right)\right)\frac{1}{1-\rho}\right).$$

If we take $a_1 := \frac{1}{2L_\theta}\sqrt{L_2^2 + \frac{1}{\gamma^2}}$, $\delta_k := \frac{\lambda_\theta^k\eta}{4\left(1 - \frac{\lambda_\theta^k\eta}{2}\right)}$, where $c_\lambda \leq \frac{\eta}{32a_1L_\theta^2}$ is a constant, $\eta := \frac{1}{\gamma} - L_2$,

$\lambda_\theta^k \leq \frac{4a_1L_\theta^2}{\rho L_{\Phi\infty}}$, and

$$\lambda_\theta^k \leq \frac{1}{32a_1L_\theta^2}\left(L_2^2 + \frac{1}{\gamma^2}\right)\frac{1-\rho}{16\left(L_2^2 + \frac{1}{\gamma^2}\right)4\left(9a_5L_2^2 + 12a_6\frac{1}{\gamma^2} + 9a_7\left(L_2^2 + \frac{1}{\gamma^2}\right)\right)}, \quad (67)$$

the coefficient of term $\mathbb{E}\|\bar{\theta}^k - \theta_\gamma^*(\bar{x}^k, \bar{y}^k)\|^2$ is $-\frac{1}{32a_1L_\theta^2}\left(L_2^2 + \frac{1}{\gamma^2}\right)\lambda_\theta^k$.

For the coefficient of term $\Delta_x^k$ is

$$\left(3\lambda_x^k\left(\mu_k^2L_1^2 + 2L_2^2\right) + 4\lambda_y^k\left(\mu_k^2L_1^2 + L_2^2\right)\right) + a_1(1+\delta_k)\lambda_\theta^k\frac{6}{\eta}L_2^2 + \tau_2\lambda_x^k(\rho - 1)$$

$$+ 8\left(9\tau_5\lambda_x^{k^3}\left(\mu_k^2L_1^2 + 2L_2^2\right) + 12\tau_6\lambda_y^{k^3}\left(\mu_k^2L_1^2 + L_2^2\right) + 9\tau_7\lambda_\theta^{k^3}L_2^2\right)$$

$$+ 12L_2^2\lambda_\theta^{k^2}\left(a_1(1+\delta_k) + 4\left(9\tau_5\lambda_x^{k^3}L_2^2 + 12\tau_6\lambda_y^{k^3}\frac{1}{\gamma^2} + 9\tau_7\lambda_\theta^{k^3}\left(L_2^2 + \frac{1}{\gamma^2}\right)\right)\frac{1}{1-\rho}\right).$$

If we take $\tau_2 := 2\frac{\left(3\left(\mu_0^2L_1^2 + 2L_2^2\right)c_\lambda + 4\left(\mu_0^2L_1^2 + L_2^2\right)c_\lambda\right) + 2a_1\frac{6}{\eta}L_2^2}{c_\lambda(1-\rho)}$ and

$$\lambda_\theta^{k^2} \leq \frac{\tau_2c_\lambda(1-\rho)}{64\left(9\tau_5c_\lambda^3\left(\mu_0^2L_1^2 + 2L_2^2\right) + 12\tau_6c_\lambda^3\left(\mu_0^2L_1^2 + L_2^2\right) + 9\tau_7L_2^2\right)}, \quad (68)$$

$$\lambda_\theta^k \leq \frac{\tau_2c_\lambda(1-\rho)}{96L_2^2\left(2a_1 + 4\left(9\tau_5c_\lambda^3L_2^2 + 12\tau_6c_\lambda^3\frac{1}{\gamma^2} + 9\tau_7\left(L_2^2 + \frac{1}{\gamma^2}\right)\right)\right)\frac{1}{1-\rho}}, \quad (69)$$

the coefficient of term $\Delta_x^k$ is $-\frac{\tau_2c_\lambda(1-\rho)}{8}\lambda_\theta^k$.

For the coefficient of term $\Delta_y^k$ is

$$\left(3\lambda_x^k\left(\mu_k^2L_1^2 + L_2^2\right) + 4\lambda_y^k\left(\mu_k^2L_1^2 + L_2^2 + \frac{1}{\gamma^2}\right)\right) + a_1(1+\delta_k)\lambda_\theta^k\frac{6}{\eta}\frac{1}{\gamma^2} + \tau_3\lambda_y^k(\rho - 1)$$

$$+ 8\left(9\tau_5\lambda_x^{k^3}\left(\mu_k^2L_1^2 + L_2^2\right) + 12\tau_6\lambda_y^{k^3}\left(\mu_k^2L_1^2 + L_2^2 + \frac{1}{\gamma^2}\right) + 9\tau_7\lambda_\theta^{k^3}\frac{1}{\gamma^2}\right)$$

$$+ 12\frac{1}{\gamma^2}\lambda_\theta^{k^2}\left(a_1(1+\delta_k) + 4\left(9a_5\lambda_x^{k^3}L_2^2 + 12a_6\lambda_y^{k^3}\frac{1}{\gamma^2} + 9a_7\lambda_\theta^{k^3}\left(L_2^2 + \frac{1}{\gamma^2}\right)\right)\frac{1}{1-\rho}\right).$$

If we take $\tau_3 := 2\dfrac{\left(3\left(\mu_0^2 L_1^2 + L_2^2\right)c_\lambda + 4\left(\mu_0^2 L_1^2 + L_2^2 + \frac{1}{\gamma^2}\right)c_\lambda\right) + 2a_1 \frac{6}{\eta}\frac{1}{\gamma^2}}{c_\lambda(1-\rho)}$ and

$$\lambda_\theta^{k\,2} \leq \frac{\tau_3 c_\lambda (1-\rho)}{64\left(9\tau_5 c_\lambda^3\left(\mu_0^2 L_1^2 + L_2^2\right) + 12\tau_6 c_\lambda^3\left(\mu_0^2 L_1^2 + L_2^2 + \frac{1}{\gamma^2}\right) + 9\tau_7 \frac{1}{\gamma^2}\right)} \tag{70}$$

$$\lambda_\theta^k \leq \frac{\tau_3 c_\lambda (1-\rho)}{96\frac{1}{\gamma^2}\left(2a_1 + 4\left(9\tau_5 c_\lambda^3 L_2^2 + 12\tau_6 c_\lambda^3 \frac{1}{\gamma^2} + 9\tau_7\left(L_2^2 + \frac{1}{\gamma^2}\right)\right)\right)\frac{1}{1-\rho}}, \tag{71}$$

the coefficient of term $\Delta_y^k$ is $-\frac{\tau_3 c_\lambda(1-\rho)}{8}\lambda_\theta^k$.

For the coefficient of term $\Delta_\theta^k$ is

$$\left(3\lambda_x^k L_2^2 + \frac{4}{\gamma^2}\lambda_y^k\right) + a_1(1+\delta_k)\lambda_\theta^k \frac{6}{\eta}\left(L_2^2 + \frac{1}{\gamma^2}\right) + \tau_4 \lambda_\theta^k(\rho-1)$$

$$+ 8\left(9\tau_5 \lambda_x^{k\,3} L_2^2 + 12\tau_6 \lambda_y^{k\,3}\frac{1}{\gamma^2} + 9\tau_7 \lambda_\theta^{k\,3}\left(L_2^2 + \frac{1}{\gamma^2}\right)\right)\frac{1}{1-\rho}$$

$$+ 12\left(\frac{1}{\gamma^2} + L_2^2\right)\lambda_\theta^{k\,2}\left(a_1(1+\delta_k) + 4\left(9\tau_5 \lambda_x^{k\,3} L_2^2 + 12\tau_6 \lambda_y^{k\,3}\frac{1}{\gamma^2} + 9\tau_7 \lambda_\theta^{k\,3}\left(L_2^2 + \frac{1}{\gamma^2}\right)\right)\frac{1}{1-\rho}\right).$$

If we take $\tau_4 := 2\dfrac{\left(3L_2^2 c_\lambda + \frac{4}{\gamma^2}c_\lambda\right) + 2a_1\frac{6}{\eta}\left(\frac{1}{\gamma^2} + L_2^2\right)}{c_\lambda(1-\rho)}$ and

$$\lambda_\theta^{k\,2} \leq \frac{\tau_4 c_\lambda(1-\rho)}{64\left(9\tau_5 c_\lambda^3 L_2^2 + 12\tau_6 c_\lambda^3 \frac{1}{\gamma^2} + 9\tau_7\left(L_2^2 \frac{1}{\gamma^2}\right)\right)} \tag{72}$$

$$\lambda_\theta^k \leq \frac{\tau_4 c_\lambda(1-\rho)}{96\left(L_2^2 + \frac{1}{\gamma^2}\right)\left(2a_1 + 4\left(9\tau_5 c_\lambda^3 L_2^2 + 12\tau_6 c_\lambda^3 \frac{1}{\gamma^2} + 9\tau_7\left(L_2^2 + \frac{1}{\gamma^2}\right)\right)\right)\frac{1}{1-\rho}}, \tag{73}$$

the coefficient of term $\Delta_\theta^k$ is $-\frac{\tau_4(1-\rho)}{8}\lambda_\theta^k$.

For the coefficient of term $\frac{1}{n}\sum_{i=1}^n \mathbb{E}\|t_{x,i}^k - \bar{t}_x^k\|^2$ is

$$\left(\tau_2 \frac{\rho^2}{1-\rho} + \tau_5(\rho-1) + 4\left(9\tau_5 \lambda_x^{k\,3}\left(\mu_k^2 L_1^2 + 2L_2^2\right)\right.\right.$$

$$\left.\left. + 12\tau_6 \lambda_y^{k\,3}\left(\mu_k^2 L_1^2 + L_2^2\right) + 9\tau_7 \lambda_\theta^{k\,3} L_2^2\right)\frac{1}{1-\rho}\right)\lambda_x^{k\,3}.$$

If we take $\tau_5 := \frac{\tau_2}{(\rho-1)^2}$ and

$$\lambda_\theta^{k\,3} \leq \frac{(\rho+1)\tau_2}{2c_\lambda^3(1-\rho)4\left(9\tau_5\left(\mu_0^2 L_1^2 + 2L_2^2\right) + 12\tau_6\left(\mu_0^2 L_1^2 + L_2^2\right) + 9\tau_7 L_2^2\right)\frac{1}{1-\rho}}, \tag{74}$$

the coefficient of term $\frac{1}{n}\sum_{i=1}^n \mathbb{E}\|t_{x,i}^k - \bar{t}_x^k\|^2$ is $-\frac{\tau_2(1+\rho)}{2(1-\rho)}\lambda_\theta^{k\,3} c_\lambda^3 \lambda_\theta^{k\,3}$.

For the coefficient of term $\frac{1}{n}\sum_{i=1}^n \mathbb{E}\|t_{y,i}^k - \bar{t}_y^k\|^2$ is

$$\left(\tau_3 \frac{\rho^2}{1-\rho} + \tau_6(\rho-1) + 4\left(9\tau_5 \lambda_x^{k\,3}\left(\mu_k^2 L_1^2 + L_2^2\right)\right.\right.$$

$$\left.\left. + 12\tau_6 \lambda_y^{k\,3}\left(\mu_k^2 L_1^2 + L_2^2 + \frac{1}{\gamma^2}\right) + 9\tau_7 \lambda_\theta^{k\,3}\frac{1}{\gamma^2}\right)\frac{1}{1-\rho}\right)\lambda_y^{k\,3}.$$

If we take $\tau_6 := \frac{\tau_3}{(\rho-1)^2}$ and

$$\lambda_\theta^{k3} \leq \frac{(\rho+1)\tau_3}{2c_\lambda^3(1-\rho)4\left(9\tau_5\left(\mu_0^2 L_1^2 + L_2^2\right) + 12\tau_6\left(\mu_0^2 L_1^2 + L_2^2 + \frac{1}{\gamma^2}\right) + 9\tau_7\frac{1}{\gamma^2}\right)\frac{1}{1-\rho}}, \tag{75}$$

the coefficient of term $\frac{1}{n}\sum_{i=1}^{n}\mathbb{E}\|t_{y,i}^k - \bar{t}_y^k\|^2$ is $-\frac{\tau_3 c_\lambda(1+\rho)}{2(1-\rho)}c_\lambda^3\lambda_\theta^{k3}$.

For the coefficient of term $\frac{1}{n}\sum_{i=1}^{n}\mathbb{E}\|t_{\theta,i}^k - \bar{t}_\theta^k\|^2$ is

$$\left(\tau_4\frac{\rho^2}{1-\rho} + \tau_7(\rho-1) + 4\left(9\tau_5\lambda_x^{k3}L_2^2 + 12\tau_6\lambda_y^{k3}\frac{1}{\gamma^2} + 9\tau_7\lambda_\theta^{k3}\left(L_2^2 + \frac{1}{\gamma^2}\right)\right)\frac{1}{1-\rho}\right)\lambda_\theta^{k3}.$$

If we take $\tau_7 := \frac{\tau_4}{(\rho-1)^2}$ and

$$\lambda_\theta^{k3} \leq \frac{(\rho+1)\tau_3}{2c_\lambda^3(1-\rho)4\left(9\tau_5 L_2^2 + 12\tau_6\frac{1}{\gamma^2} + 9\tau_7(L_2^2 + \frac{1}{\gamma^2})\right)\frac{1}{1-\rho}}, \tag{76}$$

the coefficient of term $\frac{1}{n}\sum_{i=1}^{n}\mathbb{E}\|t_{\theta,i}^k - \bar{t}_\theta^k\|^2$ is $-\frac{\tau_4(1+\rho)}{2(1-\rho)}\lambda_\theta^{k3}$.

Taking into account that $\mu_k$ in Algorithm 2 is a decaying sequence, we establish the following descent property of the Lyapunov function:

$$\begin{aligned}
\mathcal{L}_{\text{gt}}^{k+1} - \mathcal{L}_{\text{gt}}^k \leq\ & -\frac{1}{16c_\lambda\lambda_\theta^k}\left\|\mathbb{E}[\bar{x}^{k+1} - \bar{x}^k]\right\|^2 - \frac{1}{16c_\lambda\lambda_\theta^k}\left\|\mathbb{E}[\bar{y}^{k+1} - \bar{y}^k]\right\|^2 \\
& -\frac{1}{32a_1 L_\theta^2}\left(L_2^2 + \frac{1}{\gamma^2}\right)\lambda_\theta^k\mathbb{E}\left\|\bar{\theta}^k - \theta_\gamma^*(\bar{x}^k, \bar{y}^k)\right\|^2 \\
& -\frac{\tau_2 c_\lambda(1-\rho)}{8}c_\lambda\lambda_\theta^k\Delta_x^k - \frac{\tau_3 c_\lambda(1-\rho)}{8}c_\lambda\lambda_\theta^k\Delta_y^k - \frac{\tau_4(1-\rho)}{8}\lambda_\theta^k\Delta_\theta^k \\
& -\frac{\tau_2 c_\lambda(1+\rho)}{2(1-\rho)}c_\lambda^3(\lambda_\theta^k)^3\frac{1}{n}\sum_{i=1}^{n}\mathbb{E}\|t_{x,i}^k - \bar{t}_x^k\|^2 \\
& -\frac{\tau_3 c_\lambda(1+\rho)}{2(1-\rho)}c_\lambda^3(\lambda_\theta^k)^3\frac{1}{n}\sum_{i=1}^{n}\mathbb{E}\|t_{y,i}^k - \bar{t}_y^k\|^2 \\
& -\frac{\tau_4(1+\rho)}{2(1-\rho)}(\lambda_\theta^k)^3\frac{1}{n}\sum_{i=1}^{n}\mathbb{E}\|t_{\theta,i}^k - \bar{t}_\theta^k\|^2 \\
& +(\lambda_\theta^k)^2 M_4\frac{1}{n}\left(\mu_0^2\delta_f^2 + 2\delta_g^2\right) + (\lambda_\theta^k)^3 M_5\frac{1}{1-\rho}\left(\mu_0^2\delta_f^2 + 2\delta_g^2\right),
\end{aligned} \tag{77}$$

where $M_4$ and $M_5$ are defined :

$$\begin{aligned}
M_4 &:= 4a_1 + \left(\frac{L_{\Phi_0}}{2} + 4a_1 L_\theta^2\right)6c_\lambda^2, \\
M_5 &:= 8\left(9\tau_5 c_\lambda^3 L_2^2 + 12\tau_6 c_\lambda^3\frac{1}{\gamma^2} + \tau_7(L_2^2 + \frac{1}{\gamma^2})\right) + 6(\tau_5 c_\lambda^3 + \tau_6 c_\lambda^3 + \tau_7).
\end{aligned} \tag{78}$$

Building upon the preceding discussions, we present the following lemma:

**Lemma G.6** (Descent in Lyapunov function $\mathcal{L}_{\text{gt}}^k$). *Fix the number of communication rounds $K$. Define the Lyapunov function*

$$\begin{aligned}
\mathcal{L}_{gt}^k =\ & \mathbb{E}\left[\Phi_{\mu_k}(\bar{x}^k, \bar{y}^k)\right] + a_1\mathbb{E}\left\|\bar{\theta}^k - \theta_\gamma^*(\bar{x}^k, \bar{y}^k)\right\|^2 \\
& + \frac{a_2^k}{n}\sum_{i=1}^{n}\mathbb{E}\|x_i^k - \bar{x}^k\|^2 + \frac{a_3^k}{n}\sum_{i=1}^{n}\mathbb{E}\|y_i^k - \bar{y}^k\|^2 + \frac{a_4^k}{n}\sum_{i=1}^{n}\mathbb{E}\|\theta_i^k - \bar{\theta}^k\|^2 \\
& + \frac{a_5^k}{n}\sum_{i=1}^{n}\mathbb{E}\|t_{x,i}^k - \bar{t}_x^k\|^2 + \frac{a_6^k}{n}\sum_{i=1}^{n}\mathbb{E}\|t_{y,i}^k - \bar{t}_y^k\|^2 + \frac{a_7^k}{n}\sum_{i=1}^{n}\mathbb{E}\|t_{\theta,i}^k - \bar{t}_\theta^k\|^2,
\end{aligned} \tag{79}$$

*where*

$$a_1 = \frac{1}{2L_\theta}\sqrt{L_2^2 + \frac{1}{\gamma^2}}, \quad a_2^k = \tau_2\lambda_x^k, \quad a_3^k = \tau_3\lambda_y^k, \quad a_4^k = \tau_4\lambda_\theta^k,$$

$$a_5^k = \tau_5\lambda_x^{k\,2}, \qquad\qquad a_6^k = \tau_6\lambda_y^{k\,2}, \quad a_7^k = \tau_7\lambda_\theta^{k\,2}.$$

*The constants $\tau_j$ for $j = 2, \ldots, 7$ are defined by*

$$\tau_2 := 2\frac{\left(3(\mu_0^2L_1^2 + 2L_2^2)c_\lambda + 4(\mu_0^2L_1^2 + L_2^2)c_\lambda\right) + 2a_1\frac{6}{\eta}L_2^2}{c_\lambda(1-\rho)},$$

$$\tau_3 := 2\frac{\left(3(\mu_0^2L_1^2 + L_2^2)c_\lambda + 4(\mu_0^2L_1^2 + L_2^2 + \frac{1}{\gamma^2})c_\lambda\right) + 2a_1\frac{6}{\eta}\frac{1}{\gamma^2}}{c_\lambda(1-\rho)},$$

$$\tau_4 := 2\frac{\left(3L_2^2c_\lambda + \frac{4}{\gamma^2}c_\lambda\right) + 2a_1\frac{6}{\eta}(L_2^2 + \frac{1}{\gamma^2})}{c_\lambda(1-\rho)},$$

$$\tau_5 := \frac{\tau_2}{(1-\rho)^2}, \quad \tau_6 := \frac{\tau_3}{(1-\rho)^2}, \quad \tau_7 := \frac{\tau_4}{(1-\rho)^2}.$$

(80)

*Under Assumptions 3.1–3.3, let the learning rates be defined as follows: $\lambda_\theta^k = c_\theta n^{1/2} K^{-1/2}$, $\lambda_x^k = \lambda_y^k = c_\lambda\lambda_\theta^k$, where $c_\lambda \leq \frac{\eta}{32a_1L_\theta^2}$, and $c_\theta$ are positive constants that ensure the learning rates satisfy the following:*

$$\lambda_\theta^k \leq \min\left\{\frac{4}{3\eta}, \frac{4a_1L_\theta^2}{\rho L_{\Phi_\infty}}\right\}$$

(81)

*and satisfy conditions (65)–(76), where $\delta_k = \frac{\lambda_\theta^k\eta}{4\left(1 - \frac{\lambda_\theta^k\eta}{2}\right)}$ and $\rho = \frac{1}{\gamma} - L_2$. Then the following descent property holds:*

$$\mathcal{L}_{gt}^{k+1} - \mathcal{L}_{gt}^k \leq -\frac{1}{16c_\lambda\lambda_\theta^k}\left\|\mathbb{E}[\bar{x}^{k+1} - \bar{x}^k]\right\|^2 - \frac{1}{16c_\lambda\lambda_\theta^k}\left\|\mathbb{E}[\bar{y}^{k+1} - \bar{y}^k]\right\|^2$$

$$- \frac{1}{32a_1L_\theta^2}\left(L_2^2 + \frac{1}{\gamma^2}\right)\lambda_\theta^k\mathbb{E}\left\|\bar{\theta}^k - \theta_\gamma^*(\bar{x}^k, \bar{y}^k)\right\|^2$$

$$- \frac{\tau_2c_\lambda(1-\rho)}{8}c_\lambda\lambda_\theta^k\Delta_x^k - \frac{\tau_3c_\lambda(1-\rho)}{8}c_\lambda\lambda_\theta^k\Delta_y^k - \frac{\tau_4(1-\rho)}{8}\lambda_\theta^k\Delta_\theta^k$$

$$- \frac{\tau_2c_\lambda(1+\rho)}{2(1-\rho)}c_\lambda^3(\lambda_\theta^k)^3\frac{1}{n}\sum_{i=1}^n\mathbb{E}\|t_{x,i}^k - \bar{t}_x^k\|^2$$

$$- \frac{\tau_3c_\lambda(1+\rho)}{2(1-\rho)}c_\lambda^3(\lambda_\theta^k)^3\frac{1}{n}\sum_{i=1}^n\mathbb{E}\|t_{y,i}^k - \bar{t}_y^k\|^2$$

$$- \frac{\tau_4(1+\rho)}{2(1-\rho)}(\lambda_\theta^k)^3\frac{1}{n}\sum_{i=1}^n\mathbb{E}\|t_{\theta,i}^k - \bar{t}_\theta^k\|^2 + \epsilon_{sto,gt}^k,$$

(82)

*where $\epsilon_{sto,\,gt}^k$ denotes the stochastic error terms. Specifically, they are defined as:*

$$\epsilon_{sto,gt}^k := (\lambda_\theta^k)^2 M_4 \cdot \frac{1}{n}\left(\mu_0^2\delta_f^2 + 2\delta_g^2\right) + (\lambda_\theta^k)^3 M_5 \cdot \frac{1}{1-\rho}\left(\mu_0^2\delta_f^2 + 2\delta_g^2\right),$$

(83)

*where $M_4$ and $M_5$ are constants defined in (78).*

*Remark G.7.* Since all terms on the right-hand side of the inequalities in the conditions (65)–(76) are non-negative, the step size $\lambda_\theta^k$ on the left-hand side can be chosen sufficiently small to satisfy the conditions (65)–(76).

Based on Lemma G.6, we establish the convergence rate of the SUN-DSBO-GT algorithm. Note that here $p \in [0, 1/4)$ includes the case where $p \in (0, 1/4)$ in Theorem 3.7.

**Theorem G.8.** *Under Assumptions 3.1–3.3, let $\mu_k = \mu_0(k+1)^{-p}$, where $\mu_0 > 0$, $p \in [0, 1/4)$, and $\gamma \in (0, \frac{1}{2L_2})$. Choose learning rates as follows: $\lambda_\theta^k = c_\theta n^{1/2} K^{-1/2}$, $\lambda_x^k = \lambda_y^k = c_\lambda \lambda_\theta^k$, where $c_\theta, c_\lambda$ are positive constants that satisfy the conditions in Lemma G.6. Then, for SUN-DSBO-GT, we have*

$$\frac{1}{K} \sum_{k=0}^{K-1} \mathbb{E}\|\nabla \Psi_{\mu_k}(\bar{x}^k, \bar{y}^k)\|^2 = \mathcal{O}\left(\frac{\mathcal{C}_2}{n^{1/2} K^{1/2}}\right) + \mathcal{O}\left(\frac{n M_5 (\mu_0 \delta_f^2 + 2\delta_g^2)}{(1-\rho)K}\right),$$

*and*

$$\frac{1}{K} \sum_{k=0}^{K-1} \Delta^k = \mathcal{O}\left(\frac{\mathcal{C}_2}{K^{1/2}}\right) + \mathcal{O}\left(\frac{n^{3/2} M_5 (\mu_0 \delta_f^2 + 2\delta_g^2)}{(1-\rho)K}\right),$$

*where*

$$\mathcal{C}_2 := \mathcal{L}_{gt}^0 + M_4 \left(\mu_0 \delta_f^2 + 2\delta_g^2\right), \quad \Delta^k := \frac{1}{n} \sum_{i=1}^{n} \mathbb{E}\left(\|x_i^k - \bar{x}^k\|^2 + \|y_i^k - \bar{y}^k\|^2 + \|\theta_i^k - \bar{\theta}^k\|^2\right),$$

*and $M_4$ and $M_5$ are defined in (78).*

*Proof.* Based on the descent inequality in (82), the rest of the proof follows similarly to the argument in Theorem F.11 and is omitted here. ∎

**Corollary G.9** (Complexity of SUN-DSBO-GT). *Under the same assumptions as in Theorem 3.7, the sample complexity of SUN-DSBO-GT is $\mathcal{O}\left(\epsilon^{-2}\right)$. Its transient iteration complexity is $\mathcal{O}\left(\frac{n^3}{(1-\rho)^8}\right)$.*

*Proof.* The proof follows similarly to that of Corollary F.12 and is omitted for brevity. ∎

# H   MORE DETAILS OF SUN-DSBO

In Section 2.2, we present SUN-DSBO in its general form, wherein each local agent repeatedly performs the following steps:

(I) **Stochastic Gradient Computation:** Each agent $i$ computes unbiased or biased stochastic estimators $\hat{D}_{\theta,i}^k$, $\hat{D}_{x,i}^k$, and $\hat{D}_{y,i}^k$ for the descent directions $\tilde{D}_{\theta,i}^k$, $\tilde{D}_{x,i}^k$, and $\tilde{D}_{y,i}^k$, as derived from the min-max reformulation (5). These estimators are calculated using either vanilla mini-batch gradients or advanced techniques that incorporate acceleration and variance reduction:

$$\tilde{D}_{\theta,i}^k = \nabla_y g_i(x_i^k, \theta_i^k) + \frac{1}{\gamma}(\theta_i^k - y_i^k), \tag{84a}$$

$$\tilde{D}_{x,i}^k = \mu_k \nabla_x f_i(x_i^k, y_i^k) + \nabla_x g_i(x_i^k, y_i^k) - \nabla_x g_i(x_i^k, \theta_i^k), \tag{84b}$$

$$\tilde{D}_{y,i}^k = \mu_k \nabla_y f_i(x_i^k, y_i^k) + \nabla_y g_i(x_i^k, y_i^k) - \frac{1}{\gamma}(y_i^k - \theta_i^k). \tag{84c}$$

(II) **Gradient Estimator Update:** Communicate with neighbors and update the gradient estimators $\hat{D}_{\theta,i}^k$, $\hat{D}_{x,i}^k$, and $\hat{D}_{y,i}^k$ to $D_{\theta,i}^k$, $D_{x,i}^k$ and $D_{y,i}^k$ using decentralized techniques such as GT, EXTRA, and Exact-Diffusion (ED), as well as mixing strategies (Zhu et al., 2024).

(III) **Local Variable Update:** Communicate with neighbors and update the dual and primal variables using decentralized techniques, such as GT:

$$(\theta_i^{k+1}, x_i^{k+1}, y_i^{k+1}) = \sum_{j=1}^{n} w_{ij} \left(\theta_j^k - \lambda_\theta^k D_{\theta,j}^k, x_j^k - \lambda_x^k D_{x,j}^k, y_j^k - \lambda_y^k D_{y,j}^k\right). \tag{85}$$

Although we primarily focus on two specific instances of SUN-DSBO (SUN-DSBO-SE 1 and SUN-DSBO-GT 2), additional variants are provided in Table 14. Furthermore, to reduce the per-round

Table 14: More specific instances of SUN-DSBO. Here, $\Diamond \in \{x, y, \theta\}$ denotes the variable being tracked, and $(w_{ij})$ represents the weighting matrix of the communication network.

| **Step I: Stochastic Gradient Computation** | |
|---|---|
| vanilla mini-batch | sample $\xi_{g,i}^k$ and $\xi_{f,i}^k$, and compute $\hat{D}_{\theta,i}^k$, $\hat{D}_{x,i}^k$, and $\hat{D}_{y,i}^k$ using (8) |
| momentum | sample and compute $\breve{D}_{\Diamond,i}^k$ using the vanilla mini-batch approach in (8); and update $\hat{D}_{\Diamond,i}^k = (1 - \rho_{\Diamond,i}^k)\hat{D}_{\Diamond,i}^{k-1} + \rho_{\Diamond,i}^k \breve{D}_{\Diamond,i}^{k-1}$ with coefficient $\rho_{\Diamond,i}^k$ |
| STORM | sample and compute $\breve{D}_{\Diamond,i}^k$ using the vanilla mini-batch approach in (8); and update $\hat{D}_{\Diamond,i}^k = (1 - \rho_{\Diamond,i}^k)\left(\hat{D}_{\Diamond,i}^{k-1} + \breve{D}_{\Diamond,i}^k - \breve{D}_{\Diamond,i}^{k-1}\right) + \rho_{\Diamond,i}^k \breve{D}_{\Diamond,i}^k$ with coefficient $\rho_{\Diamond,i}^k$ |
| **Step II: Gradient Estimator Update** | |
| ATC-GT (Xu et al., 2015) | $D_{\Diamond,i}^k = \sum_{j=1}^n w_{ij}\left(D_{\Diamond,j}^{k-1} + \hat{D}_{\Diamond,j}^k - \hat{D}_{\Diamond,j}^{k-1}\right)$ |
| Semi-ATC-GT (Lorenzo & Scutari, 2016) | $D_{\Diamond,i}^k = \sum_{j=1}^n w_{ij}\left(D_{\Diamond,j}^{k-1} + \hat{D}_{\Diamond,j}^k - \hat{D}_{\Diamond,j}^{k-1}\right)$ |
| Non-ATC-GT (Nedic et al., 2017) | $D_{\Diamond,i}^k = \sum_{j=1}^n w_{ij} D_{\Diamond,j}^{k-1} + \hat{D}_{\Diamond,i}^k - \hat{D}_{\Diamond,i}^{k-1}$ |
| **Step III: Local Variable Update** | |
| ATC-GT | $\Diamond_i^{k+1} = \sum_{j=1}^n w_{ij}\left(\Diamond_j^k - \lambda_\Diamond^k D_{\Diamond,j}^k\right)$ |
| Semi-ATC-GT | $\Diamond_i^{k+1} = \sum_{j=1}^n w_{ij}\Diamond_j^k - \lambda_\Diamond^k D_{\Diamond,i}^k$ |
| Non-ATC-GT | $\Diamond_i^{k+1} = \sum_{j=1}^n w_{ij}\Diamond_j^k - \lambda_\Diamond^k D_{\Diamond,i}^k$ |

communication cost, SUN-DSBO offers additional flexibility beyond the variants presented in Table 14. We remark that

**(a) Step II is not strictly necessary.** For example, in SUN-DSBO-SE 1, each local agent communicates with its neighbors and updates both the dual and primal variables directly using the stochastic estimators $\hat{D}_{\theta,i}^k$, $\hat{D}_{x,i}^k$, and $\hat{D}_{y,i}^k$ computed in Step I, thereby skipping Step II. Consequently, this reduces the per-round communication cost. Moreover, in Step III, a non-ATC decentralized gradient descent (DGD)-type update can also be employed:

$$\text{Non-ATC-DGD:} \quad \Diamond_i^{k+1} = \sum_{j=1}^n w_{ij}\Diamond_j^k - \lambda_\Diamond^k D_{\Diamond,i}^k.$$

**(b) Steps II and III can be merged.** To reduce the per-round communication cost, the gradient estimation and local variable updates can be performed jointly in a single communication round using efficient decentralized techniques such as:

$$\text{EXTRA (Shi et al., 2015):} \quad \Diamond_i^{k+1} = \Diamond_i^k + \sum_{j=1}^n w_{ij}\Diamond_j^k - \sum_{j=1}^n \tilde{w}_{ij}\Diamond_j^{k-1} - \lambda_\Diamond^k\left(\hat{D}_{\Diamond,i}^k - \hat{D}_{\Diamond,i}^{k-1}\right),$$

$$\text{where } (\tilde{w}_{ij}) \text{ is a doubly stochastic matrix.}$$

$$\text{ED (Yuan et al., 2018; 2020):} \quad \Diamond_i^{k+1} = \sum_{j=1}^n w_{ij}\left(2\Diamond_j^k - \Diamond_j^{k-1} - \lambda_\Diamond^k\left(\hat{D}_{\Diamond,j}^k - \hat{D}_{\Diamond,j}^{k-1}\right)\right).$$

