# OpenReview forum: "SUN-DSBO: A Structured Unified Framework for Nonconvex Decentralized Stochastic Bilevel Optimization"
_ICLR.cc/2026/Conference — ICLR 2026 Conference Withdrawn Submission_

### Official Review · Reviewer_Rt4P · 2025-10-22

**Soundness:** 3
**Presentation:** 3
**Contribution:** 2
**Rating:** 6
**Confidence:** 3

**Summary:**

This paper proposes SUN-DSBO, a decentralized stochastic bilevel optimization framework built upon the Moreau envelope, based penalty method (Liu et al., 2024). The authors reformulate the penalty function as a nonconvex–strongly concave min–max problem and develop decentralized algorithms with convergence guarantees.

**Strengths:**

1. The unified framework idea is neat and general.
2. Writing is clear and easy to read.

**Weaknesses:**

1.  In line.163, this paper claims that the constraint   $G(x,y) - V_\gamma(x,y) \le 0 $  is equivalent to enforcing $ \nabla_y G(x,y) = 0 $ under mild conditions.  This is not true in general unless $G$ satisfies convexity or the Polyak–Łojasiewicz (PL) condition.  The equivalence only holds approximately.
I believe the authors are aware that the equivalence only holds under convexity or PL assumptions, yet they still claim to “handle general nonconvex lower-level objectives.” This is not technically incorrect but conceptually misleading, the proposed method actually optimizes a surrogate relaxation rather than the original nonconvex bilevel problem.

2. The empirical evaluation measures performance while optimizing the surrogate objective  $\Psi_{\mu}(x,y) = \mu F(x,y) + G(x,y)-V_{\gamma}(x,y)$, not the original bilevel constraints.
There is no diagnostic of how well the solutions satisfy $G(x,y) - V_{\gamma}(x,y) \le 0$ or approximate the true lower-level stationarity/KKT conditions as $\mu \to 0$.
Hence, the experiments demonstrate practical effectiveness of the surrogate method but do not validate the claimed equivalence to the original nonconvex bilevel problem.


3. The key building blocks of SUN-DSBO, gradient tracking for decentralized communication and the Moreau-envelope–based penalty for smoothing the lower-level objective are not novel. The paper does not introduce any new mechanism that fundamentally changes or extends these tools; it merely combines them in a straightforward way.Consequently, the “structured unified framework” is more of an incremental integration of existing methods rather than a conceptually new algorithmic design.


4. Typo: (i) In Eq.(39), what is $t$ is $x_i^{t,k}$? (ii) The paper writes $\mathcal{F}_0 :={ \Omega, \phi },$, is $\phi $ from eq.(26)? Or do you mean empty set?

**Questions:**

N/A

---

> ### Author Response · Authors · 2025-11-22
> **Response to Weaknesses (1)**
>
> We thank reviewer Rt4P for the detailed review, constructive suggestions, and helpful comments on both the theoretical comparison and empirical evaluation. We appreciate your careful reading and insightful feedback.
>
> > W1. In line.163, this paper claims that the constraint $G(x,y)-V_\gamma(x,y)\leq 0$   is equivalent to enforcing $\nabla_{y} G(x,y)=0$ under mild conditions. This is not true in general unless G  satisfies convexity or the Polyak–Łojasiewicz (PL) condition. The equivalence only holds approximately. I believe the authors are aware that the equivalence only holds under convexity or PL assumptions, yet they still claim to “handle general nonconvex lower-level objectives.” This is not technically incorrect but conceptually misleading, the proposed method actually optimizes a surrogate relaxation rather than the original nonconvex bilevel problem.
>
> **Response to W1:**
>
> To reply to your comment, we would first like to clarify the equivalence statement in line 163. Under the conditions that $G(x,\cdot)$ is $L_2$–smooth for any $x$ and $\gamma \in (0, 1/(2L_2))$, the constraint $G(x,y) - V_\gamma(x,y) \le 0$ becomes equivalent to $\nabla_y G(x,y) = 0$. This result follows from the proof of [1, Theorem A.1]. As noted in **lines 162–163** of the original (revised) version, under these conditions the reformulated problem (2) is equivalent to
> $ \min_{x,y} F(x,y)\ \text{s.t.}\ \nabla_y G(x,y)=0. \quad (*) $
>
> Regarding **convexity** or the Polyak–Łojasiewicz (PL) condition, our discussion in **lines 163–165** of the original (revised) version refers to a *different* aspect: such conditions could ensure that the problem $(*)$ is further equivalent to the original bilevel problem (1).
>
> > W2. The empirical evaluation measures performance while optimizing the surrogate objective $\mu F(x,y)+ G(x,y)-V_\gamma(x,y)$, not the original bilevel constraints. There is no diagnostic of how well the solutions satisfy  $G(x,y)-V_\gamma(x,y)\leq 0$ or approximate the true lower-level stationarity/KKT conditions as $\mu \rightarrow 0$. Hence, the experiments demonstrate practical effectiveness of the surrogate method but do not validate the claimed equivalence to the original nonconvex bilevel problem.
>
> **Response to W2:**
> First, we would like to note that, as discussed in **Lines 162–166** of the original (revised) version, under either convexity or the PL condition of the lower-level objective, problem (2) becomes equivalent to the original bilevel problem (1). Motivated by this observation, we designed an experiment where the lower-level objective is **merely convex** to empirically show that the surrogate objective
> $\mu F(x,y) + G(x,y) - V_\gamma(x,y)$
> progressively approaches the solution of the original problem.
>
> This numerical verification is included in **Figure 15** of Appendix B.3 (page 24), titled *“Synthetic numerical verification on lower-level merely convex case”* in the revised version. Since the lower-level problem admits a unique solution, we are able to directly measure the distance between the iterates and the true solution to assess solution quality.
> The experimental results indicate that both SUN-DSBO-GT and SUN-DSBO-SE consistently move closer to this unique solution.

---

> ### Author Response · Authors · 2025-11-22
> **Response to Weaknesses (2)**
>
> > W3. The key building blocks of SUN-DSBO, gradient tracking for decentralized communication and the Moreau-envelope–based penalty for smoothing the lower-level objective are not novel. The paper does not introduce any new mechanism that fundamentally changes or extends these tools; it merely combines them in a straightforward way.Consequently, the “structured unified framework” is more of an incremental integration of existing methods rather than a conceptually new algorithmic design.
>
> **Response to W3:**
>
> Thank you for your concern, which gives us the opportunity to clarify the innovations of this paper from both the algorithmic design and theoretical analysis perspectives.
>
> From the algorithm-design perspective, although introducing the Moreau-envelope penalty method to decentralized bilevel optimization (DSBO) is conceptually natural, **no prior work** has explored this direction or verified that it works well for DSBO, at least experimentally. As stated in line 167 of the original (revised) version, our method does not simply combine the Moreau-envelope–based penalty with decentralized techniques in a straightforward manner. Specifically, we reformulate problem (4) into the min–max structure in problem (5) and then construct a synchronized variable-updating framework based on this structure. In our opinion, this design viewpoint is simpler than that commonly used in single-agent bilevel optimization with the Moreau-envelope penalty; see, for example, [1, Algorithm 1] and [2, Algorithm 1]. A direct combination of these single-agent techniques with decentralized communication may require additional communication rounds to ensure convergence, since each variable update would need a separate communication step to refresh gradient information and maintain a correct descent direction.
>
> From the theoretical-analysis perspective, several aspects differ substantially from the single-agent work with the Moreau-envelope penalty. First, unlike the aforementioned works, our algorithm must account for the impact of **stochastic** gradient estimates. Even in the single-agent setting, the stochastic version of the Moreau-envelope penalty method remains underexplored, independent of the update mechanism.
> Second, in contrast to existing centralized bilevel algorithms that incorporate GT or other heterogeneity-correction techniques—where assumptions such as bounded gradient norms or gradient similarity are typically required—**our SUN-DSBO-GT algorithm does not rely on these assumptions**, as summarized in Table 1 in the appendix of the original (revised) version. The technical details required for handling these challenges in our convergence analysis are provided on page 37 of the original version  (page 42 of the revised version).
>
> [1] Liu, R., Liu, Z., Yao, W., Zeng, S., & Zhang, J. *Moreau Envelope for Nonconvex Bi-Level Optimization: A Single-Loop and Hessian-Free Solution Strategy.* In Forty-first International Conference on Machine Learning.
>
> [2] Yao, W., Yin, H., Zeng, S., & Zhang, J. *Overcoming Lower-Level Constraints in Bilevel Optimization: A Novel Approach with Regularized Gap Functions.* In The Thirteenth International Conference on Learning Representations.

---

> ### Author Response · Authors · 2025-11-22
>
> > Typo: (i) In Eq.(39), what is $t$ in $x_{i}^{t,k}$ ? (ii) The paper writes $\mathcal{F}_{0}:=\Omega,\phi$, is  $\phi$ from eq.(26)? Or do you mean empty set?
>
> **Response to typo:**
>
> Thank you for pointing this out.
> (i) In Eq. (39), the term $x_{i}^{(t,k)}$ should indeed be $x_{i}^{k}$. We have corrected this in the revised version.
> (ii) In $\mathcal{F}_{0} := \Omega, \phi$, the symbol $\phi$ was intended to represent the empty set. We will replace it with the notation $\varnothing$ and add a clarifying remark in the revised version.

---

### Official Review · Reviewer_S5im · 2025-10-26

**Soundness:** 3
**Presentation:** 3
**Contribution:** 2
**Rating:** 6
**Confidence:** 5

**Summary:**

This paper considers Decentralized Stochastic Bilevel Optimization (DSBO) problems, where both upper- and lower-level objectives can be nonconvex. To tackle the non-convexity in both levels, the authors first reformulates DSBO as a nonconvex–strongly-concave min–max problem using a Moreau envelope–based penalty method, then the authors propose SUN-DSBO (Structured Unified framework for Nonconvex DSBO), which allows use of first-order stochastic gradients only.

They provide solid convergence analysis of the convergence rates and concensus error of the algorithms, and they conduct numerical experiments to support the claims of their paper and showcase the advantages over existing DSBO algorithms.

**Strengths:**

1. This paper proposes a novel algorithm, which solves the DSBO problem with non-convex upper-level and lower-level objective functions. They provide solid convergence analysis of the convergence to stationarity and concensus among different agents. The authors also conduct some experiments to support their findings.

2. The algorithms achieve linear speedup effect, a non-trivial result in decentralized optimization literature.

3. Source code is provided.

**Weaknesses:**

1. One major limitation is, the idea and methodology of this paper seem to follow [1]. The proposed algorithms SUN-DSBO-SE/GT seem to be a direct extension of the centralized case considered in [1]. The techinques used in this paper, such as gradient tracking, consensus analysis, and convergence analysis are standard in existing distributed optimization literature.

2. The experiments seem to be limited on relatively simple examples/problems.

Ref:

[1] Moreau envelope for nonconvex bi-level optimization: A single-loop and hessian-free solution strategy.

**Questions:**

1. Could the authors provide some discussions on why combining the existing methods, such as Moreau envelope–based penalty method, gradient tracking, and convergence/concensus analysis is non-trivial in this nonconvex DSBO problem.

2. Would it be possible to consider some experimental setup in other types of problems, such as RLHF in [2]?

Ref:

[2] PARL: A Unified Framework for Policy Alignment in Reinforcement Learning from Human Feedback

---

> ### Author Response · Authors · 2025-11-22
> **Response to Weaknesses**
>
> We thank reviewer S5im for the detailed review, constructive suggestions, and helpful comments on both the theoretical comparison and empirical evaluation. We appreciate your careful reading and insightful feedback.
>
> > W1. One major limitation is, the idea and methodology of this paper seem to follow [1]. The proposed algorithms SUN-DSBO-SE/GT seem to be a direct extension of the centralized case considered in [1]. The techinques used in this paper, such as gradient tracking, consensus analysis, and convergence analysis are standard in existing distributed optimization literature.
> >
> > **Ref:**
> >
> > [1] Moreau envelope for nonconvex bi-level optimization: A single-loop and hessian-free solution strategy.
>
> **Response to W1**:
>
> Thank you for your concern, which gives us the opportunity to clarify the innovations of this paper from both the algorithmic design and theoretical analysis perspectives.
>
> From the algorithm-design perspective, although introducing the Moreau-envelope penalty method to decentralized bilevel optimization (DSBO) is conceptually natural, **no prior work** has explored this direction or verified that it works well for DSBO, at least experimentally. As stated in line 167 of the original (revised) version, our method is not a direct extension of [1]. Specifically, we reformulate problem (4) into the min–max structure in problem (5), and then construct a synchronized variable-updating framework based on this structure. In our opinion, this design viewpoint is simpler than that in [1]. A straightforward extension of [1] may require additional communication rounds to ensure convergence, since each variable update would need a separate communication step to refresh gradient information.
>
> From the theoretical-analysis perspective, several aspects differ substantially from [1]. First, unlike [1], our algorithm must account for the impact of **stochastic** gradient estimates. Even in the single-agent setting, the stochastic version of the Moreau-envelope penalty method remains underexplored, independent of the update mechanism.
> Second, in contrast to existing centralized bilevel algorithms that incorporate GT or other heterogeneity-correction techniques—where assumptions such as **bounded** gradient norms or **gradient similarity** are typically required—our SUN-DSBO-GT algorithm does not rely on these assumptions, as summarized in Table 1 in the appendix of the original (revised) version. The corresponding technical details in the convergence analysis can be found on page 37 of the original version  (page 42 of the revised version).
>
> > W2. The experiments seem to be limited on relatively simple examples/problems.
>
> **Response to W2**:
>
> Thank you for your comment. To demonstrate the effectiveness of our method and ensure a fair comparison with relevant works, we conducted experiments on the same tasks and datasets commonly used in prior DSBO studies. Specifically, we first evaluated our algorithm on the Data Hyper-Cleaning task using the Fashion-MNIST dataset with an MLP model, which is a standard setup adopted in existing studies [1, 2].
>
> To further assess performance, we conducted experiments on a larger and more challenging setting: the Hyper-Representation task on CIFAR-10 using a 7-layer CNN. This represents a relatively large-scale evaluation, and **very few** existing DSBO works have performed experiments at this scale. To the best of our knowledge, only [3] has used a CNN of comparable size on a small subset of CIFAR-10 for meta-learning experiments. (Since [3] focuses on differential privacy and introduces performance-affecting noise, its methods were not included in our comparisons.)
>
> In future work, we plan to further extend our experimental evaluation to more realistic and larger-scale applications, such as reinforcement learning.
>
> ##### **Ref**:
>
> [1] Kong, B., Zhu, S., Lu, S., Huang, X., & Yuan, K. (2024). Decentralized bilevel optimization: A perspective from transient iteration complexity. *arXiv preprint arXiv:2402.03167*.
>
> [2] Zhu, S., Kong, B., Lu, S., Huang, X., & Yuan, K. (2024). SPARKLE: A unified single-loop primal-dual framework for decentralized bilevel optimization. In *Advances in Neural Information Processing Systems*, 37, 62912–62987.
>
> [3] Chen, Z., & Wang, Y. (2024). Locally differentially private decentralized stochastic bilevel optimization with guaranteed convergence accuracy. In *Proceedings of the 41st International Conference on Machine Learning*, 235, 7389–7439. PMLR.

---

> ### Author Response · Authors · 2025-11-22
> **Response to Questions**
>
> > Q1. Could the authors provide some discussions on why combining the existing methods, such as Moreau envelope–based penalty method, gradient tracking, and convergence/concensus analysis is non-trivial in this nonconvex DSBO problem.
>
> **Response to Q1:**
>
> Thank you for your comment. We are pleased to discuss the non-trivial aspects of our analysis in addressing this problem.
>
> One of the key challenges in the convergence analysis of SUN-DSBO lies in handling the coupling between **heterogeneity**, **stochasticity**, and **dynamic penalty parameters**. When the penalty parameter is dynamic, it naturally increases the complexity of analyzing both heterogeneity and stochasticity. This is partly because the gradient structure changes at each iteration, making the analysis substantially more involved.
>
> To address these coupled sources of error, we design specialized **Lyapunov functions** that simultaneously control the consensus error and ensure overall convergence. In addition, **selecting appropriate step sizes** that guarantee convergence under this dynamic setting is itself a non-trivial aspect of the analysis.
>
> The technical details of this analysis can be found on page 37 of the original version  (page 42 of the revised version).
>
> > Q2.Would it be possible to consider some experimental setup in other types of problems, such as RLHF in [2]?
>
> **Response to Q2:**
>
> Thank you for your suggestion. We are currently extending our experiments to the RLHF setting using the BeamRider environment, where the single-agent version has already demonstrated strong performance in [I]. In that work, the authors train the reward model directly using ground-truth environmental rewards, providing a high-quality baseline for reward learning.
>
> Building on this paradigm, our goal is to investigate a networked multi-agent decentralized RLHF scenario, where different agents may encounter heterogeneous local states. Our focus is to examine whether the reward model and policy can remain consistent and convergent across agents under decentralized coordination, despite heterogeneity in their observations.
>
> These experiments are ongoing, and due to time constraints the results are not yet complete. If substantial results become available during the discussion period, we will update the submission accordingly and promptly inform the reviewers.
>
> [I] Yang, Y., Gao, B., & Yuan, Y. X. (2025). *Bilevel Reinforcement Learning via the Development of Hyper-gradient without Lower-Level Convexity*. AISTATS 2025.

---

### Official Review · Reviewer_sEex · 2025-11-01

**Soundness:** 3
**Presentation:** 3
**Contribution:** 3
**Rating:** 6
**Confidence:** 3

**Summary:**

This paper introduces SUN-DSBO, a Structured Unified framework for solving nonconvex decentralized stochastic bilevel optimization (DSBO) problems. The key idea is a Moreau-envelope based penalty reformulation that converts the bilevel structure into a nonconvex–strongly-concave min–max problem, allowing purely first-order, single-loop updates across distributed agents. Two algorithmic instances are studied in the paper: first, SUN-DSBO-SE, a decentralized SGD variant, and second, SUN-DSBO-GT, which incorporates gradient tracking (GT) for heterogeneous data. The authors establish finite-time convergence to stationary points under weaker assumptions than prior DSBO works without lower-level strong convexity, Hessians, or bounded-gradient conditions, and prove linear speedup in the number of agents.

**Strengths:**

1) This is the first DSBO method that rigorously handles *nonconvex–nonconvex* bilevel objectives in a decentralized, stochastic setting. Previous approaches (SPARKLE, D-SOBA, SLDBO) required lower-level strong convexity or Hessian-based hypergradients; SUN-DSBO works under far those milder assumptions.

2) The Moreau-envelope penalty and auxiliary variable (\theta) yield tractable gradient updates $((D_x,D_y,D_\theta))$ computable via mini-batch gradients, which remove all second-order dependencies.

3) The update directions depend linearly on both upper- and lower-level gradients, allowing plug-in communication schemes (GT, EXTRA, Exact-Diffusion). This unification could subsume several prior DSBO algorithms as special cases.

3) The convergence analysis does not assume bounded gradients or homogeneity; yet SUN-DSBO-GT achieves $(O(1/\sqrt{nK}))$ convergence and linear speedup.

4) SUN-DSBO-GT remains stable as network connectivity weakens (higher ρ), confirming its robustness to limited communication.

**Weaknesses:**

1) Only the GT strategy is analyzed theoretically; variants using **EXTRA** or **Exact-Diffusion** remain unexplored, and a unified convergence proof would strengthen the framework.

2) The paper lacks a systematic study of the penalty parameters (\mu) and (\gamma), which affect stability and constraint tightness.

3) Although GT achieves linear speedup, the communication-vs-computation trade-off (messages per ε-stationary point) is not quantified.

**Questions:**

- How sensitive is the method to the choice of penalty parameters (\mu_k) and (\gamma)? The paper mentions coarse-to-fine tuning, but it would be useful to know whether the method remains stable across a wide range of settings or if specific heuristics were necessary.

- For SUN-DSBO-GT, could the authors provide an analysis (empirical or theoretical) of the communication overhead versus the achieved linear speedup? For example, how many communication rounds per iteration are needed to maintain the reported convergence rate?

- In Section 3, the equivalence between the penalized formulation and the original bilevel objective seems to rely on certain smoothness or PL-type conditions. Can the authors clarify precisely which assumptions guarantee that optimizing the penalized $(\Psi_\mu)$ leads to a stationary point of the original bilevel problem when the lower-level is fully nonconvex?

- How does the method behave under sparse or dynamic communication graphs (larger $(\rho)$)? The current experiments show robustness, but it would be valuable to understand theoretical or empirical scaling limits with respect to network connectivity.

---

> ### Author Response · Authors · 2025-11-22
> **Response to Weaknesses**
>
> We thank reviewer sEex for the detailed review, constructive suggestions, and helpful comments on both the theoretical comparison and empirical evaluation. We appreciate your careful reading and insightful feedback.
>
> > w1. Only the GT strategy is analyzed theoretically; variants using **EXTRA** or **Exact-Diffusion** remain unexplored, and a unified convergence proof would strengthen the framework.
>
> **Response to W1:**
>
> We agree with you that developing a *unified convergence proof* that simultaneously covers EXTRA, Exact-Diffusion, and GT within our framework would strengthen the framework. Indeed, we mentioned in the conclusion of this paper that this is an interesting direction for future work, with relevant insights from SPARKLE [1]. We are also very happy to see any further work aiming to develop such a unified convergence theory, and we hope our contribution can provide some insights.
>
> Before we explain why we do not include such research here, we would like to say that, to the best of our knowledge, **no prior work** has provided theoretical guarantees for the GT strategy under decentralized bilevel optimization with nonconvex lower-level objectives.
>
> Next, let us discuss why extending our proofs for GT is technically challenging for two reasons:
> (a) The analytical techniques for ED, EXTRA, and GT differ substantially in the literature, and unifying them requires identifying fundamental mechanisms that remain valid even when the lower-level solution is non-unique.
> (b) The impact of dynamic penalty parameters on a unified analysis framework presents an additional challenge. While SPARKLE [1] offers a useful starting point, its analysis does not involve dynamic penalty parameters. Introducing such parameters causes the function and gradient forms to evolve with each iteration, significantly increasing the complexity of the analysis. This would require developing a more inclusive analytical framework capable of handling these changes.
>
> ##### **Ref**:
>
> [1] Zhu, S., Kong, B., Lu, S., Huang, X., & Yuan, K. (2024). SPARKLE: a unified single-loop primal-dual framework for decentralized bilevel optimization. *Advances in Neural Information Processing Systems*, 37, 62912–62987.
>
> > W2. The paper lacks a systematic study of the penalty parameters $\mu$ and $\gamma$, which affect stability and constraint tightness.
>
> **Response to W2**:
>
> We have already conducted an ablation study on how the penalty parameters $\mu$ and $\gamma$ affect the algorithm’s performance, as presented in Tables 2 and 3 of Appendix B.3 on pages 21–22 in the original version (pages 22–23 of the revised version), specifically in the last two subsections.
>
> > W3. Although GT achieves linear speedup, the communication-vs-computation trade-off (messages per ε-stationary point) is not quantified.
>
> **Response to W3:**
>
> From lines 5 and 6 of Algorithm 2 (SUN-DSBO-GT), it can be observed that each iteration requires **two rounds of communication**. Therefore, the total number of communications needed to reach an $\epsilon$-stationary point is simply twice the required number of iterations. Based on the finite-time analysis in Theorem 3.7, we obtain the communication complexity $\mathcal{O}\left(\frac{1}{n \epsilon^{2}}\right)$. Here, the communication complexity refers to the total number of communication rounds required to achieve an $\epsilon$-stationary solution.

---

> ### Author Response · Authors · 2025-11-22
> **Response to Questions**
>
> > Q1. How sensitive is the method to the choice of penalty parameters $\mu_k$ and $\gamma$? The paper mentions coarse-to-fine tuning, but it would be useful to know whether the method remains stable across a wide range of settings or if specific heuristics were necessary.
>
> **Response to Q1:**
>
> Thank you for your question.
>
> As discussed in Tables 2 and 3 of Appendix B.3 in the original version, our **ablation studies** suggest that the proposed method is relatively insensitive to the choice of the penalty parameters $\gamma$ and $\mu_k$. We further examine this behavior in Appendix C of the original version. To validate this observation more thoroughly, we additionally conducted experiments on Data Hyper-Cleaning with Fashion-MNIST, and Hyper-Representation with MNIST and CIFAR-10, using several commonly adopted parameter configurations, including:
>
> - $1/\gamma = 0.015$, $\mu_k = 2 (1 + k)^{-0.001}$
> - $1/\gamma = 0.02$, $\mu_k = 2 (1 + k)^{-0.001}$
> - $1/\gamma = 0.02$, $\mu_k = (1 + k)^{-0.001}$
>
> Across all datasets and parameter settings, the performance remained stable, and the variations introduced by changing $\gamma$ or $\mu_k$ were minimal. Overall, these results indicate that the method is reasonably robust across a broad range of penalty parameter choices.
>
> > Q2. For SUN-DSBO-GT, could the authors provide an analysis (empirical or theoretical) of the communication overhead versus the achieved linear speedup? For example, how many communication rounds per iteration are needed to maintain the reported convergence rate?
>
> **Response to Q2:**
>
> Thank you for your question. First, as mentioned in our response to **W3**, each iteration of **SUN-DSBO-GT** requires **two communication rounds**.
>
> Regarding the relationship between the communication overhead and the achieved linear speedup, since each iteration involves two communications, the total number of communication rounds is simply **twice the number of iterations**. The number of iterations needed before entering the linear-speedup regime corresponds to the **transient iteration complexity**, as defined in lines 320–322 of the original version (lines 327–329 of the revised version).
>
> As shown in **Corollary 3.8**, achieving linear speedup requires an iteration complexity of $\mathcal{O}\big(n^3 / (1 - \rho)^8\big)$. Therefore, the total communication complexity required to reach the linear-speedup phase is also
> $\mathcal{O}\big(n^3 / (1 - \rho)^8\big)$, because each iteration contributes exactly two communication rounds.
>
> > Q3.In Section 3, the equivalence between the penalized formulation and the original bilevel objective seems to rely on certain smoothness or PL-type conditions. Can the authors clarify precisely which assumptions guarantee that optimizing the penalized ($\Psi_{\mu}$)  leads to a stationary point of the original bilevel problem when the lower-level is fully nonconvex?
>
> **Response to Q3:**
>
> Thank you for your question. As you pointed out, if the lower-level objective satisfies the PL condition, optimizing the penalized objective $\Psi_\mu$ leads to the desired result. More generally, the PL condition is *not the only* sufficient condition; any condition that ensures that a critical point of the lower-level problem is a global minimizer would also suffice, as we discussed in item (i) of Section 2.1. For example, the **invexity condition** discussed in [1, Theorem 1 in Section 2] guarantees this property, and it is strictly weaker than the PL condition, as shown in [2, Section 2.2].
>
> ##### **Ref**:
>
> [1] Ben-Israel, A., & Mond, B. (1986). What is invexity? *The Journal of the Australian Mathematical Society Series B: Applied Mathematics*, 28(1), 1–9.
>
> [2] Karimi, H., Nutini, J., & Schmidt, M. (2016). Linear convergence of gradient and proximal-gradient methods under the Polyak–Łojasiewicz condition. In *Proceedings of the Joint European Conference on Machine Learning and Knowledge Discovery in Databases* (pp. 795–811).
> > Q4. How does the method behave under sparse or dynamic communication graphs (larger (\rho) )? The current experiments show robustness, but it would be valuable to understand theoretical or empirical scaling limits with respect to network connectivity.
>
> **Response to Q4:**
>
> Thank you for your question. Following your suggestion, we have extended our empirical study to include experiments under dynamic communication graphs. The new results are reported in the subsection *“Results with Dynamic Topology”* (page 23, Figure 13 in Appendix B.3 of the revised version).
>
> To more thoroughly assess robustness with respect to network connectivity, we considered two dynamic communication scenarios:
>
> - Scenario 1: Each agent randomly selects a *fixed* number of neighbors at every iteration.
> - Scenario 2: The number of neighbors is *random and varies* across iterations.
>
> Across both settings, our algorithm shows clear advantages in convergence speed and stability compared to baseline methods, especially SUN-DSBO-GT.

---

### Official Review · Reviewer_sbxh · 2025-11-01

**Soundness:** 3
**Presentation:** 3
**Contribution:** 3
**Rating:** 4
**Confidence:** 4

**Summary:**

This paper introduces SUN-DSBO, a Structured Unified framework for non-convex decentralized stochastic bilevel optimization (DSBO), which generalizes prior work by relaxing the common assumption of strong convexity in the lower-level problem. The authors present two algorithmic instances under this framework: SUN-DSBO-SE (based on decentralized SGD) and SUN-DSBO-GT (incorporating gradient tracking to address data heterogeneity).
The framework is built upon a Moreau-envelope-based penalized reformulation of the bilevel problem and further recast as a min-max formulation, allowing for efficient stochastic estimation. The authors provide rigorous theoretical guarantees for both algorithms under relaxed assumptions and validate their claims through comprehensive experiments on hyper-cleaning (Fashion-MNIST) and hyper-representation (MNIST & CIFAR-10), showing improvements over several state-of-the-art baseline algorithms.

**Strengths:**

1. The proposed SUN-DSBO framework broadens the scope of DSBO by addressing the nonconvexity of the lower-level objective, a setting largely unexplored in decentralized bilevel optimization.
2. SUN-DSBO is flexible enough to accommodate various decentralized schemes (e.g., GT, EXTRA), making it extensible and modular.
3. The paper provides finite-time convergence analysis for both SUN-DSBO-SE and SUN-DSBO-GT under realistic and relaxed assumptions. Notably, SUN-DSBO-GT achieves linear speedup with respect to the number of agents.

**Weaknesses:**

1. Although a direct comparison of the theoretical results with prior DSBO works may be somewhat unfair, given that those methods typically rely on additional assumptions such as strong convexity of the lower-level problem, it is still valuable to include such a comparison to highlight the differences and advancements introduced by this work.
2. While the paper claims that SUN-DSBO-GT introduces more overhead than SUN-DSBO-SE, no quantitative comparison of communication cost, memory usage, or wall-clock time is provided. This is particularly important in decentralized scenarios where bandwidth and device constraints are bottlenecks.
3. Although the framework claims to support a wide range of decentralized strategies (e.g., EXTRA, Exact Diffusion, and mixing strategies), only GT and SGD variants are empirically evaluated. This limits the practical demonstration of the claimed “unified” flexibility. A more comprehensive evaluation incorporating these additional strategies would significantly strengthen the contribution.
4. While the chosen tasks, hyper-cleaning and hyper-representation, are meaningful and relevant, exploring other bilevel applications such as decentralized meta-learning would help demonstrate the broader applicability and generality of the proposed framework.
5. The paper is notation-heavy, which may hinder readability. In particular, some notations could be streamlined or merged. For example, the distinction between $\hat{D}$ and $\tilde{D}$ in Line 191 could be reconsidered to simplify exposition.

**Questions:**

In Line 155, the authors state that $G(x,y)-V_\gamma (x,y)\ge 0 $ by the definition of $V_\gamma$. However, Equation (2) imposes the constraint $G(x,y)-V_\gamma (x,y)\le 0 $. Could the authors clarify how this constraint can be satisfied given the stated inequality?

---

> ### Author Response · Authors · 2025-11-22
> **Response to Weaknesses (1)**
>
> We thank reviewer sbxh for the thoughtful evaluation and the helpful perspectives offered throughout the review. We appreciate the time and careful attention devoted to reading our work. We value the feedback provided and will carefully incorporate it into our revision.
>
> > W1. Although a direct comparison of the theoretical results with prior DSBO works may be somewhat unfair, given that those methods typically rely on additional assumptions such as strong convexity of the lower-level problem, it is still valuable to include such a comparison to highlight the differences and advancements introduced by this work.
>
> **Response to W1:**
>
> Thank you for your comment. We will include a theoretical comparison with the main results from prior DSBO works in the setting where the lower-level problem is strongly convex, as summarized in the table below. In this context, $\sigma$ represents the heterogeneity level, and $\rho$ denotes the connectivity parameter of the communication matrix. (Since, to the best of our knowledge, the state-of-the-art convergence result under strongly convex lower-level objectives is achieved by SPARKLE, our comparison in the table focuses primarily on SPARKLE. A more comprehensive comparison can be obtained by combining **[1, Table 1]** in SPARKLE with the table provided here.)
> | Method | SPARKLE-GT | SUN-DSBO-SE (ours) | SUN-DSBO-GT (ours) |
> | --- | --- | --- | --- |
> | Oracles | Grad, HVP, JVP | Grad | Grad |
> | Asymptotic convergence rate | $\mathcal{O}\left(\frac{1}{\sqrt{nK}}\right)$ | $\mathcal{O}\left(\frac{1}{(nK)^{1/4}}\right)$ | $\mathcal{O}\left(\frac{1}{(nK)^{1/4}}\right)$ |
> | Transient Iteration complexity | $\mathcal{O}\left(\max\lbrace \frac{n^3}{(1-\rho)^2}, \\ \frac{n}{(1-\rho)^{8/3}} \rbrace\right)$ | $\mathcal{O}\left(\max\lbrace \frac{n^3}{(1 - \rho)^4},\\ \frac{n^3\sigma^4}{(1 - \rho)^4} \rbrace\right)$ | $\mathcal{O}\left(\frac{n^3}{(1 - \rho)^8}\right)$ |
>
>
> There are some remarks.
>
>
> (1) Our theoretical results in the strongly convex setting, presented in the table above, are obtained by directly setting $\gamma=\infty$ in the proof argument of this paper. This is valid because, even when $\gamma=\infty$, the min-max problem (5) retains the inner strong concavity structure needed in the analysis, which follows from the strong convexity of G.
>
> (2) It can be observed from the table above that our theoretical results do not match the best-known results provided by SPARKLE. However, recall that SPARKLE is an AID-based method that utilizes second-order derivative information, while our algorithm relies solely on gradient information. Moreover, even for centralized bilevel optimization, it remains an open question whether algorithms using only gradient information can achieve the theoretical results obtained by those that leverage second-order derivative information, see, e.g., the recent paper [2].
>
> [1] Zhu, S., Kong, B., Lu, S., Huang, X., & Yuan, K. (2024). SPARKLE: a unified single-loop primal-dual framework for decentralized bilevel optimization. *Advances in Neural Information Processing Systems*, 37, 62912–62987.
>
> [2] Chen, L., Li, J., & Zhang, J. (2025). Faster Gradient Methods for Highly-smooth Stochastic Bilevel Optimization. *arXiv preprint arXiv:2509.02937*.

---

> ### Author Response · Authors · 2025-11-22
> **Response to Weaknesses (2)**
>
> > W2. While the paper claims that SUN-DSBO-GT introduces more overhead than SUN-DSBO-SE, no quantitative comparison of communication cost, memory usage, or wall-clock time is provided. This is particularly important in decentralized scenarios where bandwidth and device constraints are bottlenecks.
>
> **Response to W2:**
>
> Thank you for the comment.
>
> First, we have already compared SUN-DSBO-SE and SUN-DSBO-GT in **Appendix B.3 (Ablation Studies)** on page 20 of the original submission (page 21 of the revised version). These experiments, conducted on both synthetic and MNIST hyperparameter optimization tasks, report the communication cost and runtime (Runtime refers to the total execution time of the algorithm, including communication time. The results reveal a clear trade-off:
>
> - SUN-DSBO-SE is more efficient in terms of runtime and total communication cost because each of its iterations is substantially cheaper; see Fig. 10 (left and middle) and Fig. 11 (left and middle).
>
> - SUN-DSBO-GT typically reaches the target performance in fewer iterations due to its more effective utilization of gradient information; see Fig. 10 (right) and Fig. 11 (right).
>
>
> Second, by comparing the iterations of SUN-DSBO-SE (line 5 of Algorithm 1) and SUN-DSBO-GT (lines 5–6 of Algorithm 2), we observe that each iteration of SUN-DSBO-GT requires **two rounds of communication**, whereas SUN-DSBO-SE requires only **one**. Moreover, SUN-DSBO-GT needs to maintain an additional tracking variable, which increases memory usage, whereas SUN-DSBO-SE does not require this extra state. The **per-iteration overhead for each agent $i$** in SUN-DSBO-SE and SUN-DSBO-GT can thus be summarized as follows.
>
> | Algorithm | Communication cost (sent) | Gradient storage (required memory) | Tracking variable storage (required memory) |
> | --- | --- | --- | --- |
> | SUN-DSBO-SE | $n_i(d_x + 2d_y)$ | $(n_i + 1)(d_x + 2d_y)$ | 0   |
> | SUN-DSBO-GT | $2n_i(d_x + 2d_y)$ | $(n_i + 1)(d_x + 2d_y)$ | $(n_i + 1)(d_x + 2d_y)$ |
>
> Here, $n_i$ denotes the number of neighbors of agent $i$, and to avoid any ambiguity, we define the amount of information sent as the communication cost (the amount of data received is the same as the data sent). $d_x$ and $d_y$ represent the dimensions of the variables $x$ and $y$, respectively, and the dimension of $\theta$ is also $d_y$.
>
> > W3. Although the framework claims to support a wide range of decentralized strateigies (e.g., EXTRA, Exact Diffusion, and mixing strategies), only GT and SGD variants are empirically evaluated. This limits the practical demonstration of the claimed “unified” flexibility. A more comprehensive evaluation incorporating these additional strategies would significantly strengthen the contribution.
>
> **Response to W3:**
>
> Thank you for your suggestion. In the revised version, we have included additional variants of SUN-DSBO to further demonstrate the flexibility of our unified framework. Specifically, we added experiments for SUN-DSBO combined with the corresponding heterogeneity-correction techniques, including **EXTRA**, **Exact Diffusion**, as well as mixed strategies that combine ED with GT and EXTRA with GT. These results are reported in **Figure 14** in Appendix B.3 (“Results with SUN-DSBO Extended Algorithms”, page 23) of the revised version, where we empirically verify that the framework can indeed accommodate a broader range of decentralized strategies beyond SGD and GT.
>
> > W4. While the chosen tasks, hyper-cleaning and hyper-representation, are meaningful and relevant, exploring other bilevel applications such as decentralized meta-learning would help demonstrate the broader applicability and generality of the proposed framework.
>
> **Response to W4:**
>
> Thank you for your suggestion. We are currently extending our experiments to decentralized meta-learning on the Omniglot and MiniImageNet datasets. These experiments are still in progress, and we do not yet have the full set of results. Preliminary findings are provided in **Figure 16** in **Appendix B.5 (page 30)** of the revised version, and we will continue to update the results and notify the reviewers as soon as additional outcomes become available.
>
> > W5. The paper is notation-heavy, which may hinder readability. In particular, some notations could be streamlined or merged. For example, the distinction between \hat{D}  and \tilde{D} in Line 191 could be reconsidered to simplify exposition.
>
> **Response to W5**:
>
> Thank you for the comment. We use **$\mathcal{D}$** consistently to denote the descent direction, where **$\tilde{D}$** refers to the deterministic gradient, **$\hat{D}$** denotes the corresponding stochastic gradient estimator, and **D** represents the update direction derived from these stochastic gradient estimators. We will clarify and emphasize these distinctions in the revised version of the paper.

---

> ### Author Response · Authors · 2025-11-22
> **Response to Questions**
>
> > **Q1.** In Line 155, the authors state that  $G(x,y)-V_\gamma(x,y)$ by the definition of $V_\gamma(x,y)$. However, Equation (2) imposes the constraint $G(x,y)-V_\gamma(x,y)\leq0$. Could the authors clarify how this constraint can be satisfied given the stated inequality?
>
> **Response to Q**:
>
> First, thank you for the question, which gives us the opportunity to clarify why the inequality constraint $G(x,y)-V_\gamma(x,y)\leq0$ is used in Eq. (2).
>
> A brief answer is that the purpose of doing so is to obtain a positive sign for the penalty parameter. Although the two constraints $G(x,y)-V_\gamma(x,y)\leq0$ and $G(x,y)-V_\gamma(x,y)= 0$ are equivalent, because $G(x,y)-V_\gamma(x,y)\geq0$ is intrinsic to the definition of $V_\gamma(x,y)$.
>
> Indeed, if the equality constraint $G(x,y)-V_\gamma(x,y) = 0$ was used in Eq. (2), then the corresponding penalty parameter would be a real number and could potentially be **negative**. In contrast, when the inequality constraint $G(x,y)-V_\gamma(x,y)\leq0$ is used in Eq. (2), the penalty method leads to the following penalty formulation:
>
> $$
> F(x, y) + \lambda \max\lbrace 0, G(x, y) - V_\gamma(x, y)\rbrace ,
> $$
>
> where $\lambda$ is the penalty parameter, which is now **positive**. By setting $\mu=1/\lambda$ and using the intrinsic condition $G(x,y)-V_\gamma(x,y)\geq0$, we obtain the following penalty formulation:
>
> $$
> \min_{x, y} \mu F(x, y) + G(x, y) - V_\gamma(x, y),
> $$
>
> which corresponds to Eq. (4) in our submission.
>
> Second, in numerical optimization, we generally cannot guarantee that the inequality constraint, such as $G(x,y)-V_\gamma(x,y)\leq0$ in our setting, will always be satisfied. This is precisely why we employ the penalty method and let the penalty parameter $\lambda$ tend to infinity (equivalently, $\mu = 1/\lambda$ gradually approaches zero in our formulation).

---

> ### Author Response · Authors · 2025-11-28
> **Supplement to the Response on W4**
>
> We have now updated our experiments on decentralized meta-learning and compared our framework with several popular algorithms to demonstrate the broader effectiveness and potential applicability of the proposed approach. Please refer to **Figure 16, Table 10, and Table 11** in Appendix B.5 (page 30) of the revised version.

---

### Author Response · Authors · 2025-12-03
**General Response (2)**

**Below are the comments regarding key theoretical aspects**:

1. Reviewer sbxh suggested a direct comparison of theoretical results with algorithms assuming strong convexity in the lower level.
  **Response**: We have made the corresponding comparison and provided some remarks in **Response to W1** for sbxh.

2. Reviewer sbxh raised a concern about the reason that the inequality constraint $G(x,y) - V_\gamma(x,y) \leq 0$ is used in Eq. (2).
  **Response**: A brief answer is that the purpose of doing so is to obtain a positive sign for the penalty parameter. More details are provided in **Response to Q** for sbxh.

3. Reviewer sEex suggested proposing a unified convergence proof that includes EXTRA and Exact Diffusion to strengthen the framework.
  **Response**: As we discussed in the Conclusion of the original version, we agree that developing a unified convergence proof would strengthen the framework. We have explained why we do not include such results in **Response to W1** for sEex.

4. Reviewer sEex raised the concern of the relationship between communication overhead and linear speedup for SUN-DSBO-GT.
  **Response**: Based on the fact that each iteration requires two rounds of communication from lines 5 and 6 of Algorithm 2 (SUN-DSBO-GT), we have discussed this in **Response to W3** and **Response to Q2** for sEex.

5. Reviewer sEex raised a concern about the assumptions that guarantee that optimizing the penalized objective $ \Psi_{\mu} $ leads to a stationary point of the original bilevel problem when the lower level is fully nonconvex.
  **Response**: While the PL condition guarantees the desired result, other conditions can also ensure that a critical point is a global minimizer, such as the invexity condition. More details are provided in **Response to Q3**.

6. Reviewer Rt4P raised the concern that the claim in line 163 that the constraint $G(x,y)-V_\gamma(x,y)\leq 0$ is equivalent to enforcing $\nabla_{y} G(x,y)=0$ under mild conditions.
  **Response**: As discussed in line 163 of the original version (the same line of the revised version), the equivalence statement is based on the condition that $G(x,\cdot)$ is $L_2$–smooth for any $x$ and $\gamma \in (0, 1/(2L_2))$, rather than convexity or PL assumptions. More details are given in **Response to W1** for Rt4P.

1. Reviewers S5im and Rt4P raised the concern that our method seems to be a direct extension of methods based on the Moreau envelope penalty combined with decentralized techniques, as well as the non-trivial aspects of our analysis.
   **Response**: We will clarify the innovations of this paper from both the algorithmic design and theoretical analysis perspectives. More details are provided in **Response to W1** and **Response to Q1** for S5im, and **Response to W3** for Rt4P.

---

### Author Response · Authors · 2025-12-03
**General Response (1)**

We sincerely thank all the reviewers for their insightful feedback. We are highly encouraged by their recognition of the unified framework of our algorithm (sbxh, sEex, Rt4P) and the convergence guarantees of the proposed methods (sbxh, sEex, S5im). We appreciate that reviewers (sbxh, sEex, S5im) underscored our contributions, particularly in using milder assumptions compared to prior works, such as addressing the non-convexity in the lower-level problem or relaxed conditions like gradient boundedness or specific heterogeneity assumptions. We are also pleased to see that they acknowledged our fully first-order algorithms (sEex), the proof of linear speedup with the number of agents (sbxh, sEex, S5im), and the experimental results that validate these theoretical findings (sbxh, S5im). Additionally, we are grateful for Reviewer Rt4P's positive feedback on the readability of the paper.

Next, we will briefly summarize the reviewers' concerns and our responses from both experimental and theoretical perspectives.

**Our comments on the key experimental aspects are as follows**:

1. Reviewer sbxh suggested a quantitative comparison of communication cost, memory usage, and wall-clock time between SUN-DSBO-GT and SUN-DSBO-SE.
  **Response**: The original submission already provided the comparison in **Figures 10 and 11** in Appendix B.3 on page 20 (page 21 of the revised version).

2. Reviewer sbxh suggested adding an empirical evaluation of a broader range of decentralized strategies (e.g., EXTRA, Exact Diffusion, and mixing strategies) beyond GT and SGD variants.
  **Response**: We have added the corresponding experiments in **Figure 14** in Appendix B.3 (page 24) of the revised version.

3. Reviewer sEex suggested a systematic study of the penalty parameters $ \mu $ and $ \gamma $.
  **Response**: The original submission already provided an ablation study and summarized the findings in **Tables 2 (on $ \mu $) and 3 (on $ \gamma $)** of Appendix B.3, in the final two subsections on pages 21–22 (pages 22–23 of the revised version).

4. Reviewer sEex suggested an empirical performance evaluation with respect to network connectivity, specifically under sparse or dynamic communication graphs.
  **Response**: We have extended our empirical study to include experiments under dynamic communication graphs in **Figure 13** in Appendix B.3 (page 23) of the revised version.

5. Reviewer Rt4P suggested an empirical evaluation of the performance of our algorithm based on the penalty function $ \mu F(x,y) + G(x,y) - V_\gamma(x,y) $, ensuring the approximation of the original nonconvex bilevel problem.
  **Response**: We have carried out synthetic numerical verification, which is shown in **Figure 15** of Appendix B.3 (page 24) of the revised version.

6. Reviewers sbxh and S5im suggested adding experiments in additional settings.
  **Response**: We have extended our experiments to include decentralized meta-learning, as presented in **Figure 16, Table 10, and Table 11** in Appendix B.5 (page 30) of the revised version. Additionally, we are extending our experiments to the RLHF setting, with results to be updated if available.


The code will be available at [https://anonymous.4open.science/r/SUN-DSBO-1616](https://anonymous.4open.science/r/SUN-DSBO-1616).

---

### Note · Authors · 2026-04-17

I have read and agree with the venue's withdrawal policy on behalf of myself and my co-authors.

---

### Meta-Review · Area_Chair_37t7 · 2025-12-22

**Summary:**

This paper introduces SUN-DSBO, a Structured Unified framework for Nonconvex DSBO, for solving decentralized stochastic bilevel optimization problems in which both the upper- and lower-level objective functions may be nonconvex. From the reviews and upon checking the paper by the AC, it is found that the main convergence results are not for the original nonconvex DSBO problem (1). First of all, (2) is not equivalent to (1). As the authors discussed in lines 162-166, under the assumption that G(x, .) is L2-smooth for any x and gamma is in certain interval, (2) is equivalent to a form that is equivalent to (1) only under the assumption of convexity or PL conditions. This is not reflected in the abstract, nor in the main results presented in Theorems 3.5 and 3.7. Moreover, the authors didn’t discuss the optimality measure used in Theorem 3.5 and 3.7. From the main results in these two theorems, it is not clear what the pair (\bar{x}^k,\bar{y}^k) represents. Specially, it is not clear at all whether it is some kind of approximate solution to the original problem (1). The authors are urged to clearly discuss these issues, as well as some other issues pointed out in the reviews, when they submit the work to a future venue.

**Reviewer Concerns:**

Addressed: More experimental results.
Outstanding: Clarification of the assumptions and discussion on optimality measure (by Reviewer Rt4P)

**Reviewer Scores:**

The authors added more experimental results and added some clarification on some of the presentation. But questions raised by Reviewer Rt4P regarding the optimality measure and clarification on the assumptions are still outstanding, and I don't think this reviewer will raise their score.

---

### Decision · Program_Chairs · 2026-01-26

Reject